# Implicit pattern learning predicts individual differences in belief in God in the United States and Afghanistan

Adam B. Weinberger [1,2,6✉], Natalie M. Gallagher [1,3,6], Zachary J. Warren[1,4], Gwendolyn A. English[1,5], Fathali M. Moghaddam[1] & Adam E. Green [1✉]

Most humans believe in a god, but many do not. Differences in belief have profound societal impacts. Anthropological accounts implicate bottom-up perceptual processes in shaping religious belief, suggesting that individual differences in these processes may help explain variation in belief. Here, in findings replicated across socio-religiously disparate samples studied in the U.S. and Afghanistan, implicit learning of patterns/order within visuospatial sequences (IL-pat) in a strongly bottom-up paradigm predict 1) stronger belief in an intervening/ordering god, and 2) increased strength-of-belief from childhood to adulthood, controlling for explicit learning and parental belief. Consistent with research implicating IL-pat as a basis of intuition, and intuition as a basis of belief, mediation models support a hypothesized effect pathway whereby IL-pat leads to intuitions of order which, in turn, lead to belief in ordering gods. The universality and variability of human IL-pat may thus contribute to the global presence and variability of religious belief.

[1] Georgetown University, Washington, DC 20057, USA. [2] University of Pennsylvania, Philadelphia, PA 19104, USA. [3] Northwestern University, Evanston, IL 60208, USA. [4] The Asia Foundation, 1779 Massachusetts Ave NW #815, Washington, DC 20036, USA. [5] ETH Zurich, 8092, Zurich CH-8092, Switzerland. [6]These authors contributed equally: Adam B. Weinberger, Natalie M. Gallagher. ✉email: abw58@georgetown.edu; aeg58@georgetown.edu

Religious beliefs, especially those pertaining to the influence of Gods, are among the most widely shared and deeply personally important human beliefs[1–4]. They are also among the most varied. Variability in religious belief and non-belief is as universal as the presence of religion across human cultures. Differences in belief have substantial impacts ranging from personal identity formation[5], to social/group affiliation and exclusion[6], to national and international political dynamics[7,8]. Key questions thus concern the neurocognitive bases of individual differences in belief. Theoretical accounts in the psychology and anthropology of religion suggest that religious beliefs emerge, at least in part, from perceptual mechanisms evolved for predictive processing of environmental information[1,2,9–16]. Though it is not possible to directly observe the co-evolution of religious beliefs with human information processing, differences among humans in religious and perceptual information processing phenotypes provide another window into such relationships. That is, individual differences in relevant perceptual mechanisms for predictive processing of environmental stimuli may bias individuals towards or away from religious beliefs.

A point of general consensus among explanatory frameworks for human information processing is the simultaneous operation of automatic bottom-up processes, driven primarily by sensory and perceptual input, and top-down processes that are more accessible to conscious awareness[17–19]. Critically, these two strata of cognition are not fully discrete from each other in their operation, and automatic bottom-up processes were likely the evolutionary substrate from which more explicit forms of top-down cognition arose[20]. Several theories of human cognition delineate trajectories of influence by which bottom-up processes direct individuals toward the formation of certain explicit beliefs[17,20–25]. According to these accounts, intuitions (i.e., reportable experiences of knowledge that was not consciously learned) develop as products of bottom-up gathering of environmental signals/information via unconscious information processing[20–25]. Because individuals are not aware of such bottom-up influences, intuitions drawn from unconscious processing may instead be consciously interpreted via explicit belief narratives that provide a rationalized context for beliefs and behaviors[2,23,24]. Indeed, intuitions frequently bias more explicit top-down views and judgments[2,13,17,26], and certain explicit beliefs may be more compelling and difficult to override when they stem from intuitive impressions[2,13,17,27]. These effects have been found to operate across diverse modalities of sensory information processing[28–30]. For example, in the context of interpersonal evaluations, humans rely on rapid, nonconscious face processing to form intuitions of trustworthiness[29–31], which has substantial influence on subsequent decision-making[31].

With respect to religious belief, extant theoretical accounts[1,2,9,10,13–16,32] posit that evolved neurocognitive processes contribute to default intuitions about the nature of environmental stimuli which direct individuals toward explicit beliefs. For instance, neurobiology evolved for cooperative interactions among humans is implicated as a basis for intuitions concerning morality and fairness[2,10,15]. These intuitions, in turn, are theorized to bias individuals toward religious beliefs in divine justice and watchful Gods, which align with moral intuitions, and may insulate against challenges to intuitions about justice when commensurate retribution or compensation for a given act is not materially available[33]. Other theoretical work has suggested a connection of evolutionarily-favored pathogen-avoidance mechanisms (e.g., evolved avoidance of individuals who are sick to minimize transmission of pathogens) to intuitions concerning cleanliness and religious beliefs concerning purity (e.g., belief in the healing effects of touching holy individuals or items, belief in unseen agents of spiritual corruption)[16,34]. Intuitions of

anthropomorphism, which appear to be biologically rooted in systems evolved to support face processing and social-cognition[35], may contribute to beliefs in watchful invisible agents[36,37]. Relatedly, the automatic bias to detect agency in the environment is thought to yield over-attribution of intentional agency at the intuitive level (i.e., intuitions that non-agentic things have agency), which supports explicit beliefs about supernatural, intelligent agents (e.g., deities)[1,13,14]. Empirical support for this theory, however, has been mixed; recent work has suggested that individuals may under attribute agency[38], and that believers in the paranormal—but not those who endorse a set of more traditional religious beliefs—display illusory agency detection[39,40].

Other literature has suggested a link between top-down, effortful analytic thinking—a propensity to critically examine and override prepotent automatic[41] responding—and religious disbelief[41,42]. Gervais and Norenzayan[43] reported that priming analytic thinking reduced strength of religious belief. However, replication attempts have not supported the efficacy of the so-called analytic prime employed by Gervais and Norenzayan[44,45].

A core belief across major religions is that the sequence and structure of events in human lives (and in the universe more broadly) reflects an underlying order determined by the intervention of Gods[3,4,46–50]. Believers are more inclined to perceive events in the world as adhering to a purpose or design rather than as a series of random, unpredictable occurrences[47,49]. Indeed, such interventionist belief is an explicit focus of religious practices in Christianity, Islam, Hinduism, and Judaism[50]. Empirical work has associated belief in Gods with explanations of events as adhering to an intelligently designed order[49,51,52], and greater belief that a deity plays a role in ordering life events is posited to lead to increased strength of religious belief broadly[50,53]. This suggests that identifying individual differences that influence interventionist belief may provide explanatory insights into variations in strength of religious belief in general. Moreover, because belief in intervening/ordering Gods is a shared element of disparate religions[3,50], it provides a suitable testbed for hypotheses concerning links between information processing and belief that may transcend cultural context.

In view of the associations between intuition and religious beliefs, it is notable that multiple lines of work in cognitive science have specifically implicated order-related perceptual information processing as a basis of intuitions[21–23,54]. In particular, this work points to bottom-up learning of predictive order (patterns) in the environment without conscious awareness (i.e., implicit pattern learning; IL-pat)[20–23,25,54] as an underlying influence. Reber[21] identifies IL-pat as a bottom-up perceptual basis of intuitions of order, including tacit-knowledge of complex visuospatial patterns after extended exposure. This work is situated within a broader framework developed by Epstein and others in which, "The implication of automatic implicit learning from experience is that the information acquired… is the primary source of intuitive 'knowing without knowing how one knows'"[23], and in which such intuition serves as a bridge between implicit and explicit levels of cognition[23,24,54]. Convergently, research into individual differences in IL-pat[55–58] has demonstrated that IL-pat is positively associated with self-reported measures of intuition across several paradigms.

The work identifying bottom-up information processing and intuitions rooted in evolved neurocognitive systems as influences on belief suggests the hypothesis that processing information related to order in the environment via IL-pat could influence the formation of belief in an intervening/ordering God. Specifically, accounts implicating IL-pat as a basis of intuitions[21–23,54], and intuition as a basis of beliefs[2,4,9,15,17,24], indicate a potential pathway of effect whereby implicit learning of actual patterns/

order in the environment could yield intuitions of ambient order that, in turn, influence belief in an ordering God. In other words, intuitions of order that result from IL-pat might disproportionately direct stronger implicit learners toward explicit beliefs in an intervening/ordering God.

Implicit pattern learning occurs frequently in real-world settings and operates on a broad range of stimuli, including many forms of visual modality input[59]; merely being in an environment with recurring elements is sufficient to implicitly learn associations[60,61]. Individual differences in implicit learning emerge early in development[60,62,63], with performance approximating adult levels[64]. While extended longitudinal measurement is challenging, extant evidence suggests that differences in IL-pat are relatively stable across time[65], including during early childhood[66], and may be genetically determined[67,68]. Thus, any impacts of IL-pat on intuitions of order are likely to begin in young childhood via exposure to learnable patterns in the environment. If IL-pat affects belief, it could thus manifest in individual differences in change in strength of religious belief from childhood to adulthood.

Because IL-pat is a fundamental, largely subcortically-mediated aspect of perceptual information processing[69,70], it is likely to be present (and subject to individual differences) in all human groups. Thus, if IL-pat helps shape belief, then individual differences in IL-pat should be predictive of differences in belief across disparate cultures and religions. As noted above, belief in intervening/ordering Gods is a common element across disparate religions[3,4,46–48,50], thus this aspect of belief presents an opportunity to evaluate the influence of IL-pat across religions. Moreover, beyond the context of this study, a timely priority for empirical understanding of religious cognition is the inclusion of non-Western cultural contexts that are substantially underrepresented in this literature[71]. Middle Eastern Islamic samples, in particular, are largely absent from psychological research broadly. Turkish Muslims are a relative exception[72–74], but less westernized nations in the region are dramatically underrepresented, such that sociocultural influences on cognitive and behavioral phenotypes (religious and otherwise) remain poorly understood[75,76]. In addition to the inherent value of expanding psychological research in the Middle East, investigating the replicability of findings across U.S. and Middle Eastern samples provides the opportunity for strong tests of mechanisms hypothesized to operate across disparate sociocultural contexts.

Multiple experimental paradigms have been developed to investigate IL-pat. Perhaps the most widely used measure is the serial reaction time task (SRTT)[77,78], which prior work suggests is reflective of environmental IL-pat[21], and which has recently been shown to load heavily on a broad implicit learning ability factor[65]. Notably, extant evidence indicates that IL-pat during the SRTT is not improved—and may actually be impaired—by top-down influences, including instructions to look for patterns[61,69,79,80] (see "Methods"). In addition, modifications of the SRTT have been devised to limit opportunities for explicit awareness of patterns that could be a precondition for top-down influence[69,81]. The SRTT thus presents a good experimental measure to investigate putative bottom-up link of IL-pat to belief: stronger IL-pat might contribute to stronger belief, but a belief-related top-down bias to seek patterns is unlikely to strengthen IL-pat, thus, any positive association observed between IL-pat and belief is most likely to be bottom-up.

Here, in culturally and religiously distinct samples studied in Washington, D.C. and Kabul, Afghanistan, we test and support the hypotheses that individual differences in implicit pattern learning, measured via a modified SRTT[77,78,82], predict individual differences in (1) strength of belief in an intervening/ordering God, and (2) change in strength of belief in God from childhood

to adulthood. We distinguish effects of IL-pat from potential confounding influences of schizotypal ideation and parental religious belief as well as potential effects of explicit learning. Subsequent analyses in the U.S. sample further support the hypothesis that intuitions of universal order mediate the relationship of IL-pat to belief. Data from a predominantly European sample provide additional replication and extension. Results indicate that superior bottom-up, implicit learning of visuospatial patterns is associated with stronger belief in an intervening God and increased strength of belief from childhood to adulthood.

## Results

**Implicit pattern learning (IL-pat).** Because a goal of this research was to test whether results replicated across religiously and culturally disparate contexts, all analyses were performed separately for each of the samples.

U.S. ($N = 199$, $M_{age} = 19.83 \pm 2.72$ years, 65.83% female, 34.17% male, 52.26% Christian, 25.13% unaffiliated, all others <4%, Supplementary Table 1) and Afghan ($N = 148$, $M_{age} = 26.99 \pm 4.57$ years, 41.22% female, 58.78% male, religious affiliation not queried due to potential risks associated with non-Islamic affiliation) participants completed a modified SRTT[78,82], a widely used measure considered to be reflective of ecological implicit learning ability[21]. Participants responded to the positions of filled circles (targets) that rapidly appeared at four positions onscreen by pressing a corresponding key ("Methods"; Fig. 1). Each block of targets either adhered to a repeating 10-position sequence (pattern), or was random (50% pattern blocks; 50% random blocks). Consistent with recommended procedures[77], implicit pattern learning was calculated as the difference between slope-of-change in response time (RT) during random vs. pattern blocks to enable measurement of faster responding due to learning distinct from confounding influences (e.g., motivation, familiarity).

In order to verify that pattern blocks were distinguishable from random blocks at the implicit level (i.e., based on RT differences), we conducted paired $t$-tests for each sample between the rate of change for pattern and random blocks. Both U.S. and Afghan participants displayed a faster rate of change on pattern blocks, indicating that they were able to implicitly learn the patterns

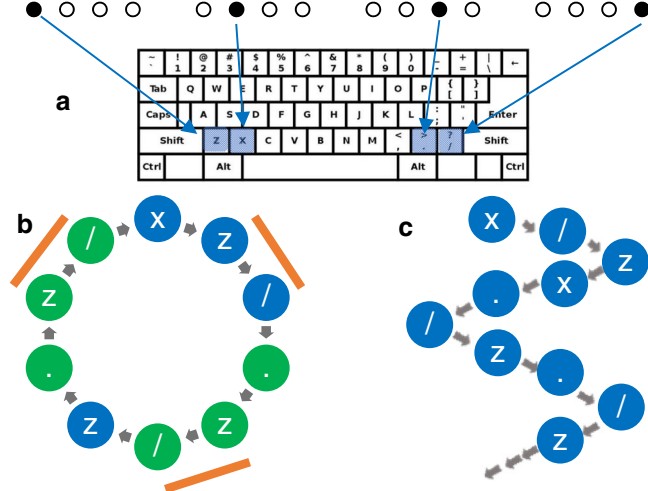

**Fig. 1 SRTT.** Participants indicated target locations corresponding to four keys (**a**). For example, when the target appeared in the left most location, participants responded by pressing the "z" key. Pattern blocks (**b**) consisted of 10-target repeating (5×) sequences, containing three first-order structures (orange bar) and two second-order structures (green circles). Random blocks (**c**) consisted of 50 non-repeating targets.

(U.S.: $t(198) = -10.95$, $P < 0.0001$; Afghanistan: $t(147) = -7.61$, $P < 0.0001$). To further evaluate whether learning was implicit, following each block, participants were asked to indicate whether they thought the sequence they saw was a pattern or was random. In both samples, IL-pat was uncorrelated with explicit accuracy about whether patterns were present (U.S.: $r = -0.12$, $P = 0.10$; Afghanistan: $r = 0.08$, $P = 0.40$), confirming that learning was implicit. Relatedly, IL-pat was not associated with illusory explicit pattern detection, i.e., incorrectly reporting the presence of a pattern on random blocks (U.S.: $r = 0.03$, $P = 0.67$; Afghanistan: $r = 0.01$, $P = 0.93$).

**Relationship of IL-pat to belief.** An interventionist belief (IB) component score (see "Methods"; Supplementary Methods) was calculated from three belief measures: the Belief in Divine Intervention Scale[3] (BDIS; Supplementary Table 3) and two versions of the Overlapping Circles Task[83] (Supplementary Fig. 1), in which participants used object representations to indicate beliefs about the extent to which God influences events in the world and their own actions. Briefly, IB was created in order to integrate estimates of belief in divine intervention obtained from all three belief measures into a single DV for regression models (see "Methods").

Next, linear regression models in each sample tested our hypothesis that IL-pat predicted IB. We controlled for schizotypal ideation and parents' strength of belief to distinguish effects of IL-pat from key psychological and environmental factors that are associated with religious belief[84,85] and biases to perceive order[86,87] but distinct from the hypothesized influence of IL-pat (see Supplementary Tables 10 and 11). Implicit pattern learning significantly predicted IB in both the U.S. ($b = 1.24$, $\beta = 0.17$, $SE = 0.47$, $P = 0.009$) and Afghanistan ($b = 1.60$, $\beta = 0.26$, $SE = 0.48$, $P = 0.001$; Fig. 2). Because of the non-normal distribution of IB (Supplementary Fig. 2), we conducted additional analyses, which confirmed that the observed associations with IL-pat were not due to a violation of OLS assumptions (Supplementary Table 9).

We further performed zero-order correlations (Supplementary Tables 5 and 6) with scores on the individual intervening belief measures to confirm that the relationship between IL-pat and IB was not a result of associations with just one or two of the belief measures from which IB was derived. Implicit pattern learning was positively associated with BDIS at a trend level in the U.S.

($r = 0.13$, $P = 0.06$) and significantly in Afghanistan ($r = 0.19$, $P = 0.02$). In both samples, implicit pattern learning was also significantly associated with self-overlap (U.S.: $r = 0.19$, $P = 0.007$; Afghanistan: $r = 0.19$, $P = 0.02$) and world overlap (U.S.: $r = 0.16$, $P = 0.03$; Afghanistan: $r = 0.26$, $P = 0.001$). These findings indicated a consistent underlying association between implicit pattern learning and diverse individual measures of belief in an interventionist God across these two culturally disparate samples.

Belief change was surveyed using a lab-developed measure, which consisted of 9-point Likert scales on which participants reported their own strength of belief in God starting at age 6 and then at 3-year intervals up to age 24 (see "Methods", Supplementary Methods). This method to assess change in religious belief is consistent with prior work that has used a retrospective approach to assess changes in religiosity over the lifespan[41,88].

IL-pat was a significant predictor of belief change in both the U.S. ($b = 3.53$, $\beta = 0.15$, $SE = 1.67$, $P = 0.036$) and Afghanistan ($b = 3.53$, $\beta = 0.17$, $SE = 1.67$, $P = 0.036$; Fig. 2), with schizotypal ideation and parents' strength of belief again included as covariate regressors (Supplementary Tables 10 and 11).

In order to further confirm that implicit pattern learning—rather than explicit awareness—was driving the association with religious belief, we performed zero-order correlations for explicit accuracy on the SRTT with IB and belief change. Explicit accuracy was not related to IB (U.S.: $r = 0.07$, $P = 0.30$; Afghanistan: $r = 0.13$, $P = 0.14$) or belief change (U.S.: $r = 0.06$, $P = 0.42$; Afghanistan: $r = 0.06$, $P = 0.51$). We also tested the association of IB and belief change with illusory explicit pattern detection (i.e., reporting patterns for random blocks) and more frequent endorsements of patterns regardless of block type. IB was not correlated with illusory detection (U.S.: $r = 0.08$, $P = 0.23$; Afghanistan: $r = 0.15$; $P = 0.09$) or overall tendency to reported patterns (U.S.: $r = 0.04$, $P = 0.56$; Afghanistan: $r = 0.06$, $P = 0.51$). Similarly, we did not observe an association between belief change and illusory detection (U.S.: $r = 0.11$, $P = 0.11$; Afghanistan: $r = 0.05$, $P = 0.55$) or an overall bias towards reporting patterns (U.S.: $r = 0.09$, $P = 0.21$; Afghanistan: $r = 0.02$, $P = 0.86$). When these explicit awareness variables were included as additional covariate regressors (along with parents' belief and schizotypal thinking; Supplementary Tables 12, 13), IL-pat remained a significant predictor of both IB (all $Ps \le 0.009$) and belief change (all $Ps \le 0.05$).

Finally, we conducted an initial probe of the relationship between IL-pat and belief in the existence of God broadly (Existence Belief; EB) using a single-item measure in the U.S. sample only (no viable translation of this item could be found for the Afghan sample; see Supplementary Methods). As expected, EB was positively correlated with IB ($r = 0.66$, $P < 0.0001$). However, exploratory analyses to assess whether EB was associated with IL-pat revealed no correlation ($r = 0.05$, $P = 0.50$). Thus, while IB and EB were strongly correlated, only the former was associated with IL-pat. Because EB was closely related to IB in some participants (but not in others), we next asked whether IL-pat was more associated with Existence Belief in individuals for whom IB more closely approximated Existence Belief and less associated with Existence Belief in those for whom Existence Belief was more divergent from IB (putatively explaining the discrepant relationships of IL-pat to IB vs. EB). After calculating similarity between EB and IB for each participant (see Supplementary Methods), we explored whether IL-pat differentially predicted EB based on how closely EB and IB were related (i.e., a belief similarity X IL-pat interaction). This model revealed a significant interaction, such that the association between IL-pat and Existence Belief was stronger for those

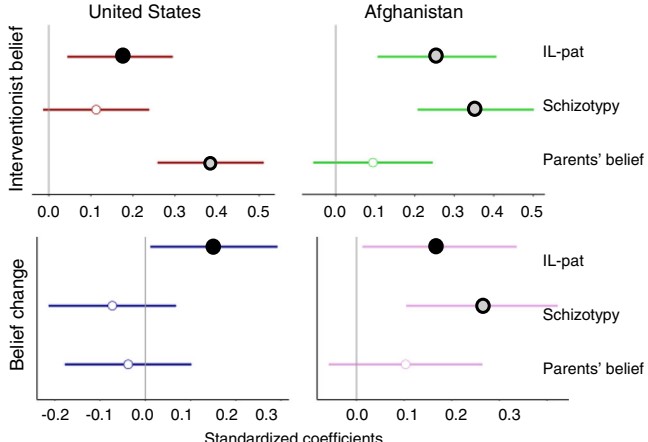

**Fig. 2 Linear regression models to predict interventionist belief (IB) and belief change (BC) in the U.S. and Afghan samples.** Significant $\beta$ are indicated by filled circles. Black fill indicates $P < 0.05$ ($P = 0.036$ for effect of IL-Pat on belief change in both samples; $P = 0.009$ for IL-Pat on IB in U.S. sample) and gray fill indicates $P \le 0.001$. Error bars represent 95% CI.

participants whose IB more closely approximated Existence Belief ($\beta = -0.20$, $P = 0.002$). Taken together, these post hoc analyses demonstrate that, consistent with our hypothesis concerning IL-pat as a contributor to IB, IL-pat was more predictive of Existence Belief in participants for whom the level of Existence Belief was closely tied to the level of IB, and less predictive in those whose belief in God appeared to be less tied to belief in the intervening influence of God (i.e., those who report higher levels of Existence Belief relative to IB).

**Relationship of IL-pat and belief to intuitions of order.** Having identified relationships between IL-pat and explicit belief, we sought to further test the hypothesis that intuitions about order in the universe might mediate these relationships. Given prior work indicating that IL-pat contributes to intuitions of order[17,20–23] and that intuitions contribute to religious belief[1,2,10–14,16,32], we hypothesized a pathway of effect in which IL-pat contributes to intuitions of order that incline individuals toward religious belief narratives. The hypothesized directionality of this pathway was also based on theoretical considerations, e.g., that intuitions of order can plausibly develop from implicit learning of order even without explicit learning of religious narratives, and that the content of what is learned via IL-pat (i.e., order itself) is conceptually closer to intuitions directly concerning order than to religious beliefs about interventionist deities (see "Discussion").

Participants were presented with statements concerning the presence of order in the universe but not referring to religion or God, and indicated their level of agreement (1–9; see "Methods"; Supplementary Table 4). Responses were summed to produce a universal order (UO) score. Agreement with presented statements has been shown to provide an effective explicit measure of intuition[89]. Because data collection for these questions in the Afghan sample was not deemed viable (i.e., Afghan experimenters indicated that the Dari-translated questions were not properly understood; see "Methods"), we measured UO and IB in a separate, predominantly European replication sample online ($N = 96$, $M_{age} = 28.21 \pm 9.31$ years, 83.33% European, 61.46% female, 37.50% male; see "Methods", Supplementary Table 2).

First, we asked whether UO and IL-pat were associated with one another (i.e., Fig. 3, a path). Zero-order correlations indicated that UO was positively correlated with IL-pat ($r = 0.20$, $P = 0.005$). Based on indications that schizotypal thinking may be associated with a tendency to perceive order[86,87], and to account for potential influences of parental religious belief[84,85], we included schizotypal thinking and parents' religious belief as covariates in a linear regression model to predict UO. IL-pat remained a significant predictor of UO in this model ($b = 5.12$, $\beta = 0.18$, $SE = 1.90$, $P = 0.008$, Supplementary Table 10).

UO was also significantly correlated with both IB (U.S.: $r = 0.48$, $P < 0.001$; European sample: $r = 0.56$, $P < 0.001$) and belief change (U.S. sample: $r = 0.36$, $P < 0.001$; Fig. 3, b path). Consistent with our hypotheses, UO significantly mediated the effects of IL-pat on both IB (indirect effect $P = 0.005$) and belief change (indirect effect $P = 0.006$; bootstrapped bias-corrected 95% CI; Fig. 3) in the U.S. sample. These analyses do not establish causation because mediation analysis is fundamentally correlational. Nonetheless, results suggest that the a priori directional hypothesis remains plausible a posteriori in view of the data. By contrast, if modeling the pathway had not yielded significant indirect effects, this would have empirically indicated against the hypothesized model. Though the hypothesized pathway of effect is based on theory and prior literature, rather than empirical comparisons between competing models, further exploratory analyses (Supplementary Methods, Supplementary Figs. 6 and 7) in which IB and individual belief measures were modeled as

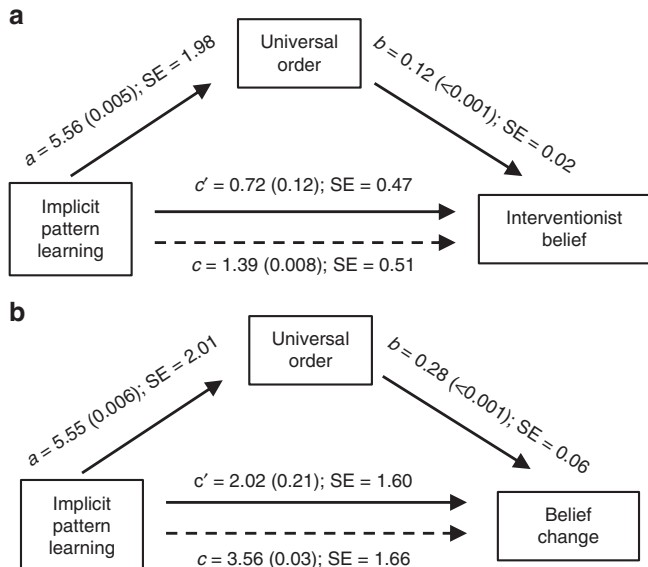

**Fig. 3 Explicitly reported intuitions of universal order (UO) mediate effects of implicit pattern learning on belief. a** IB and **b** belief change in the U.S. Sample. The influence of implicit pattern learning on religious belief without (dashed lines) and with (solid line) UO as a regressor is shown. The model displays the unstandardized coefficients for each path, with $P$ values indicated parenthetically.

mediators, and in which IB was modeled as the independent variable and IL-pat as the dependent variable, provided modest but consistent additional support for the hypothesized model.

**Cognitive Reflection Test (CRT).** The above results supported hypothesized links of intuition to IL-pat and belief—these hypotheses specifically concerned intuitions of order (i.e., UO). Subsequently, we explored a secondary question concerning the CRT[90], a measure of calculation-based problem-solving devised to distinguish responding based on a more effortful and analytic approach from responding based on a less effortful and analytic (and more error-prone) approach; the latter approach has been referred to as intuitive thinking. Investigation of the CRT was motivated by prior research that has suggested a connection between nonbelief and analytic problem-solving on the CRT[42,44,45,91–93]. It was thus of interest to determine whether IL-pat was related to CRT performance, and whether the associations of IL-pat to belief and to intuitions of order (as measured by UO) were independent of the kind of analytic vs. intuitive problem-solving measured by the CRT. All U.S. participants who completed the full study battery were re-contacted, and respondents (U.S. re-contact sample; $N = 65$) completed the CRT. The CRT consists of three arithmetic problems. For each problem, there is a response that is considered to be prepotent (i.e., the first answer that comes to mind), which is referred to as the intuitive response, whereas calculating the correct answer is posited to require greater analytic, top-down effort. Correct (0–3) and intuitive (0–3) responses are summed for each participant. Incorrect responses that are not the intuitive response are ignored in the calculation of correct and intuitive scores. Following Frederick, the number of correct responses was interpreted as an estimate of analytic problem-solving, or the extent to which individuals are able to override prepotent (incorrect) responses.

Consistent with the associations observed in the full U.S. sample, IL-pat was significantly correlated with IB ($r = 0.46$, $P = 0.0001$), belief change ($r = 0.39$, $P = 0.002$), and UO ($r = 0.39$, $P = 0.001$; Supplementary Table 14). Also consistent with results

obtained from the full sample, there was no association between IL-pat and accuracy of explicit pattern judgments ($r = -0.18$, $P = 0.16$), indicating that learning in this sub-sample was implicit. Results for the CRT indicated that analytic (i.e., correct) and intuitive (incorrect) responses were strongly negatively correlated with each other (both samples $r < -0.85$, $P < 0.0001$), as in prior research[90]. In the analyses of greatest interest, we found that IL-pat was not associated with analytic responding ($r = -0.07$, $P = 0.56$) or intuitive responding ($r = -0.03$, $P = 0.84$; Supplementary Table 14). Linear regression models in which CRT performance was added as an additional covariate regressor (in addition to schizotypal thinking and parents' belief) indicated that IL-pat remained a significant predictor of IB ($b = 3.22$, $\beta = 0.43$, $SE = 0.81$, $P < 0.001$), belief change ($b = 9.88$, $\beta = 0.40$, $SE = 2.96$, $P = 0.001$), and UO ($b = 11.64$, $\beta = 0.38$, $SE = 3.61$, $P = 0.002$; Supplementary Table 16). Together these results indicate that the relationship of IL-pat to IB, belief change, and UO was meaningfully distinct from CRT-based problem-solving approach. That is, it is unlikely that individual differences in the tendency to effortfully apply analytic thinking to override prepotent responding (at least in the context of CRT calculation problems) are responsible for variation in IL-pat or the association between IL-pat and belief. The CRT was also collected online in the European replication sample. Generally consistent with prior work associating analytic thinking more closely with disbelief than belief[41,42], associations were found between IB and CRT performance (analytic: $r = -0.22$, $P = 0.03$; intuitive: $r = 0.23$, $P = 0.02$; Supplementary Table 15). These relationships were not significant within the U.S. re-contact sample, though opposite directions/signs were again observed for the nominal associations of IB to analytic ($r = -0.10$, $P = 0.41$) and intuitive ($r = 0.14$, $P = 0.27$) responding (Supplementary Table 14).

**Assessment of belief in science.** Last, we investigated whether associations similar to those observed for belief in an interventionist God (i.e., associations with intuitions of universal order and implicit pattern learning) might also be observed for belief in science, a construct that may plausibly relate to explanations of order in the universe. All participants in the U.S. re-contact and European samples completed the Belief in Science Scale[94]. This scale consists of 10 questions intended to measure attitudes toward science, such as its value, reliability, and ability to provide an understanding of the world and human culture.

Consistent with prior research that has examined relationships between religious belief and scientific belief[94], zero-order correlations indicated that scientific belief was inversely associated with IB (U.S. re-contact sample: $r = -0.22$, $P = 0.08$; European sample: $r = -0.36$, $P = 0.0003$) and belief change (U.S. re-contact sample only: $r = -0.28$, $P = 0.02$). Despite this negative direct correlation, it remained possible that stronger intuitions of universal order and/or stronger IL-pat might be associated with stronger belief in science if, for example, a subset of participants who held strong intuitions of order in the universe were biased toward scientific interpretations of such order. However, results indicated this was not the case; belief in science was unrelated to UO (U.S. re-contact sample: $r = -0.19$, $P = 0.13$; European sample: $-0.11$, $P = 0.27$; Supplementary Table 15) and negatively correlated with IL-pat (U.S. re-contact sample: $r = -0.28$, $P = 0.02$; Supplementary Table 14).

## Discussion

The present study explored the relationship between implicit pattern learning and belief in Gods in the U.S. and Afghanistan, countries that differ substantially along multiple cultural and religious dimensions[7,95–97]. In findings replicated across these samples, individuals exhibiting stronger implicit learning of order in visuospatial sequences held stronger belief in an intervening/ ordering God, and increased more in strength of belief from childhood to adulthood. UO mediated these effects in the U.S., suggesting that IL-pat may be associated with interventionist belief because it yields intuitions of order that bias individuals toward belief in ordering Gods. Neither IL-pat nor belief was associated with explicit awareness of learned patterns, indicating that the observed effects are specific to learning of patterns without conscious awareness.

The present work builds on substantial literatures indicating implicit learning as a basis of intuition[21–23,54], and intuition as a basis of belief[2,11,12,14,15,17,26]. Our findings accord with theories in the psychology and anthropology of religion[1,2,9,10,12–16,32] that account for religious belief as a reflective elaboration[2] on intuitions derived from bottom-up processing of environmental stimuli. The present findings support a conceptually aligned hypothesis that individuals who more readily learn, at an implicit level, patterns that are actually present in the environment may be biased toward belief in ordering Gods. Conversely, those who less readily learn available patterns via IL-pat may be less predisposed toward such beliefs.

The results of the current study are consistent with a bottom-up pathway of effect by which stronger implicit learning of patterns/order leads to belief in an intervening/ordering God. While top-down pathways are also conceivable whereby stronger belief in an intervening deity leads individuals to more effectively learn implicit patterns, and/or whereby religious individuals may be more likely to search for patterns because of an explicit belief in the ordering influence of a deity, a number of empirically-based indicators make such explanations less likely in this case. First, prior research has demonstrated that explicit searching does not lead to faster responding during the SRTT, and may actually hinder performance[61,69,79]. For example, Fletcher and colleagues[79] showed that explicitly directing participants to search for a pattern impeded implicit learning on the SRTT (i.e., slower RT during the task), putatively because this explicit search disrupts elements of task performance that facilitate IL-pat. Thus, even if interventionist belief promotes explicit searching for patterns, it is unlikely that this would have led to improved performance on the SRTT in the present study.

A bottom-up interpretation of the present data is further bolstered by modifications to the SRTT paradigm that were incorporated into the present study. Namely, the sequence advanced (i.e., the next target appeared) immediately after the participant provided a correct response. This so-called no-response-time-interval (no-RSI) version of the SRTT has been demonstrated by Destrebecqz and Cleeremans and others[69,81] to yield sequence learning without explicit knowledge acquisition. Removing time between target presentations is intended to ensure that learners are consistently engaged in the task and do not have the opportunity to consciously attempt to recall/rehearse the target sequence in an effort to identify patterns while waiting for the next target to appear. Consistent with prior work, we did not observe a significant correlation between sequence learning (i.e., rate of change on pattern sequences relative to random ones) and explicit awareness, indicating that learning was implicit. The lack of association between CRT performance and implicit learning on the SRTT may further suggest that differences in IL-pat were unlikely to be driven by a bias towards effortful top-down thinking.

The observed association between IL-pat and change in belief from childhood to adulthood can be taken as further indication against the likelihood of religious belief exerting a top-down influence on IL-pat. Change in belief across the lifespan, and particularly from childhood to adulthood, is not uncommon[41,88].

Thus, to the extent that individual differences in implicit learning are present at early ages[60,62,63], and may be stable across time[64–66], it appears more likely that differences in IL-pat could drive changes in belief across development than that changes in belief, or adult level of belief in an interventionist God as measured in the present study, are primary drivers of differences in IL-pat. Evidence of genetic influences on IL-pat[67,68], further suggests sources of individual differences in IL-pat that may be present prior to, and largely independent of, top-down influences.

While the bottom-up directionality of effect we have outlined is consistent with prior work on the neurocognitive bases of religious belief[1,2,9–12,16], it is not possible—and is not the intent of the present study—to conclusively rule out any degree of top-down influence on the relationship between IL-pat and religious belief. Indeed, a complete rendering of the cognitive and environmental influences on a phenotype as complex as religious belief very likely includes a multiplex of interconnected loop architectures between bottom-up and top-down processes. Examples of top-down influences on automatic perceptual information processing are abundant[19,23,98]. One pertinent example is research showing that induced meaning threats devised to challenge participants' existential sense of meaning were associated with better implicit learning in an artificial grammar task[99]. Religious belief is associated with a desire for meaning[4,11,46], and it is conceivable that more religious participants might tonically feel greater threat to meaning, which may have an influence on at least some forms of IL-pat. It should be noted that the improvement in grammar learning in this prior work was linked to an increased explicit motivation to find grammatical letter strings in the grammar learning task, whereas extant evidence indicates such explicit motivation does not improve learning on the SRTT[61,69,79]. More directly to the point, even if it is the case that some amount of top-down influence has bearing on implicit learning, this would not necessarily be inconsistent with bottom-up influences of implicit learning on religious belief. While the present study focused on individual differences in a strongly bottom-up IL-pat paradigm to explore the thus far untested relationship of bottom-up IL-pat to belief, the intent of the present research is not to suggest that all bottom-up influences on belief, and perhaps not even the IL-pat studied here, operate entirely independently of top-down processes.

Regarding the specific directional pathway represented by our mediation models, this directionality is based on a hypothesized pathway of effect within which there are a priori reasons to position intuitions of universal order prior to interventionist belief (i.e., as the mediator and dependent variable, respectively). Specifically, the directionality of the pathway is based on the a priori consideration that implicit learning of patterns/order in environmental stimuli is less likely to directly yield specific beliefs about deities, and more likely to yield a general sense/intuition that there is ambient order. Broad intuitions about the presence of order do not depend on externally learned narratives about the identities and powers of deities, and thus appear more likely to arise intrinsically (i.e., within the individual, directly from that individual's bottom-up implicit learning of patterns/order). Relatedly, intuitions of order appear more conceptually proximate to the content being learned via implicit learning of order than to the content of beliefs about deistic intervention (i.e., both the intuitions and the implicitly learned content directly concern order itself, rather than explanations of order). Thus, a progression from IL-pat to UO appears more likely than a direct leap from implicit learning to religious narratives about interventionist deities. As noted above, the hypothesized pathway of effect is also based on prominent extant accounts, developed to interpret decades of research on implicit learning, which indicate that IL-pat gives rise to intuitions of order[17,20–25]. Thus, in view of

theoretical considerations and extant literature, we hypothesized a pathway of effect whereby implicit learning influences UO and UO influences interventionist belief (i.e., implicit learning yields broad intuitions of order that predispose individuals toward belief narratives that fit these intuitions).

It is important to emphasize that the mediation analyses we conducted to test models of the hypothesized pathway should not be taken as establishing causation. Mediation is fundamentally a correlation-based technique, and is frequently over-interpreted to make causal claims[100–102]. Establishing causation requires a comprehensive research program that should ideally include longitudinal and intervention-based paradigms, a careful accounting of the influence of a host of measured and unmeasured mediating variables, and replication across varied experimental paradigms[100]. While this is beyond the scope of a single study, the pathway of effect modeled in the present study provides a framework for more comprehensive investigations of causation in the observed relationships, and the present data on belief change from childhood to adulthood suggest that longitudinal study might be particularly informative.

The present research also explored the extent to which IL-pat is related to individual differences in analytic problem-solving (measured by the CRT). An influence of analytic thinking on IL-pat might be posited whereby analytic thinking is associated with increased top-down searching for patterns, which could hinder IL-pat on the SRTT in nonreligious participants, given previous associations of analytic problem-solving on the CRT with lower levels of religious belief[41,42]. However, the lack of association between CRT and IL-pat suggests no such influence in the present study, perhaps owing in part to the attributes of the SRTT paradigm devised to minimize top-down influence (described above). Our findings further indicated that the associations of IL-pat to IB, belief change, and UO were independent of CRT performance. To our knowledge, this is the first research to empirically test whether performance on the CRT is associated with implicit pattern learning. Prior research that has suggested an association of implicit learning to thinking style has assessed intuitive thinking via self-report[55–57]. For example, Woolhouse and Bayne measured intuition using the self-report sensing-intuition scale of the Myers-Briggs Type Indicator. These previous approaches are more similar to the self-report measure of universal order intuitions in the present study than to the CRT. In contrast, assessment of thinking style in the context of arithmetic calculation on the CRT is most likely to be related to differences in so-called system 1 (automatic) and system 2 (deliberate) problem-solving[17], which appears conceptually distinct from the presence of broad intuitions of order that we and others[20–23,54] have suggested may arise from IL-pat, and that the present work investigated as a potential mediator of the effect of IL-pat on belief. Arithmetic calculation is also likely to reflect common math/calculation-specific influences on elements of thinking style related to motivation, avoidance and cognitive function[103–105].

While UO and IL-pat were consistently associated with IB across the various samples we studied, UO and IL-pat were not found to be associated with scores on the Belief in Science Scale[94]. Thus, it is likely that intuitions of ambient order more closely relate to IB than to belief in science, at least within the (certainly non-exhaustive) scope of what is measured by the presently employed survey of UO and Belief in Science Scale. More generally, the present findings do not rule out the possibility that nonreligious order-related beliefs, scientific or otherwise, could be related to UO or IL-pat. For example, because at least one item on the Belief in Science Scale directly contrasts belief in science with religious belief, it is possible that the present study underestimated the extent to which religion and science might operate concurrently as ordering beliefs in some individuals. However, even if the relationships of IL-pat and UO to

IB were found to operate in parallel to other relationships involving other ordering beliefs, this would not necessarily diminish the relationships to IB.

We did not observe an association between IL-pat and a single-item measure of belief in God's existence broadly (i.e., Existence Belief). This finding provides initial indication that the putative influence of IL-pat may be specific to interventionist belief (i.e., the hypothesized effect of learning orders implicitly on belief in an ordering God). Exploratory data analysis indicated that IL-pat was more strongly related to Existence Belief in a subset of individuals for whom belief that God exists is largely focused and/or dependent on belief in the interventionist influence of God, and less related to Existence Belief in others for whom belief in the existence of God depends more on other factors.

We also did not observe an association between explicit judgments about sequence blocks of the SRTT and religious belief. We are not aware of any research that has described an association between accurate, conscious detection of patterns and religious belief, though some prior research has reported an increased tendency among believers in the paranormal to report some forms of illusory contingency[106,107]. Erroneous reporting of patterns on the random blocks, as well as overall tendency to report patterns across both block types (random and pattern), was not correlated with IB or belief change in the present study. When these variables were included as additional covariate regressors in the above-reported models, the effects of IL-pat on both IB and belief change were not altered. Because IL-pat was also uncorrelated with explicit accuracy in reporting patterns, the results consistently supported the interpretation that implicit (rather than explicit) learning was driving the observed effects.

It is not evident, however, that the present results bear substantively on prior findings of explicit contingency reporting. Prior research investigating reporting of illusory contingency among believers in the supernatural was based on tasks and theoretical models that are quite distinct from those in the present study. In particular, prior studies investigated contingency in the form of causal influence of a participant's actions on observed outcomes (i.e., the agency of the participant in a causal context). This kind of contingency differs from the implicit pattern learning we studied because (1) our paradigm was not related to the participant's agency (participants learned pattern sequences that proceeded without any apparent opportunity for the participant to influence them, and no instruction was given regarding agency), and (2) the implicit learning sequences in the present study were not devised to indicate any form of causality, nor were participants given any instruction to consider causality as in prior studies. In addition, many aspects of the experimental designs, instructions, and stimulus modalities in these studies are quite different from those employed in the present study. For example, the contingency task employed by Blanco et al. involved a series of judgments about whether or not to act by administering medicine to patients, and illusions of causality were assessed based on the extent to which a participant incorrectly believed their administration of a non-contingent medicine was an effective treatment. Thus, illusory contingency was based on the participants' behavior and decision-making (and their perception of their own agency), rather than how they perceive/respond to implicitly learned stimuli. In addition, the authors related performance on their contingency task to superstitious and paranormal beliefs, which is a distinct outcome from the IB measures used in the present study.

As cognitive and neural inquiries into religious phenotypes advance, it is unlikely that any single large-scale explanatory factor will fully capture the complexity and diversity of influences that shape belief[1,2,4,10,12,13,15,16]. However, more modest effects that replicate across diverse groups of believers have the potential to

provide meaningful componential insights into fundamental neurocognitive bases of belief. Evidence linking religious belief to IL-pat suggests that belief, and variation in belief, may be embedded in fundamental bottom-up perceptual processing more deeply than has previously been empirically explored. The observed association of IL-pat with change in belief from childhood to adulthood (distinct from the powerful influence of parental belief) suggests that the emergence of individual differences in IL-pat in early childhood[69] might set individuals on bottom-up trajectories whereby literally perceiving the world differently (i.e., differential processing of patterned visual information available in the environment) helps to move them either towards or away from religious belief.

The primarily subcortical information processing pathways that instantiate IL-pat[69,70] presumably operate in all healthy human brains. Thus, if variations in belief are partially shaped by the presence and individual variability of IL-pat, this effect may be shared across diverse religious and cultural contexts. The present research is consistent with this hypothesis, identifying shared associations between IL-pat and belief in Western and Middle Eastern contexts characterized by marked differences in religious heterogeneity[95], societal tolerance of religious diversity[7], separation of religion and government[97], and religious restrictions[96]. Optimistically, evidence of a shared link to a core component of human perceptual information processing might help to emphasize an underlying human commonality between believers and non-believers, and between disparate religious groups.

## Methods

**Participants and design.** The U.S. sample comprised 199 participants recruited from Georgetown University and the local community. A subset of the U.S. sample (U.S. re-contact; $N = 65$) completed additional measures (described below) to assess analytic problem-solving as well as belief in science. Afghan participants ($N = 148$) were recruited from primarily ethnic Hazara sections of Kabul (e.g., Karte Seh, Dast-e Barchi) following study practices for culturally sensitive topics in Afghanistan developed by Z.W., who has extensive data-gathering experience in Afghanistan, including as director of the Asia Foundation's Survey of the Afghan People[108], and tested in a private secure space by same-sex Afghan experimenters (trained by Z.W.). Experimenters also completed online ethics training required by the Georgetown University IRB. All forms, including the consent form, were translated to Dari (one of two national languages, native for Hazaras) and then back-translated to English. Extensive quality control efforts for Afghan data collection are described in Supplementary Methods. Finally, we recruited an additional predominantly European online sample ($N = 96$; 83.33% European, Supplementary Table 2) to complete measures related to interventionist belief, universal order intuitions, analytic problem-solving, and belief in science.

All procedures were approved by the Georgetown University IRB. Samples were sized sufficiently to detect small-to-moderate effects at $\alpha = 0.05$ as there were not prior studies of IL-pat-belief associations to indicate anticipated effect sizes, and because religious belief is a complexly influenced phenotype such that any individual influence is unlikely to account for a large proportion of the variance.

**Measures.** U.S. and Afghanistan participants completed a battery of tasks in a single session lasting ~1.5 h. The SRTT was presented in E-Prime (Version 2.0). All other task elements were administered using Qualtrics (https://www.qualtrics.com/). Descriptions of measures not used in the present study can be found in Supplementary Methods.

Participants completed a modified SRTT, a widely used measure considered to reflect ecologically valid IL-pat[21]. Participants were instructed to quickly and accurately indicate the position of the target circle as it appeared at each of four positions arranged horizontally onscreen (i.e., left, center-left, center-right, right; Fig. 1). Each target position was designated a corresponding key on the keyboard, and participants were instructed on which key was associated with each target positions prior to beginning the task (moving from left to right, the mappings were "z", "x", ".", "/"). The version of the SRTT employed in the present study consisted of six blocks (3 pattern blocks; 3 random blocks). Each pattern block was composed of a distinct 10-target repeating sequence (repeated five times), consisting of three first-order structures (Fig. 1b, orange bars) and two second-order triplets (Fig. 1b, green circles). Random blocks did not include regular repetitions. For each block, we calculated the correlation between target number (1–50) and response time. Mean rates of change (average $r$ values) were calculated for each participant for pattern and random blocks. Consistent with recommended procedures[77], IL-pat was operationalized as the difference between random vs. pattern blocks to distinguish faster responding that is due to learning from faster responding due to confounding influences (e.g., motivation, familiarity). To assess explicit awareness,

following each block, participants were asked to give their best guess as to whether the block was or was not a pattern. Participants could respond from 1 (Definitely Not a Pattern) to 4 (Definitely a Pattern). We calculated an overall accuracy score, with a high score of 1 (responses of Definitely Not a Pattern for all random blocks and definitely a pattern for all pattern blocks) and low of 4 (responses of Definitely a Pattern for all random blocks and Definitely Not a Pattern for all pattern blocks). Illusory detection was calculated as the average score on random blocks only, ranging from 4 (low score: Definitely a Pattern for all) to 1 (high score: Definitely Not a Pattern for all). An overall bias towards reporting patterns regardless of block type was measured by subtracting accuracy for pattern blocks alone (high score = 1, low score = 4) from illusory detection, such that higher scores (max score = 3) reflected more frequent reporting of patterns.

The extent to which individuals gain conscious access to implicitly learned information has been a subject of considerable investigation[21,69,77,81]. A number of steps were taken during the development of the task used in the present study to isolate IL-pat and minimize the potential for explicit awareness. First, the blocks were short in duration, as research has consistently demonstrated that participants can gain some level of awareness with extended exposure to pattern SRTT sequences[69]. In addition, the opportunity to explicitly recall/rehearse target sequences in an effort to explicitly identify patterns for patterns was mitigated by removing the response-stimulus interval (RSI) between key-press and the appearance of the next target[81]. During no-RSI versions of the SRTT, the next target appears immediately after the correct response is made; participants cannot use time between target appearances (because the next target appears immediately) to consciously attempt to identify a pattern structure.

Although participants were not explicitly told to search for any patterns, nor were they told that any of the blocks would contain patterns, it is reasonable to assume that questions about the presence of patterns may have caused some participants to look for embedded patterns (note that this is quite distinct from providing participants with any advance knowledge of the particular pattern sequences to be presented). Of course, it is also possible that participants would try to find patterns in the sequences even without any questions related to patterns, and there are likely to be individual differences in the tendency to seek patterns. Of relevance to the present study, one possibility is that more religious people might be more likely to search for patterns, e.g., because this might align with an explicit, top-down belief that a God orders the universe. However, prior work has demonstrated that this explicit searching does not improve implicit learning, and may actually be deleterious[61,69,79]. It is therefore unlikely that a top-down tendency to explicitly seek patterns would manifest in superior IL-pat in the employed paradigm. In addition, within the present study, a measure of analytic problem-solving (i.e., one's tendency to engage a top-down, effortful approach to solving problems; CRT; see below) was unrelated to implicit learning on the SRTT.

Next, participants completed a number of measures to assess beliefs about Gods. The Belief in Divine Intervention Scale (BDIS[3]; Supplementary Table 3) surveys participants' level of agreement (six-point scale) with statements concerning God intervening in the world and human affairs. Cronbach's alpha was similarly high in both samples (U.S. = 0.88; Afghanistan = 0.77), indicating that the translated version of the measure was suitable for use in the Afghan sample.

Two overlapping circles (Supplementary Fig. 1) tasks—modified from prior work[83]—were administered to measure beliefs about God's influence on (1) their own actions (i.e., self-overlap), and (2) events in the world (world overlap). Participants arranged two circles onscreen such that the extent of overlap indicated their own representation of the extent of God's influence. More overlap was interpreted as stronger belief in an intervening God.

Belief change was surveyed using a lab-developed change of belief measure, which consisted of 9-point Likert scales on which participants reported their own strength of belief in God starting at age 6 and then at 3-year intervals up to age 24. Instructions stated that participants should not respond to items beyond their present age. Asking participants to report belief at 3-year intervals was intended to encourage more precise and thorough consideration of responses than might have been achieved by a single query (e.g., how much has your belief changed since childhood?). This method to assess change in religious belief is consistent with prior work that has demonstrated the efficacy of assessing religiosity retrospectively[41,88]. The primary outcome variable for belief change was the difference between reported strength of belief at the youngest surveyed age (6 years) vs. belief at the time of the study. Further characterization of this variable and its calculation can be found in the Supplementary Methods. U.S. participants also completed a single-item Existence Belief measure for which they rated the strength of their belief in the existence of God broadly using a 9-point Likert scale (1 = do not believe that God is real at all, 9 = absolutely certain God is real).

Participants also completed measures to assess intuitions of order in the universe. We initially created a 4-item survey (Supplementary Table 4) in which participants were presented with statements concerning the presence of order in the universe (without reference to religion or God) and asked to indicate their level of agreement (1–9), consistent with demonstrated means of explicitly measuring intuition[89]. Two of the original four UO questions were ultimately retained for analysis. These were Q1—everything happens for a reason, and Q2—there is order to the universe, for which responses were summed to produce a universal order (UO) score. The two other UO questions, which were not retained, included elements that could be interpreted as implied references to God: Q3—there is a

plan that guides events in the universe, and Q4—something beyond physics plays a part in deciding what happens. Indeed, participant responses to these two items were more strongly correlated with IB (both $r \geq 0.62$) than were the two questions retained for analysis (both $r \leq 0.42$). The determination to exclude responses to Q3 and Q4 from analysis was made in order to more clearly distinguish UO from belief in an intervening God (although post hoc analyses showed that inclusion of all items did not meaningfully change the reported results; Supplementary Fig. 5). Note that, because Q1 and Q2 were presented before Q3 and Q4, any implied reference to a deity in Q3 or Q4 would not have influenced participants' interpretation of Q1 or Q2. The UO survey for the European replication sample consisted of only the first two items.

We anticipated that UO was unlikely to be meaningful for the Afghan sample because of a lack of culture- and language-specific interpretability of the UO prompts—much of the difficulty arose in the attempt to make the prompts secular (i.e., to use wording/framing that referred to an order in the universe without referring to a deity or any religious belief). These attempts ultimately led to prompts that were not interpretable in the intended ways, as determined by Z.W. in consultation with Dari-speaking experimenters. Therefore, scores on this measure were not analyzed in this sample.

Finally, participants completed a number of other measures to estimate potential confounding variables. U.S. participants completed the 37-question yes–no Schizotypal Questionnaire, with the primary focus on the 17-item unusual perceptual experiences subscale[109], to distinguish effects of IL-pat from belief in supernatural agency[84,86,87]. After translation and cultural adaptation, Afghan participants answered a 36-item version, with a 16-item perceptual experiences subscale. Cronbach's alpha for the subscale was high (U.S. = 0.78; Afghanistan = 0.83), reflecting appropriateness in both samples. To assess parents' strength of religious belief, participants used a 9-point Likert scale to rate the strength of their parents' religious belief during the participants' childhood. Additional measures included the CRT[90], which consists of three calculation-based questions devised to elicit prepotent—but incorrect—responses, referred to as intuitive. Overriding these prepotent responses in favor of more effortful calculation, which is more likely to yield the correct solutions, is considered analytic thinking (also referred to as reflection). The number of correct and intuitive responses are summed for each participant. The CRT was completed by the U.S. re-contact and online European samples. U.S. re-contact and European samples completed the Belief in Science Scale[94], which consists of 10 questions intended to measure attitudes toward science, such as its value, reliability, and ability to provide an understanding of the world and human culture.

**Statistical analyses**. In order to reduce the number of variables in subsequent analysis, we sought to integrate three estimates of the same measurable characteristic (belief in an intervening God) into a single DV for regression models. To determine whether this was appropriate, we first ran zero-order correlations with all three measures (BDIS, self-overlap, world overlap; Supplementary Tables 5–7). For all samples, all scores were strongly positively correlated (all $r > 0.54$). Next, we performed a principal component analysis (PCA; Supplementary Table 8) to examine the structure of the data. We elected to perform a PCA because we theorized that variance on the three interventionist belief measures would be primarily due to a single, large component, consistent with the assumption imposed by PCA that all factors are orthogonal. As expected, a single component was retained for all samples. The ratio of the first to second eigenvalue was >2.5 (U.S.: component 1 eigenvalue = 2.30, component 2 eigenvalue = 0.48; Afghanistan: component 1 eigenvalue = 2.42, component 2 eigenvalue = 0.50; European sample: component 1 eigenvalue = 2.30, component 2 eigenvalue = 0.41), suggesting that scores on the three measures were indeed related to a single component. The one component solution was further confirmed by parallel analysis (U.S.: component 1 simulated eigenvalue = 1.11, component 2 simulated eigenvalue = 1.01; Afghanistan: component 1 simulated eigenvalue = 1.17, component 2 simulated eigenvalue = 0.99; European sample: component 1 simulated eigenvalue = 1.16, component 2 simulated eigenvalue = 1.00) and Velicer's MAP (U.S.: VSS 1 maximum complexity of 0.93 with one factor; Afghanistan: VSS 1 maximum complexity of 0.94 with one factor; European sample: VSS 1 maximum complexity of 0.93 with one factor). We therefore created an Interventionist Belief (IB) principal component score for each participant using a least squares regression approach such that scores for each sample had a mean of 0 and a standard deviation of 1 (Supplementary Fig. 2). All statistical analyses were performed in STATA 15 and R (Version 1.1.4).

**Reporting summary**. Further information on research design is available in the Nature Research Reporting Summary linked to this article.

## Data availability

The study data are available on the OSF repository (https://osf.io/g5ywe/). A reporting summary for this Article is available as a Supplementary Information file.

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

## Acknowledgements

This research was supported by grants from the John Templeton Foundation to A.E.G. (ID 51971) and to A.E.G, A.B.W., F.M.M, and Z.J.W. (ID 61114), and a National Science Foundation Graduate Research Fellowship Program award to N.M.G. We thank Nicholas Gibson, Ian Lyons, and Chandan Vaidya for insightful discussion and commentary, and Emily Dyke and Griffin Colaizzi for data collection efforts. Thanks also to the Afghan Cultural House and the Afghan translation team, as well as to Basir Bita for remote coordination of data collection in Afghanistan.

## Author contributions

A.E.G. conceived the experiment. F.M.M. and A.E.G. supervised the project. Z.J.W. oversaw all data collection in Afghanistan. A.E.G., A.B.W., N.M.G., and G.A.E. designed the study tasks. A.B.W. and N.M.G. collected and analyzed the data, including quality control of Afghanistan data (with guidance from Z.J.W.). A.B.W., N.M.G., and A.E.G. wrote the paper.

## Competing interests

The authors declare no competing interests.
