## [Peer Review File · Nature Communications]

Reviewers' comments:

Reviewer #1 (Remarks to the Author):

Thanks for this interesting paper: you are of course right that more cross-cultural work needs to be done, especially on claims about the relationship between "basic" cognition and cultural phenomena like religion. Here are some notes. At the end of the notes, I will have a little more to say about the logic of the paper and what it tells us. Spoiler: I'm recommending that this paper be accepted for publication, with revisions that speak to (a) theory, and (b) alternative models.

Introduction:

1. "Deistic belief" is an odd phrase, in part because "deist" has a well-established referent in religious studies: deism is the belief in a distant noninterventionist god. This is in direct contradiction to the kind of god that might be detected acting in our immediate environment.
2. Famously, the Gervais & Norenzayan study has failed to replicate in two independent attempts (Sanchez et al., 2015; Camerer et al. 2018). The correlational result has proven more robust. Gervais has continued to contribute to this literature, as has Gordon Pennycook.
3. Aaron Kay's experiment doesn't replicate either: he himself contributed to the replication study (<https://psyarxiv.com/vqu2x/>).
4. Re: the paucity of Middle Eastern Muslim data, you are of course right: but there is more Turkish Muslim data that almost anything else besides White US/European Christian data vis-a-vis religiosity.
5. The relationships between implicit learning tendency, implicit thinking style, and intuitions needs to be made a bit more explicit. The first two are clearly causes of the third. But what is the theoretical relationship between implicit learning style (which you go on to measure) and the sort of thing measured by CRT? You cite the Wollhouse and Bayne as showing that implicit learning predicts use of intuition in cognitive tasks, but if I'm not mistaken what that paper actually shows is that self-reported intuitiveness (on the MBTI) predicted performance on an implicit learning task. This is not quite the same thing, of course.

Method & Results:

1. The PCA is fine, but it would be good to know what Horn's Parallel Analysis and Velicer's MAP suggests too.
2. I don't see this in the text or SM, but I imagine you reverse scored two items in the BIDS?
3. Why are beliefs about universal order causally prior to interventionist beliefs? Given that both are, ex hypothesi, driven by intuition, it seems up for empirical grabs which, if either, comes before the other in the causal chain. Therefore, it seems best to also test other mediation models, rearranging the mediator and DV especially. This then allows you to compare between models.

Data availability:

I urge you to make your data open access. I understand the desire to embargo data until such a time as you are "done" with it, but perhaps the threat of competition will spur you to complete the other analyses and papers you plan for this dataset. The merit of making your data open access is enhanced by

the admitted messiness of your raw data. You have made various choices about exclusion, and readers have no real way of knowing how much difference those choices make unless they have access to the data (or if you run robustness checks). Messy raw data should not be penalised, of course: especially when it comes to cross-cultural work in difficult circumstances.

General comments:

The most interesting thing about your paper is the purported IV, implicit learning as an individual difference variable. I say "purported" because it is not as obvious to me that religiosity cannot shape implicit learning tendencies. Certainly, religious upbringing might shape implicit learning. Now, fortunately, you have as one of your DVs a measure of religious *change*, which--I agree--is a much less plausible causal factor for implicit learning. The case for implicit learning being the IV needs to be made more strongly, I think, and what I have just said might provide a start.

As mentioned above, there is a theoretical question to be asked about the relationship between implicit learning as measured by your task and implicit cognitive style as measured by, for example, the Cognitive Reflection Test. If they are the same thing, then this paper does not really report a novel result. So you are motivated to argue that they are different. But this is a difficult argument to make because you cannot do so on empirical groups unless you run another study that looks at the correlation between implicit learning and cognitive style. As far as I know, there has not been a study like this yet. You cite the Woolhouse and Bayne study, but that one relies on a self-report measure of cognitive style: perhaps that is good enough, but I am disinclined to think so. There is nothing really wrong with self-report measures, but cognitive style is a hybrid of attitude and ability, and self-report measures of ability are questionable indeed. In any case, the argument needs to be made that implicit learning is especially interesting to look at independent of cognitive style.

Finally, it is not clear what the theoretical motivation behind the mediation model is. Is there a debate about the relationship between "order" beliefs (others might say "teleological beliefs"; see for example work by Deb Kelemenn, Konika Banerjee, Beth Heywood) and religious beliefs that you are trying to resolve? One can, of course, test whatever model we fancy, but the reader should be led to believe that there is some kind of theoretical payoff. Otherwise, it looks like you have just put together a plausible model to pad the paper. That's a look to avoid.

Reviewer #2 (Remarks to the Author):

Thanks for the opportunity to review this interesting paper. The authors report that implicit learning of patterns in a computerized task predicts stronger belief in an intervening god and increased belief from childhood to adulthood. Impressively, the authors replicate this finding in two religiously diverse cultural contexts, the US and Afghanistan. They also adduce several methodological considerations in support of a causal story: implicit awareness of patterns in the environment predisposes people to believe in an intervening god (rather than the reverse). I found their arguments in this respect quite persuasive (though I was less convinced by their mediation analysis in the US sample).

Overall, though I like the study and think the result is interesting and potentially important, I have two main concerns which bear on the editorial decision. I also have some more minor recommendations for improving the paper.

Major issues

- The first issue is methodological. To my mind, in 2019, a psychology paper in a top outlet like Nature Communications should be pre-registered, and the data should be available to reviewers at the point of submission (rather than being “available from the corresponding author upon request”). Neither is the case here. The Reporting Summary and Supplementary Information indicate that a large number of participants were excluded (about 42% of the tested sample, across both cohorts), for all sorts of potentially arbitrary reasons (e.g., taking less than 25 minutes to complete the survey). In addition, the study protocol included administration of a wide range of measures that are not reported here (including measures of teleological bias, magical ideation and locus of control). These considerations raise the spectre of undue analytic flexibility and selective reporting. To cite just one example: To measure belief change, the authors had participants report their strength of belief in God from age 6 to their age at the time of the study, in 3-year intervals. But the score they computed only makes use of the first and last of these ratings. Why bother collecting the other intervals if the authors knew in advance that this was how they would compute this belief-change index?
- The second issue is conceptual. The take-home message of the paper is that individuals who are good at implicitly detecting patterns in the environment may intuit that the world is orderly, and thus are more likely to believe in an interventionist god. The obvious gap in this reasoning is that an interventionist god is not the only – and arguably not the best – explanation for apparent order in the universe. The authors say (lines 78-79) that “Believers are more inclined to perceive events in the world as adhering to a purpose or design rather than as a series of random, unpredictable occurrences”, but ever since Darwin it has been easy to disavow cosmic “purpose or design” without having to argue that the world is “a series of random, unpredictable occurrences”. Indeed, no-one with a scientific worldview will accept that the world is random and unpredictable, and it seems likely to me that people who are good at implicitly detecting patterns in the environment may also be more likely to adopt a scientific worldview. The authors’ data are unable to speak to this possibility. By analogy, it could be that those with greater implicit appreciation of rhythmic structure are more likely to enjoy death metal than those with a poorer rhythmic sensibility – but of course they would probably also be more likely to enjoy Mozart or Stravinsky. A headline about death metal would be quite misleading. In the same way, the finding that religious believers are better at apprehending environmental patterns than non-believers is potentially misleading, as scientists may also be better at apprehending environmental patterns than non-scientists.

Other issues

- In terms of revising the paper, one main point is that the nature of the perceptual task (the SRTT) was not clear (to me at least) from the authors’ description. What is it that participants actually see on a particular trial? On a first reading, I assumed that (say) an x in a blue circle would appear in one of four quadrants; but on closer inspection the x refers to the response key used to indicate one of the four

locations. So, what do the targets actually look like? It would be worth making this central component of the method as clear as possible.

- Some of the literature review is a bit out of date. In particular, there is uncritical acceptance of certain concepts and findings from the literature that have been queried or that have failed to replicate. For example, the authors describe (lines 61-63) how an “automatic bias to detect agency in the environment is thought to yield over-attribution of intentional agency at the intuitive level... which supports explicit beliefs about supernatural, intelligent agents”, yet there is very little empirical evidence that this is the case. They cite van Elk’s (2013) study here, but van Elk found that religious beliefs were not related to agency attribution. Later work by van Elk’s group suggests, if anything, that humans under-attribute intentional agency (Maij et al., 2019). Another example: on lines 71-72 they say “Gervais and Norenzayan demonstrated that promoting analytic thinking (rather than intuitive thinking) reduced strength of religious belief”. Multiple studies have failed to replicate this finding, and Will and Ara have themselves backed away from it (see Gervais, 2017; Gervais & Norenzayan, 2018).
- Lines 216-8: The authors note that explicit accuracy was not related to IB or Belief Change, but what about explicit belief that there was a pattern? Are people who are more likely to explicitly identify patterns (whether or not they’re accurate about that) more likely to be believers? There’s an emerging literature on how illusions of contingencies predict supernatural/paranormal belief (e.g., see Blanco, Barberia & Matute, 2015; Griffiths, Shehabi, Murphy & Le Pelley, 2018).
- Lines 434-6: The authors state that “because Q1 and Q2 were presented before Q3 and Q4, any implied reference to a deity in Q3 or Q4 would not have influenced participants’ interpretation of Q1 or Q2.” But that presumably depends on how the items were presented – was it possible for participants to look ahead to subsequent items?
- Lines 442-5: “Afghan participants showed lower variance... and stronger correlations between UO items... suggesting that they did not detect the subtle differences between items.” But what was the correlation between UO and IB for the Afghan participants? Isn’t that the critical feature here?
- There are a number of typos in the manuscript. Here are some: “Judiasm” (line 81), “evalute” (line 129), “across religious.” (line 129), “substancially” (line 131), “individual difference(s)” (line 150); “gods the U.S.” (line 256) “religious individual(s)” (line 278); “Definitely a Not a Pattern” (line 379); “extend(ed) exposure” (line 385); “lead” should be “led” (line 440); “measureable” (line 456) etc. etc.

References

- Blanco, F., Barberia, I., & Matute, H. (2015). Individuals who believe in the paranormal expose themselves to biased information and develop more causal illusions than nonbelievers in the laboratory. *PloS One*, 10(7), e0131378. <https://doi.org/10.1371/journal.pone.0131378>
- Gervais, W. M. (2017). Post publication peer review. <http://willgervais.com/blog/2017/3/2/post-publication-peer-review>.
- Gervais, W. M. & Norenzayan, A. (2018). Analytic atheism revisited. *Nature Human Behaviour*, 2, 609.
- Griffiths, O., Shehabi, N., Murphy, R. A., & Le Pelley, M. E. (2018). Superstition predicts perception of

illusory control. *British Journal of Psychology*, 110(3), 499–518. <https://doi.org/10.1111/bjop.12344>

- Maij, D. L. R., van Schie, H. T. & van Elk, M. (2019). The boundary conditions of the hypersensitive agency detection device: An empirical investigation of agency detection in threatening situations. *Religion, Brain & Behavior*, 9(1), 23-51. doi:10.1080/2153599X.2017.1362662

Reviewer #3 (Remarks to the Author):

The authors investigate if individual differences in implicit pattern learning predicts both belief in an intervening god, as well as changes in theological beliefs over time. The authors use a Serial Reaction Time Task (SRTT) to represent implicit pattern learning, a composite self-report measure of belief in an ordering and intervening god, and a retrospective measure of belief change since childhood. They (a) show support for both hypothesis, in samples from the U.S. and Afghanistan; as well as, (b) beliefs in a (non-theologically) ordered universe mediates the observed relationship within the US sample.

This is an interesting and well-executed contribution to our understanding of the cognitive underpinnings of religious belief. As the authors mention, there is much work on the role of intuitive information processing as a driving force behind the acquisition, maintenance, and rejection of religious belief. Two aspects of this work jump out as clear contributions:

First, much of the previous work has used simple measures of “intuition” – e.g. the Cognitive Reflection Test (CRT) or variations of heuristics and biases batteries. The approach outlined in this paper is operating at a much lower level and is less subject to deflationary counter explanations.

Second, as were much work is purported to be about “intuition”, the vast majority report negative relationships between belief and “reflection”, e.g. higher scores on the CRT (with one exception: Shenhav, Rand, & Greene, 2012). This work, however, does a nice job of selecting a task that explicitly focuses on intuitive information processing as the mechanism driving the relationship (though see point 1 below).

Overall, I like this paper and can see suggesting its acceptance upon the completion of several changes and clarifications.

More substantive thoughts/ concerns:

1. This paper frames the argument as- evidence shows religious beliefs are underpinned by intuitive factors and thus, variation in intuition-based information processing can be thought of as possibly driving variation in religious belief. However, some work in this area argues it’s reflection that’s driving this relationship (Pennycook, et al, 2012). This work notes how a non-trivial amount of the Cognitive Science of Religion has cited violations of intuitions to be an important component of many, if not most,

religious content such as narrative (Boyer, 2007). Reflection has been described as the process of overriding intuitive responses, intuitive responses that all individuals experience, but only a subset override. Thus, it's hard to know if variation in individual intuition, vs variation in individual reflection, is driving this relationship with religious belief.

Noting my point above, much of the previous literature reports variation in "reflection" and not "intuition" as driving the relationship with belief. In the paper, the authors note how attempts at identifying patterns on the SRTT can result in lower scores. One could imagine a case where there is little variation in "intuition" based performance; however, notable variation in attempts to detect a pattern, driven by individual difference in attempts to critically evaluate the information, driven by individual differences in "reflection". This may be a subtle point, but important for understanding the underlying mechanism. What would the authors say about this hypothetical case? Is there some way of understanding performance on the SRTT that would address this question?

2. Much of the previous literature has looked at overall belief in god as a main outcome of interest. This project's focus, looking specifically at an ordering/ intervening god, is reasonable given the way the authors frame it. That said, it seems they do in fact have a measure of current level of belief (how the "change" variable is created). It would be helpful to see how IL-pat scores correlate with this outcome, given the previous literature. If for whatever reason, this relationship is less robust, some explanation would be helpful for why. If this was the case, one possibility could be that a subset of the participants believes strongly in a god, but a god that is less causally relevant or present in their lives (e.g., some kind of deists). Though this would be surprising.

3. Though I understand mediation analysis is a common statistical practice in psychology, there are issues with using it to support causal claims (Green, Ha, & Bullock, 2010). So, I would just suggest a more nuanced tone when interpreting the mediation result. This is still inherently a correlational finding. That said, there could be more here to provide support for this conclusion. I'd be curious what this model looks like when you flip the MV and DV – so what is the overall relationship between IL-pat and OU with and without including interventionist beliefs as a mediator. If the original model is much stronger, that would help some here.

4. One would expect an effect of parental beliefs on the development of children's beliefs in an organized and ordered universe. Therefore, it seems a bit notable that this isn't included in the UO model – given the argument for including it in the IB model. Why is this? What does this relationship look like when you control for parental belief? It may be helpful to have a regression table in the supplement that orders these models and allows for side-by-side comparisons.

Smaller thoughts/ suggestions:

1. You cite Gervais & Norenzayan, 2012 as one of the main papers linking intuition with belief. However, it probably would be good/ transparent to note there are some failed replications for the most prominent manipulation in that paper (Sanchez, et al., 2017; Camerer, et al, 2018).

2. You note the zero-order correlations for each belief variable and IL-pat in the supplement, but it would be great to see this in the main text. The $p=.06$ is not concerning given the main result. I was curious/ suspicious when I got to that section that I didn't see any zero-order correlations; especially with you presenting the other zero orders after the main model, on line 216. It makes sense to control for parental beliefs to try and isolate the cognitive effect, so squashing the reader's concerns at the onset, and being upfront with the trending relationship (which is just an arbitrary cutoff) seems better to me (personally).

3. One curiosity I have is if there is any evidence that IL-pat can be heritable. My thoughts here are that the main mechanism being cited is biologically based and thus could easily be heritable. If this assumption is true, parents' IL-pat would not be fully independent from their children's – unless I'm missing something. I think it would be great to have one or two sentences somewhere in the discussion. I know that this is very minor given the zero-order I just noted in #2; however, something that helps an overly cautious reader realize that this is not a concern and help guide their understanding here would be welcomed.

Boyer, P. (2007). Religion explained: The evolutionary origins of religious thought. Basic books.

Camerer, C. F., Dreber, A., Holzmeister, F., Ho, T. H., Huber, J., Johannesson, M., ... & Altmeld, A. (2018). Evaluating the replicability of social science experiments in Nature and Science between 2010 and 2015. *Nature Human Behaviour*, 1

Frederick, S. (2005). Cognitive reflection and decision making. *Journal of Economic perspectives*, 19(4), 25-42.

Green, D. P., Ha, S. E., & Bullock, J. G. (2010). Enough already about "black box" experiments: Studying mediation is more difficult than most scholars suppose. *The Annals of the American Academy of Political and Social Science*, 628(1), 200-208.

Pennycook, G., Cheyne, J. A., Seli, P., Koehler, D. J., & Fugelsang, J. A. (2012). Analytic cognitive style predicts religious and paranormal belief. *Cognition*, 123(3), 335-346.

Sanchez, C., Sundermeier, B., Gray, K., & Calin-Jageman, R. J. (2017). Direct replication of Gervais & Norenzayan (2012): No evidence that analytic thinking decreases religious belief. *PloS one*, 12(2), e0172636.

Shenhav, A., Rand, D. G., & Greene, J. D. (2012). Divine intuition: Cognitive style influences belief in God. *Journal of Experimental Psychology: General*, 141(3), 423.

Reviewer #1

Remarks to Author

Thanks for this interesting paper: you are of course right that more cross-cultural work needs to be done, especially on claims about the relationship between "basic" cognition and cultural phenomena like religion. Here are some notes. At the end of the notes, I will have a little more to say about the logic of the paper and what it tells us. Spoiler: I'm recommending that this paper be accepted for publication, with revisions that speak to (a) theory, and (b) alternative models.

Response: We appreciate the reviewer's careful reading, insightful questions and suggestions, and encouraging general assessment. We have endeavored to address each question and suggestion below.

Point 1: "Deistic belief" is an odd phrase, in part because "deist" has a well-established referent in religious studies: deism is the belief in a distant noninterventionist god. This is in direct contradiction to the kind of god that might be detected acting in our immediate environment.

Response: We thank the reviewer for this feedback. We no longer use this terminology in the revised manuscript, and quite agree that this helps avoid potential confusion.

Points 2 – 3: Famously, the Gervais & Norenzayan study has failed to replicate in two independent attempts (Sanchez et al., 2015; Camerer et al. 2018). The correlational result has proven more robust. Gervais has continued to contribute to this literature, as has Gordon Pennycook. Aaron Kay's experiment doesn't replicate either: he himself contributed to the replication study (<https://psyarxiv.com/vqu2x/>).

Response: We thank the reviewer for these notes and have revised the introduction to indicate the failed replication of the analytic prime employed by Gervais and Norenzayan (Page 5):

Other literature has suggested a link between top-down, effortful analytic thinking – a propensity to critically examine and override prepotent “automatic”⁴¹ responding – and religious disbelief^{41,42}. Gervais and Norenzayan⁴³ reported that priming analytic thinking reduced strength of religious belief. However, replication attempts have not supported the efficacy of the so-called “analytic prime” employed by Gervais and Norenzayan^{44,45}.

The sentence that referred to compensatory control research (e.g., Kay et al., 2008), has been removed from the revised manuscript. We have retained descriptions of the more relevant work establishing a connection between interventionist belief and the perception

of events in the world as being ordered, and of the presence of such an association across religious denominations (Page 5):

A core belief across major religions is that the sequence and structure of events in human lives (and in the universe more broadly) reflects an underlying order determined by the intervention of gods^{3,4,46-50}. Believers are more inclined to perceive events in the world as adhering to a purpose or design rather than as a series of random, unpredictable occurrences^{47,49}. Indeed, such interventionist belief is an explicit focus of religious practices in Christianity, Islam, Hinduism, and Judaism⁵⁰. Empirical work has associated belief in gods with explanations of events as adhering to an intelligently designed order^{49,51,52}, and greater belief that a deity plays a role in ordering life events is posited to lead to increased strength of religious belief broadly^{50,53}.

Point 4: Re: the paucity of Middle Eastern Muslim data, you are of course right: but there is more Turkish Muslim data that almost anything else besides White US/European Christian data vis-a-vis religiosity.

Response: With thanks to the reviewer for pointing this out, we now note prior research on Turkish Muslims, and indicate more specifically that less westernized nations in the Middle East remain underrepresented in psychological research (Pages 7-8):

Middle Eastern Islamic samples, in particular, are largely absent from psychological research broadly. **Turkish Muslims are a relative exception⁷²⁻⁷⁴, but less westernized nations in the region are dramatically underrepresented**, such that sociocultural influences on cognitive and behavioral phenotypes (religious and otherwise) remain poorly understood^{75,76}. In addition to the inherent value of expanding psychological research in the Middle East, investigating the replicability of findings across U.S. and Middle Eastern samples provides the opportunity for strong tests of mechanisms hypothesized to operate across disparate socio-cultural contexts.

Point 5: The relationships between implicit learning tendency, implicit thinking style, and intuitions needs to be made a bit more explicit. The first two are clearly causes of the third. But what is the theoretical relationship between implicit learning style (which you go on to measure) and the sort of thing measured by CRT? You cite the Wollhouse and Bayne as showing that implicit learning predicts use of intuition in cognitive tasks, but if I'm not mistaken what that paper actually shows is that self-reported intuitiveness (on the MBTI) predicted performance on an implicit learning task. This is not quite the same thing, of course.

Response: We greatly appreciate this reviewer's questions regarding how implicit learning relates to reliance on intuitive-verses-analytic thinking as measured by the cognitive reflection test; CRT). Indeed, the literature we cited in the original version of

the manuscript (including the Woolhouse and Bayne study) operationalized intuition using self-report measures (e.g., the sensing-intuition scale of the Myers-Briggs Type Indicator). We agree with the reviewer that the sort of thing measured by the CRT is not addressed by measuring the sort of intuitions that we assessed via the UO measure, specifically intuitions of order. What the CRT measures appears to be quite different (i.e., whether participants engage effortful analytic/reflective approaches during calculation-based problem solving or whether they default to less effortful prepotent responding, which is referred to as “intuitive”). Given the presence of the CRT in extant research that has investigated belief-related constructs, we agree that it is useful to characterize the extent to which IL-pat and the relationships between IL-pat and other variables of interest in the present study might be related to, or independent from, performance on the CRT. Like the reviewer, we are not aware of any published work that has explored the extent to which performance on the CRT is related to performance on an implicit learning task (such as the SRTT).

In order to address this, all U.S. participants were re-contacted and asked to complete the CRT. Analysis of data collected from the 65 participants who responded to our re-contact request has now been added to the results section of the revised manuscript on Pages 16-18:

Cognitive Reflection Test

The above results supported hypothesized links of intuition to IL-pat and belief – these hypotheses specifically concerned intuitions of order (i.e., UO). Subsequently, we explored a secondary question concerning the Cognitive Reflection Test (CRT⁹⁰), a measure of calculation-based problem solving devised to distinguish responding based on a more effortful and analytic approach from responding based on a less effortful and analytic (and more error-prone) approach; the latter approach has been referred to as “intuitive” thinking. Investigation of the CRT was motivated by prior research that has suggested a connection between non-belief and analytic problem solving on the CRT^{42,44,45,91-93}. It was thus of interest to determine whether IL-pat was related to CRT performance, and whether the associations of IL-pat to belief and to intuitions of order (as measured by UO) were independent of the kind of analytic vs. “intuitive” problem solving measured by the CRT. All U.S. participants who completed the full study battery were re-contacted, and respondents (“U.S. re-contact” sample; N = 65) completed the CRT. The CRT consists of three arithmetic problems. For each problem, there is a response that is considered to be prepotent (i.e., the first answer that comes to mind), which is referred to as the “intuitive” response, whereas calculating the correct answer is posited to require greater analytic, top-down effort. Correct (0-3) and intuitive (0-3) responses are summed for each participant. Incorrect responses that are not the intuitive response are ignored in the calculation of correct and intuitive scores. Following Frederick (2005), the number of correct responses was interpreted as an estimate of analytic problem-solving, or the extent to which individuals are able to override prepotent (incorrect) responses.

Consistent with the associations observed in the full U.S. sample, IL-pat was significantly correlated with IB ($r = .46, P = 0.0001$), Belief Change ($r = .39, P = 0.002$), and UO ($r = .39, P = 0.001$; Supplementary Table 14). Also consistent with results obtained from the full sample, there was no association between IL-pat and accuracy of explicit pattern judgements ($r = -0.18, P = 0.16$), indicating that learning in this sub-sample was implicit. Results for the CRT indicated that analytic (i.e., correct) and intuitive (incorrect) responses were strongly negatively correlated with each other (both samples $r < -0.85, P < 0.0001$), as in prior research⁹⁰. In the analyses of greatest interest, we found that IL-pat was not associated with analytic responding ($r = -0.07, P = 0.56$) or intuitive responding ($r = -0.03, P = 0.84$; Supplementary Table 14). Linear regression models in which CRT performance was added as an additional covariate regressor (in addition to schizotypal thinking and parents' belief) indicated that IL-pat remained a significant predictor of IB ($b = 3.22, \beta = 0.43, SE = 0.81, P < 0.001$), Belief Change ($b = 9.88, \beta = 0.40, SE = 2.96, P = 0.001$), and UO ($b = 11.64, \beta = .38, SE = 3.61, P = 0.002$; Supplementary Table 16). Together these results indicate that the relationship of IL-pat to IB, Belief Change, and UO was meaningfully distinct from CRT-based problem solving approach. That is, it is unlikely that individual differences in the tendency to effortfully apply analytic thinking to override prepotent responding (at least in the context of CRT calculation problems) are responsible for variation in IL-pat or the association between IL-pat and belief. The CRT was also collected online in the European replication sample. Generally consistent with prior work associating analytic thinking more closely with disbelief than belief^{41,42}, associations were found between IB and CRT performance (analytic: $r = -0.22, P = 0.03$; intuitive: $r = 0.23, P = 0.02$; Supplementary Table 15). These relationships were not significant within the U.S. re-contact sample, though opposite directions/signs were again observed for the nominal associations of IB to analytic ($r = -0.10, P = 0.41$) and intuitive ($r = 0.14, P = 0.27$) responding (Supplementary Table 14).

Supplementary Table 16. Linear regression results in U.S. re-contact sample with analytic and intuitive thinking during the CRT as additional covariate regressor

	Analytical Thinking					Intuitive Thinking				
	β	B	SE	t	P	β	B	SE	t	P
Interventionist Belief										
IL-Pat	0.43	3.22	0.81	3.98	<.001	0.43	3.26	0.80	4.06	<.001
Schizotypy	0.01	0.06	0.54	0.12	0.906	0.02	0.09	0.53	0.16	0.872
Parents' Belief	0.32	0.13	0.04	2.96	0.004	0.31	0.13	0.04	2.83	0.006
CRT Performance	-0.03	-0.02	0.09	-0.24	0.813	0.10	0.09	0.10	0.90	0.373
Belief Change										
IL-Pat	0.40	9.88	2.96	3.34	0.001	0.39	9.67	2.95	3.28	0.002
Schizotypy	0.02	0.36	1.96	0.18	0.855	0.02	0.28	1.96	0.14	0.886
Parents' Belief	-0.09	-0.12	0.16	-0.74	0.464	-0.08	-0.11	0.16	-0.67	0.503
CRT Performance	0.08	0.22	0.34	0.63	0.531	-0.10	-0.31	0.37	-0.84	0.407
Universal Order										
IL-Pat	0.38	11.64	3.61	3.23	0.002	0.37	11.44	3.61	3.16	0.002
Schizotypy	0.12	2.53	2.34	1.06	0.292	0.12	2.51	2.40	1.05	0.299
Parents' Belief	0.16	0.27	0.20	1.37	0.177	0.15	0.25	0.20	1.27	0.209
CRT Performance	0.09	0.32	0.42	0.78	0.439	-0.02	-0.08	0.45	-0.17	0.868

Interpretation of these results has also been added in the Discussion on pages 24-25:

The present research also explored the extent to which IL-pat is related to individual differences in analytic problem solving (measured by the CRT). An influence of analytic thinking on IL-pat might be posited whereby analytic thinking is associated with increased top-down searching for patterns, which could hinder IL-pat on the SRTT in *nonreligious* participants, given previous associations of analytic problem-solving on the CRT with lower levels of religious belief^{41,42}. However, the lack of association between CRT and IL-pat suggests no such influence in the present study, perhaps owing in part to the attributes of the SRTT paradigm devised to minimize top-down influence (described above). Our findings further indicated that the associations of IL-pat to IB, Belief Change, and UO were independent of CRT performance. To our knowledge, this is the first research to empirically test whether performance on the CRT is associated with implicit pattern learning. Prior research that has suggested an association of implicit learning to thinking style has assessed intuitive thinking via self-report⁵⁵⁻⁵⁷. For example, Woolhouse and Bayne (2000) measured intuition using the self-report sensing-intuition scale of the Myers-Briggs Type Indicator. These previous approaches are more similar to the self-report measure of universal order intuitions in the present study than to the CRT. In contrast, assessment of thinking style in the context of arithmetic calculation on the CRT is most likely to be related to differences in so-called “system 1” (automatic) and “system 2” (deliberate) problem-solving¹⁷, which appears conceptually distinct from the presence of broad intuitions of order that we and others^{20-23,54} have suggested may arise from IL-pat, and that the present work investigated as a potential mediator of the effect of IL-pat on belief. Arithmetic calculation is also likely to reflect common

math/calculation-specific influences on elements of thinking style related to motivation, avoidance and cognitive function¹⁰³⁻¹⁰⁵.

In sum, we think these new data and analyses are helpful in assessing for the first time the relationship of CRT to implicit pattern learning, and specifically in determining that the kind of analytic vs. “intuitive” thinking measured by the CRT does not account for the main results we observed. As noted in the excerpted text, IL-pat remained a significant predictor of IB, Belief Change, and UO even when controlling for CRT performance.

Point 6: The PCA is fine, but it would be good to know what Horn's Parallel Analysis and Velicer's MAP suggests too.

Response: Thank you for this recommendation. As indicated in the below text (now included in Method section of the revised manuscript) results from Horn’s Parallel Analysis and Velicer’s MAP also suggest a single factor (Pages 34-35):

As expected, a single component was retained for all samples. The ratio of the first to second eigenvalue was greater than 2.5 (U.S.: Component1 Eigenvalue = 2.30, Component2 Eigenvalue = 0.48; Afghanistan: Component1 Eigenvalue = 2.42, Component2 Eigenvalue = 0.50; **European Sample: Component1 Eigenvalue = 2.30, Component2 Eigenvalue = 0.41**), suggesting that scores on the three measures were indeed related to a single component. **The one component solution was further confirmed by parallel analysis (U.S.: Component1 Simulated Eigenvalue = 1.11, Component2 Simulated Eigenvalue = 1.01; Afghanistan: Component1 Simulated Eigenvalue = 1.17, Component 2 Simulated Eigenvalue = .99; European Sample: Component 1 Simulated Eigenvalue = 1.16, Component2 Simulated Eigenvalue = 1.00) and Velicer’s MAP (U.S.: VSS 1 maximum complexity of .93 with one factor; Afghanistan: VSS 1 maximum complexity of .94 with one factor; European Sample: VSS 1 maximum complexity of .93 with one factor).**

Point 7: I don't see this in the text or SM, but I imagine you reverse scored two items in the BIDS?

Response: This is correct; items 3 and 5 are reverse scored. We have added a note to Supplementary Table 3 to indicate this.

Supplementary Table 3. Belief in Divine Intervention Scale.

"Please indicate your agreement or disagreement with the following statements, from Strongly Disagree to Strongly Agree"						
	-3 (Strongly disagree)	-2	-1	1	2	3 (Strongly agree)
1. God sometimes directly intervenes to heal individuals of diseases like cancer.						
2. God sometimes communicates directly with individuals.						
3. God does not intervene directly in our lives						
4. God sometimes directly intervenes to change the course of damaging weather conditions like hurricanes.						
5. Real miracles of healing from God do not occur today.						
6. God sometimes uses dreams to communicate with us.						

Note: Items 3 and 5 are reverse scored.

Point 8: Why are beliefs about universal order causally prior to interventionist beliefs? Given that both are, *ex hypothesi*, driven by intuition, it seems up for empirical grabs which, if either, comes before the other in the causal chain. Therefore, it seems best to also test other mediation models, rearranging the mediator and DV especially. This then allows you to compare between models.

Response: We appreciate this question. There are several directions we have taken in addressing it. Most directly related to reviewer’s suggestion, we have added substantially to the manuscript in order to more clearly motivate and specify the hypothesized directionality modeled in our mediation analyses. We also agree that it is useful to explore alternative models, and have now added a fairly extensive set of exploratory analyses to do so. We note that some of the text excerpted for this response is also excerpted in response to Point 10 below. With apologies for the repetition, we judged that some shared elements of the revised manuscript might be useful/informative in relation to both comments. Thus the text was included in both responses in the interest of ensuring the completeness of each response individually.

New text regarding the theoretical basis of the hypothesized pathway of effect (IL-pat to intuition to belief) is now included in the Results section on page 14:

Having identified relationships between IL-pat and explicit belief, we sought to further test the hypothesis that intuitions about order in the universe might mediate these relationships. Given prior work indicating that IL-pat contributes to intuitions of order^{17,20-23} and that intuitions contribute to religious belief^{1,2,10-14,16,32}, we hypothesized a pathway of effect in which IL-pat contributes to

intuitions of order that incline individuals toward religious belief narratives. **The hypothesized directionality of this pathway was also based on theoretical considerations, e.g., that intuitions of order can plausibly develop from implicit learning of order even without explicit learning of religious narratives, and that the content of what is learned via IL-pat (i.e., order itself) is conceptually closer to intuitions directly concerning order than to religious beliefs about interventionist deities (see Discussion).**

And a more expanded treatment is now presented in the Discussion on page 23:

Regarding the specific directional pathway represented by our mediation models, this directionality is based on a hypothesized pathway of effect within which there are *a priori* reasons to position intuitions of universal order prior to interventionist belief (i.e., as the mediator and dependent variable respectively). Specifically, the directionality of the pathway is based on the *a priori* consideration that implicit learning of patterns/order in environmental stimuli is less likely to directly yield specific beliefs about deities, and more likely to yield a general sense/intuition that there is ambient order. Broad intuitions about the presence of order do not depend on externally learned narratives about the identities and powers of deities, and thus appear more likely to arise intrinsically (i.e., within the individual, directly from that individual's bottom-up implicit learning of patterns/order). Relatedly, intuitions of order appear more conceptually proximate to the content being learned via implicit learning of order than to the content of beliefs about deistic intervention (i.e., both the intuitions and the implicitly learned content directly concern order itself, rather than explanations of order). Thus, a progression from IL-pat to UO appears more likely than a direct leap from implicit learning to religious narratives about interventionist deities. As noted above, the hypothesized pathway of effect is also based on prominent extant accounts, developed to interpret decades of research on implicit learning, which indicate that IL-pat gives rise to intuitions of order^{17,20-25}. Thus, in view of theoretical considerations and extant literature, we hypothesized a pathway of effect whereby implicit learning influences UO and UO influences interventionist belief (i.e., implicit learning yields broad intuitions of order that predispose individuals toward belief narratives that fit these intuitions).

This new text is in addition to text that was largely retained from the original manuscript that addresses motivations for the hypothesized directional pathway.

In the Introduction on pages 3-5:

Several theories of human cognition delineate trajectories of influence by which bottom-up processes direct individuals toward the formation of certain explicit beliefs^{17,20-25}. According to these accounts, intuitions (i.e., reportable experiences of knowledge that was not consciously learned) develop as products of bottom-up

gathering of environmental signals/information via unconscious information processing²⁰⁻²⁵. Because individuals are not aware of such bottom-up influences, intuitions drawn from unconscious processing may instead be consciously interpreted via explicit belief narratives that provide a rationalized context for beliefs and behaviors^{2,23,24}. Indeed, intuitions frequently bias more explicit top-down views and judgements^{2,13,17,26}, and certain explicit beliefs may be more compelling and difficult to override when they stem from intuitive impressions^{2,13,17,27}. These effects have been found to operate across diverse modalities of sensory information processing²⁸⁻³⁰. For example, in the context of interpersonal evaluations, humans rely on rapid, nonconscious face processing to form intuitions of trustworthiness²⁹⁻³¹, which has substantial influence on subsequent decision-making³¹.

With respect to religious belief, extant theoretical accounts^{1,2,9,10,13-16,32} posit that evolved neurocognitive processes contribute to “default intuitions” about the nature of environmental stimuli which direct individuals toward explicit beliefs. **For instance, neurobiology evolved for cooperative interactions among humans is implicated as a basis for intuitions concerning morality and fairness^{2,10,15}. These intuitions, in turn, are theorized to bias individuals toward religious beliefs in divine justice and watchful gods, which align with moral intuitions, and may insulate against challenges to intuitions about justice when commensurate retribution or compensation for a given act is not materially available³³. Other theoretical work has suggested a connection of evolutionarily-favored pathogen-avoidance mechanisms (e.g., evolved avoidance of individuals who are sick to minimize transmission of pathogens) to intuitions concerning cleanliness and religious beliefs concerning purity (e.g., belief in the healing effects of touching holy individuals or items, belief in unseen agents of spiritual corruption)^{16,34}. Intuitions of anthropomorphism, which appear to be biologically rooted in systems evolved to support face processing and social-cognition³⁵, may contribute to beliefs in watchful invisible agents^{36,37}. Relatedly, the automatic bias to detect agency in the environment is thought to yield over-attribution of intentional agency at the intuitive level (i.e., intuitions that non-agentic things have agency), which supports explicit beliefs about supernatural, intelligent agents (e.g., deities)^{1,13,14}. Empirical support for this theory, however, has been mixed; recent work has suggested that individuals may *under* attribute agency³⁸, and that believers in the paranormal – but not those who endorse a set of more traditional religious beliefs – display illusory agency detection^{39,40}.**

And later in the Introduction on pages 6-7:

In view of the associations between intuition and religious beliefs, it is notable that multiple lines of work in cognitive science have specifically implicated order-related perceptual information processing as a basis of intuitions^{21-23,54}. In particular, this work points to bottom-up learning of predictive order (patterns) in the environment without conscious awareness (i.e., implicit pattern learning; IL-pat)^{20-23,25,54} as an underlying influence. Reber²¹ identifies IL-

pat as a bottom-up perceptual basis of intuitions of order, including “tacit-knowledge” of complex visuospatial patterns after extended exposure. This work is situated within a broader framework developed by Epstein and others in which, “The implication of automatic implicit learning from experience is that the information acquired... is the primary source of intuitive ‘knowing without knowing how one knows’”²³, and in which such intuition serves as a bridge between implicit and explicit levels of cognition^{23,24,54}. Convergently, research into individual differences in IL-pat^{55–58} has demonstrated that IL-pat is positively associated with **self-reported measures of intuition across several paradigms**.

The work identifying bottom-up information processing and intuitions rooted in evolved neurocognitive systems as influences on belief suggests the hypothesis that processing information related to order in the environment via IL-pat could influence the formation of belief in an intervening/ordering god. Specifically, accounts implicating IL-pat as a basis of intuitions^{21–23,54}, and intuition as a basis of beliefs^{2,4,9,15,17,24}, indicate a potential pathway of effect whereby implicit learning of actual patterns/order in the environment could yield intuitions of ambient order that, in turn, influence belief in an ordering god. In other words, intuitions of order that result from IL-pat might disproportionately direct stronger implicit learners toward explicit beliefs in an intervening/ordering god.

Implicit pattern learning occurs frequently in real-world settings and operates on a broad range of stimuli, including many forms of visual modality input⁵⁹; merely being in an environment with recurring elements is sufficient to implicitly learn associations^{60,61}. **Individual differences in implicit learning emerge early in development^{60,62,63}, with performance approximating adult levels⁶⁴. While extended longitudinal measurement is challenging, extant evidence suggests that differences in IL-pat are relatively stable across time⁶⁵, including during early childhood⁶⁶, and may be genetically determined^{67,68}**. Thus, any impacts of IL-pat on intuitions of order are likely to begin in young childhood via exposure to learnable patterns in the environment. If IL-pat affects belief, it could thus manifest in individual differences in change in strength of religious belief from childhood to adulthood.

Relevant text also addresses the directional interpretation of the observed association between IL-pat and more explicit constructs based on task attributes of the SRTT.

In the Introduction on page 8:

Multiple experimental paradigms have been developed to investigate IL-pat. Perhaps the most widely used measure is the Serial Reaction Time Task (SRTT)^{77,78}, which prior work suggests is reflective of environmental IL-pat²¹, and which has recently been shown to load heavily on a broad implicit learning ability factor⁶⁵. Notably, extant evidence indicates that IL-pat during the SRTT is not improved – and may actually be impaired – by top-down influences, including instructions to look for patterns^{61,69,79,80} (see Methods). In addition, modifications of the SRTT have been devised to limit opportunities for explicit awareness of

patterns that could be a precondition for top-down influence^{69,81}. The SRTT thus presents a good experimental measure to investigate putative bottom-up link of IL-pat to belief: stronger IL-pat might contribute to stronger belief, but a belief-related top-down bias to seek patterns is unlikely to strengthen IL-pat, thus, any positive association observed between IL-pat and belief is most likely to be bottom-up.

In the Methods on page 30:

In addition, the opportunity to explicitly recall/rehearse target sequences in an effort to explicitly identify patterns for patterns was mitigated by removing the response-stimulus-interval (RSI) between key-press and the appearance of the next target⁸¹. During no-RSI versions of the SRTT, the next target appears immediately after the correct response is made; participants cannot use time between target appearances (because the next target appears immediately) to consciously attempt to identify a pattern structure.

In the Discussion on pages 20-22:

The results of the current study are consistent with a bottom-up pathway of effect by which stronger implicit learning of patterns/order leads to belief in an intervening/ordering god. While top-down pathways are also conceivable whereby stronger belief in an intervening deity leads individuals to more effectively learn implicit patterns, and/or whereby religious individuals may be more likely to search for patterns because of an explicit belief in the ordering influence of a deity, a number of empirically-based indicators make such explanations less likely in this case. First, prior research has demonstrated that explicit searching does not lead to faster responding during the SRTT, and may actually hinder performance^{61,69,79}. For example, Fletcher and colleagues⁷⁹ showed that explicitly directing participants to search for a pattern impeded implicit learning on the SRTT (i.e., slower RT during the task), putatively because this explicit search disrupts elements of task performance that facilitate IL-pat. Thus, even if interventionist belief promotes explicit searching for patterns, it is unlikely that this would have led to improved performance on the SRTT in the present study.

A bottom-up interpretation of the present data is further bolstered by modifications to the SRTT paradigm that were incorporated into the present study. Namely, the sequence advanced (i.e., the next target appeared) immediately after the participant provided a correct response. This so-called no-response-time-interval (no-RSI) version of the SRTT has been demonstrated by Destrebecqz and Cleeremans and others^{69,81} to yield sequence learning without explicit knowledge acquisition. Removing time between target presentations is intended to ensure that learners are consistently engaged in the task and do not have the opportunity to consciously attempt to recall/rehearse the target sequence in an effort to identify patterns while waiting for the next target to appear. Consistent with prior work, we did not observe a significant correlation between sequence learning (i.e., rate

of change on pattern sequences relative to random ones) and explicit awareness, indicating that learning was implicit. **The lack of association between CRT performance and implicit learning on the SRTT may further suggest that differences in IL-pat were unlikely to be driven by a bias towards effortful top-down thinking.**

The observed association between IL-pat and change in belief from childhood to adulthood can be taken as further indication against the likelihood of religious belief exerting a top-down influence on IL-pat. Change in belief across the lifespan, and particularly from childhood to adulthood, is not uncommon^{41,88}. Thus, to the extent that individual differences in implicit learning are present at early ages^{60,62,63}, and may be stable across time⁶⁴⁻⁶⁶, it appears more likely that differences in IL-pat could drive changes in belief across development than that changes in belief, or adult level of belief in an interventionist god as measured in the present study, are primary drivers of differences in IL-pat. Evidence of genetic influences on IL-pat^{67,68}, further suggests sources of individual differences in IL-pat that may be present prior to, and largely independent of, top-down influences.

And later in the Discussion on page 23:

Regarding the specific directional pathway represented by our mediation models, this directionality is based on a hypothesized pathway of effect within which there are *a priori* reasons to position intuitions of universal order prior to interventionist belief (i.e., as the mediator and dependent variable respectively). Specifically, the directionality of the pathway is based on the *a priori* consideration that implicit learning of patterns/order in environmental stimuli is less likely to directly yield specific beliefs about deities, and more likely to yield a general sense/intuition that there is ambient order. Broad intuitions about the presence of order do not depend on externally learned narratives about the identities and powers of deities, and thus appear more likely to arise intrinsically (i.e., within the individual, directly from that individual's bottom-up implicit learning of patterns/order). Relatedly, intuitions of order appear more conceptually proximate to the content being learned via implicit learning of order than to the content of beliefs about deistic intervention (i.e., both the intuitions and the implicitly learned content directly concern order itself, rather than explanations of order). Thus, a progression from IL-pat to UO appears more likely than a direct leap from implicit learning to religious narratives about interventionist deities. As noted above, the hypothesized pathway of effect is also based on prominent extant accounts, developed to interpret decades of research on implicit learning, which indicate that IL-pat gives rise to intuitions of order^{17,20-25}. Thus, in view of theoretical considerations and extant literature, we hypothesized a pathway of effect whereby implicit learning influences UO and UO influences interventionist belief (i.e., implicit learning yields broad intuitions of order that predispose individuals toward belief narratives that fit these intuitions).

Nonetheless, the directional interpretation is also qualified within the Discussion to acknowledge that it is extremely unlikely to provide a complete accounting of the influences on religious belief formation, and that top-down influences on bottom-up processing are still entirely possible (Page 22-23):

While the bottom-up directionality of effect we have outlined is consistent with prior work on the neurocognitive bases of religious belief^{1,2,9-12,16}, it is not possible – and is not the intent of the present study – to conclusively rule out any degree of top-down influence on the relationship between IL-pat and religious belief. Indeed, a complete rendering of the cognitive and environmental influences on a phenotype as complex as religious belief very likely includes a multiplex of interconnected “loop” architectures between bottom-up and top-down processes. Examples of top-down influences on automatic perceptual information processing are abundant^{19,23,98}. One pertinent example is research showing that induced “meaning threats” devised to challenge participants’ existential sense of meaning were associated with better implicit learning in an artificial grammar task⁹⁹. Religious belief is associated with a desire for meaning^{4,11,46}, and it is conceivable that more religious participants might tonically feel greater threat to meaning, which may have an influence on at least some forms of IL-pat. It should be noted that the improvement in grammar learning in this prior work was linked to an increased explicit motivation to find grammatical letter strings in the grammar learning task, whereas extant evidence indicates such explicit motivation does not improve learning on the SRTT^{61,69,79}. More directly to the point, even if it is the case that some amount of top-down influence has bearing on implicit learning, this would not necessarily be inconsistent with bottom-up influences of implicit learning on religious belief. While the present study focused on individual differences in a strongly bottom-up IL-pat paradigm to explore the thus far untested relationship of bottom-up IL-pat to belief, the intent of the present research is not to suggest that all bottom-up influences on belief, and perhaps not even the IL-pat studied here, operate entirely independently of top-down processes.

To re-state the above, the directionality modelled in the mediation analysis is not drawn from the result of the model; rather, the model we constructed reflects the theorized directionality of a hypothesized pathway of effect. Empirically, the most crucial results are that IL-pat was correlated with UO, that UO was correlated with IB, and that IL-pat was correlated with IB. Had these correlations not been observed, our theoretical model would not be supported by these data. In addition, if analysis of the model we constructed to reflect the hypothesized directional pathway did not yield a significant indirect effect of IL-pat on IB via UO, that would have indicated against the theorized directionality of the pathway. Thus, while the analyses we conducted are not a basis for making a directional hypothesis, they suggest that the *a priori* directional hypothesis we formulated on theoretical grounds remains plausible *a posteriori* in view of the data.

This is noted in the Results on page 15:

Nonetheless, results suggest that the *a priori* directional hypothesis remains plausible *a posteriori* in view of the data. By contrast, if modeling the pathway had not yielded significant indirect effects, this would have empirically indicated against the hypothesized model.

Based on the reviewers' suggestion, exploratory analyses were conducted to test a model in which IB was positioned as the mediator and UO as the dependent variable. Additional exploratory models tested the finer-grained relationships between IL-pat, UO, and each of the three belief variables from which IB was derived. We also ran a fully reversed model in which the IV and DV were exchanged (i.e., IB/Belief Change → UO → IL-pat). These exploratory analyses are now noted in the Results on page 15-16:

Though the hypothesized pathway of effect is based on theory and prior literature, rather than empirical comparisons between competing models, further exploratory analyses (Supplementary Information, Supplementary Figs. 6, 7) in which IB and individual belief measures were modeled as mediators, and in which IB was modeled as the independent variable and IL-pat as the dependent variable, provided modest but consistent additional support for the hypothesized model.

And described in Supplementary Information on pages 17-18:

First, we tested fully reversed mediation models in which the belief measures were positioned as the independent variables and IL-pat was positioned as the dependent variable (i.e., Belief → UO → IL-pat; Supplementary Fig. 6.) The model in which IB was the independent variable yielded a nonsignificant indirect effect ($\beta = 0.07$, $P = 0.08$; bootstrapped bias-corrected 95% CI, as compared to $\beta = 0.07$, $P = 0.005$ in the original model) and a smaller proportion of the total effect that was mediated (0.35 compared to 0.48 in the original model). The model in which Belief Change was the independent variable yielded an indirect effect that remained marginally significant but also accounted for a smaller proportion of the total effect ($P = 0.047$; proportion of total effect that was mediated = 0.38 compared to 0.43 in the original model). A model in which IB was positioned as the mediator (rather than the dependent variable) and UO was positioned as the dependent variable (instead of the mediator; Supplementary Fig. 7) yielded an indirect effect ($\beta = .087$, $P = .022$, as compared to $\beta = .090$, $P = 0.005$ in the original model), and the proportion of the total effect that was mediated was again smaller than in the original model (0.44 compared to 0.48 in the original model).

Further exploratory analyses tested whether UO mediated the effects of IL-pat on each of the three individual belief measures from which the IB component score was derived (BDIS, Self Overlap, and World Overlap). These analyses revealed significant indirect effects in each case (all P s ≤ 0.006 , bootstrapped bias-corrected 95% CI). By contrast, models in which

the mediator and DV were reversed for each of these belief measures (such that each belief measure was positioned as the mediator, with UO as the dependent variable), revealed nonsignificant indirect effects for the BDIS and World Overlap variables, and smaller proportions of the total effect mediated for all three models (BDIS = 0.33; World Overlap = 0.30; Self Overlap = 0.36) compared to what was observed with UO as the mediator (BDIS = 0.73; World Overlap = 0.43; Self Overlap = 0.37). Similarly, the fully reversed mediation models with each individual IB measure positioned as the independent variable and IL-pat as the dependent variable yielded a nonsignificant indirect effect for the Self Overlap model, and smaller proportions of the total effect mediated for all three models (BDIS = 0.65 $p = .017$; World Overlap = 0.36 $p = .032$; Self Overlap = 0.28) than in the above-described models with IL-pat as the independent variable and each of the three IB measures as dependent variables. Thus, taken as a group, these exploratory analyses suggest modest but consistent empirical support for the hypothesized models relative to alternatives models.

Supplementary Figure 6. Fully reversed mediation model. Alternative models position religious belief variables of a) IB, and b) Belief Change as the IV and implicit pattern learning as the DV. Mediated effects are nonsignificant (a) and weaker (b) than those indicated in Fig. 3 in the main text.

Supplementary Figure 7. Alternative mediation models. Alternative models positioned religious belief variables of a) IB, and b) Belief Change as the MV and UO as DV. Evidence of mediation is nominally weaker than in the theoretically-based model (Fig. 3) described in the main text.

Thus, while the hypothesized directionality was based on an *a priori* theory, it is instructive that all alternative models indicated nominally weaker mediated effects than the originally hypothesized model, with some alternative models yielding nonsignificant indirect effects. Nonetheless, it is primarily because the mediation modeling employed in the present study was intended to assess the viability of the specific hypothesized directional pathway in view of the data, rather than to compare between multiple hypotheses, that we have retained a description of the model and results in the text. To underscore the point, even if switching the DV and MV had yielded a stronger indirect effect, the directionality of our originally hypothesized model would still be rooted in theory and viable *a posteriori* so long as a significant indirect effect was observed. However, if the reviewers feel strongly on this point, we are open to the possibility of removing the mediation modeling elements of the manuscript, and focusing only on the concurrence of the relevant bivariate correlations.

Relatedly, we have revised the manuscript to clarify the ultimately correlational nature of mediation analysis, and emphasize that statistical mediation does not provide grounds for determining causation (also see response to Point 10).

This is first addressed in the initial instance of mediation within the Results on page 15:

UO was also significantly correlated with both IB (U.S.: $r = 0.48$, $P < 0.001$; European sample: $r = 0.56$, $P < 0.001$) and Belief Change (U.S. sample: $r = 0.36$, $P < 0.001$; Fig. 3., b path). Consistent with our hypotheses, UO significantly mediated the effects of IL-pat on both IB (indirect effect $P = 0.005$) and Belief Change (indirect effect $P = 0.006$; bootstrapped bias-corrected 95% CI; Fig. 3) in the U.S. sample. **These analyses do not establish causation because mediation analysis is fundamentally correlational.**

We then expand on this to emphasize that causal claims cannot be established via mediation analysis in the Discussion on page 24:

It is important to emphasize that the mediation analyses we conducted to test models of the hypothesized pathway should not be taken as establishing causation. Mediation is fundamentally a correlation-based technique, and is frequently over-interpreted to make causal claims¹⁰⁰⁻¹⁰². Establishing causation requires a comprehensive research program that should ideally include longitudinal and intervention-based paradigms, a careful accounting of the influence of a host of measured and unmeasured mediating variables, and replication across varied experimental paradigms¹⁰⁰. While this is beyond the scope of a single study, the pathway of effect modeled in the present study provides a framework for more comprehensive investigations of causation in the observed relationships, and the present data on belief change from childhood to adulthood suggest that longitudinal study might be particularly informative.

Point 9: I urge you to make your data open access. I understand the desire to embargo data until such a time as you are "done" with it, but perhaps the threat of competition will spur you to complete the other analyses and papers you plan for this dataset. The merit of making your data open access is enhanced by the admitted messiness of your raw data. You have made various choices about exclusion, and readers have no real way of knowing how much difference those choices make unless they have access to the data (or if you run robustness checks). Messy raw data should not be penalized, of course: especially when it comes to cross-cultural work in difficult circumstances.

Response: We agree with the reviewer. We undertook several precautions during and after data collection – particularly in Afghanistan – to identify potentially invalid data. The revised documents now include a complete csv file for all variables relevant to the present study in both samples, including variables that were not used in the main analyses. Further, the csv file contains columns that indicate which tasks were excluded for each participant (and why). Readers interested in understanding the statistical ramifications of various exclusion decisions may include or exclude participants using these columns. To aid in this process, we have also shared the STATA code that was used to perform the analyses described in the paper. Within this file, all major inferential statistical tests are performed on the final sample used in the manuscript, as well as on larger samples with less stringent exclusionary criteria. These materials have also been posted to the OSF (<https://osf.io/g5ywe/>). The QC section of Supplementary Information now includes all main-text results using the less stringent exclusionary criteria (see Supplementary Tables 10, 11, which are also included below) to provide robustness checks for the decisions to exclude participants, particularly in the Afghan sample.

As evident in these materials, the various exclusion choices do not meaningfully alter the results. Although maximizing the size of our samples would be desirable, all else being equal, we believe that the exclusions (especially in Afghanistan) are likely to improve the quality of our data, such that participants who did not exhibit appropriate

levels of attention and/or comprehension are removed. The QC section of Supplementary Information now reads as follows (pages 2-4):

Substantial data loss/exclusion for the Afghan sample was anticipated because issues related to cross-cultural relevance and comprehension are common for data collection in Afghanistan¹. Local experimenters were hired to conduct data collection sessions with local participants in Kabul, consistent with best practices based on Z.W.'s extensive experience collecting data in Afghanistan, e.g., as director of the Asia Foundation's Survey of the Afghan People¹. Also consistent with best practices, we implemented additional post-collection quality control protocols to ensure that all data were properly collected and recorded by the local experimenters. **This involved screening of the data to identify any irregularities associated with individual experimenters, reviewing experimenter notes for problems with study participants, and interviews with experimenters after the study to determine whether participants properly understood tasks, and whether the experimenters themselves had properly understood and implemented all study procedures.**

We received data for 354 participants in Afghanistan. The majority of data collection occurred on site at the Afghan Cultural House in Kabul. Upon reviewing the data, we identified irregularities in data collection for two local experimenters (survey and task responses collected by these experimenters showed very low variability within and across participants, and response times were consistently different from the rest of the sample). Because these two experimenters were not able to satisfactorily account for their data, we excluded all 157 participants from whom these two experimenters collected data (all during the final months of the collection period). Two additional participants were excluded because notes at the time of the study indicated that they were responding dishonestly and/or that they were uncooperative with study procedures (e.g., one participant expressed unwillingness to cooperate because they perceived that they were "working with Westerners").

Among the 195 remaining participants, 8 were excluded from analyses for failing to finish the survey (i.e., they did not reach the end of the survey) and 9 were removed for improper completion times on the survey measures (defined as < 25 minutes or > 4 hours). An additional 14 participants were excluded because of inappropriate responding during the SRTT (defined as average RTs on the SRTT of >2000ms or <100 ms). In accordance with principles of informed consent and the Georgetown University institutional review board, participants were permitted to skip questions as desired. Sixteen participants did not complete key measures related to the hypotheses addressed by the current study (i.e., SRTT, IB measures, covariate personality/belief variables), and were thus excluded from analyses. After these exclusions, the final Afghan sample was 148 participants.

In the U.S. sample, the initial cohort of 240 was reduced to 199 for the following reasons. Five participants were removed because they did not comply with protocol for the testing session. Five participants did not complete the SRTT and two participants did not complete all of the belief measures. An additional 29 participants experienced a malfunction in the stimulus

presentation during the SRTT. Specifically, the first random block consisted of only 22 targets (rather than 50). Theoretically, the reduced number of trials and task duration might have diminished participants' opportunity to increase their speed of responding over the course of the block. These 29 participants were conservatively removed from the primary analyses reported in the main text.

The online sample (i.e., European sample, Supplementary Table 2) was recruited through *Prolific*. Recent evidence indicates that *Prolific* offers higher quality data (e.g., based on participant attention and honest responding) and a more diverse participant pool compared to alternative online research platforms, such as Amazon Mechanical Turk^{2,3}. The survey contained two attention check questions embedded within the study measures (e.g., "Blue is virtually always the same color as orange?"). Twelve participants did not correctly answer both of these attention check questions and were excluded from analysis. One additional participant was excluded for being under 18 years of age at the time of the study. The constituency of this sample was primarily (83%) European (Supplementary Table 2).

In sum, the vast majority of all exclusions were related to either experimenter unreliability (in the Afghan sample) or a malfunction in stimulus presentation (in the U.S. sample). The impact of all of these decisions on the results reported in the main text are displayed in Supplementary Tables 10, 11. Results were largely unchanged across different inclusion/exclusion criteria, indicating that these inclusion/exclusion determinations did not substantially impact the observed findings.

Supplementary Table 10. Impact of QC data inclusion/exclusion on IB, Belief Change, and Universal Order in U.S. sample

	Interventionist Belief					Belief Change					Universal Order				
	β	B	SE	t	P	β	B	SE	t	P	β	B	SE	t	P
Manuscript Sample (N=199)															
IL-Pat	0.17	1.24	0.47	2.65	0.009	0.15	3.53	1.67	2.11	0.036	0.18	5.13	1.90	2.69	0.008
Schizotypy	0.11	0.53	0.30	1.75	0.082	-0.07	-1.10	1.08	-1.02	0.311	0.06	1.06	1.23	0.86	0.393
Parents' Belief	0.38	0.15	0.03	5.96	0.000	-0.04	-0.05	0.09	-0.53	0.595	0.28	0.42	0.10	4.10	0.000
No QC (N=233)															
IL-Pat	0.18	1.31	0.43	3.02	0.003	0.15	3.53	1.53	2.31	0.022	0.19	5.50	1.74	3.16	0.002
Schizotypy	0.14	0.64	0.28	2.32	0.021	-0.05	-0.78	0.97	-0.80	0.425	0.12	2.15	1.11	1.94	0.054
Parents' Belief	0.39	0.15	0.02	6.60	0.000	-0.04	-0.05	0.08	-0.68	0.499	0.28	0.41	0.09	4.51	0.000
No SRTT QC (N=228)															
IL-Pat	0.17	1.25	0.43	2.89	0.004	0.15	3.52	1.55	2.27	0.024	0.19	5.25	1.76	2.99	0.003
Schizotypy	0.13	0.62	0.28	2.22	0.027	-0.06	-0.89	0.99	-0.90	0.371	0.12	2.23	1.13	1.97	0.050
Parents' Belief	0.39	0.15	0.02	6.63	0.000	-0.05	-0.06	0.08	-0.73	0.468	0.28	0.41	0.09	4.51	0.000

Supplementary Table 11. Impact of QC data inclusion/exclusions on IB and Belief Change in Afghan sample

	Interventionist Belief					Belief Change				
	β	B	SE	t	P	β	B	SE	t	P
Manuscript Sample (N=148)										
IL-Pat	0.26	1.60	0.48	3.33	0.001	0.17	3.53	1.67	2.11	0.036
Schizotypy	0.35	1.40	0.30	4.72	0.000	0.26	3.37	1.04	3.24	0.001
Parents' Belief	0.09	0.11	0.09	1.22	0.225	0.10	0.38	0.31	1.25	0.212
No QC (N=176)										
IL-Pat	0.21	1.38	0.46	3.00	0.003	0.16	3.23	1.52	2.12	0.036
Schizotypy	0.33	1.29	0.28	4.69	0.000	0.20	2.50	0.92	2.73	0.007
Parents' Belief	0.05	0.06	0.08	0.69	0.489	0.10	0.35	0.28	1.27	0.207
No SRTT QC (N=162)										
IL-Pat	0.24	1.51	0.47	3.21	0.002	0.16	3.24	1.62	2.00	0.047
Schizotypy	0.35	1.37	0.28	4.86	0.000	0.23	2.91	0.97	3.00	0.003
Parents' Belief	0.10	0.11	0.09	1.29	0.200	0.10	0.39	0.30	1.29	0.200
No Questionnaire QC (N=162)										
IL-Pat	0.23	1.45	0.47	3.10	0.002	0.18	3.50	1.57	2.23	0.027
Schizotypy	0.33	1.31	0.29	4.52	0.000	0.23	2.88	0.97	2.95	0.004
Parents' Belief	0.04	0.05	0.08	0.60	0.547	0.10	0.35	0.28	1.24	0.217

Point 10: The most interesting thing about your paper is the purported IV, implicit learning as an individual difference variable. I say "purported" because it is not as obvious to me that religiosity cannot shape implicit learning tendencies. Certainly, religious upbringing might shape implicit learning. Now, fortunately, you have as one of your DVs a measure of religious *change*, which--I agree--is a much less plausible causal factor for implicit learning. The case for implicit learning being the IV needs to be made more strongly, I think, and what I have just said might provide a start.

Response. Thank you for this suggestion. We agree that it is important to more clearly justify the directionality we hypothesize (i.e., implicit pattern learning as an IV). We have expanded sections of the Introduction, Methods, and Discussion to further describe the bases for hypothesizing implicit learning as the IV, and for interpreting the data from the SRTT as primarily reflecting bottom-up processing. Some of the largest revisions and most critical pieces of retained text regarding the hypothesized directional pathway of effect are described and excerpted above in the response to Point 8. Other new and retained text related to this point, and text particularly related to implicit learning as the IV, are excerpted here.

In the Introduction on page 8:

Multiple experimental paradigms have been developed to investigate IL-pat. Perhaps the most widely used measure is the Serial Reaction Time Task

(SRTT)^{77,78}, which prior work suggests is reflective of environmental IL-pat²¹, and which has recently been shown to load heavily on a broad implicit learning ability factor⁶⁵. Notably, extant evidence indicates that IL-pat during the SRTT is not improved – and may actually be impaired – by top-down influences, including instructions to look for patterns^{61,69,79,80} (see Methods). In addition, modifications of the SRTT have been devised to limit opportunities for explicit awareness of patterns that could be a precondition for top-down influence^{69,81}. The SRTT thus presents a good experimental measure to investigate putative bottom-up link of IL-pat to belief: stronger IL-pat might contribute to stronger belief, but a belief-related top-down bias to seek patterns is unlikely to strengthen IL-pat, thus, any positive association observed between IL-pat and belief is most likely to be bottom-up.

In the Discussion on pages 20-21:

The results of the current study are consistent with a bottom-up pathway of effect by which stronger implicit learning of patterns/order leads to belief in an intervening/ordering god. While top-down pathways are also conceivable whereby stronger belief in an intervening deity leads individuals to more effectively learn implicit patterns, and/or whereby religious individuals may be more likely to search for patterns because of an explicit belief in the ordering influence of a deity, a number of empirically-based indicators make such explanations less likely in this case. First, prior research has demonstrated that explicit searching does not lead to faster responding during the SRTT, and may actually hinder performance^{61,69,79}. For example, Fletcher and colleagues⁷⁹ showed that explicitly directing participants to search for a pattern impeded implicit learning on the SRTT (i.e., slower RT during the task), putatively because this explicit search disrupts elements of task performance that facilitate IL-pat. Thus, even if interventionist belief promotes explicit searching for patterns, it is unlikely that this would have led to improved performance on the SRTT in the present study.

A bottom-up interpretation of the present data is further bolstered by modifications to the SRTT paradigm that were incorporated into the present study. Namely, the sequence advanced (i.e., the next target appeared) immediately after the participant provided a correct response. This so-called no-response-time-interval (no-RTI) version of the SRTT has been demonstrated by Destrebecqz and Cleeremans and others^{69,81} to yield sequence learning without explicit knowledge acquisition. Removing time between target presentations is intended to ensure that learners are consistently engaged in the task and do not have the opportunity to consciously attempt to recall/rehearse the target sequence in an effort to identify patterns while waiting for the next target to appear. Consistent with prior work, we did not observe a significant correlation between sequence learning (i.e., rate of change on pattern sequences relative to random ones) and explicit awareness, indicating that learning was implicit.

As the reviewer notes, the relationship of implicit pattern learning to the Belief Change variable empirically strengthens the case for bottom-up directionality. This argument is bolstered by research indicating the emergence of individual differences in implicit learning early in childhood, that IL-pat may be stable across time, and pointing to genetic influences on IL-pat. This point is first addressed in the Introduction on page 7:

Individual differences in implicit learning emerge early in development^{60,62,63}, with performance approximating adult levels⁶⁴. While extended longitudinal measurement is challenging, extant evidence suggests that differences in IL-pat are relatively stable across time⁶⁵, including during early childhood⁶⁶, and may be genetically determined^{67,68}. Thus, any impacts of IL-pat on intuitions of order are likely to begin in young childhood via exposure to learnable patterns in the environment. If IL-pat affects belief, it could thus manifest in individual differences in change in strength of religious belief from childhood to adulthood.

We have also now expanded the Discussion to address this point more clearly in view of the observed data (page 21):

The observed association between IL-pat and change in belief from childhood to adulthood can be taken as further indication against the likelihood of religious belief exerting a top-down influence on IL-pat. Change in belief across the lifespan, and particularly from childhood to adulthood, is not uncommon^{41,88}. Thus, to the extent that individual differences in implicit learning are present at early ages^{60,62,63}, and may be stable across time⁶⁴⁻⁶⁶, it appears more likely that differences in IL-pat could drive changes in belief across development than that changes in belief, or adult level of belief in an interventionist god as measured in the present study, are primary drivers of differences in IL-pat.

Continuing on from the above, we also note evidence of genetic influences on IL-pat, which would further suggest IL-pat as a bottom-up IV:

Evidence of genetic influences on IL-pat^{67,68}, further suggests sources of individual differences in IL-pat that may be present prior to, and largely independent of, top-down influences

We have also retained text positing interpretation of the relationship between IL-pat and Belief Change later in the Discussion on page 27:

The observed association of IL-pat with change in belief from childhood to adulthood (distinct from the powerful influence of parental belief) suggests that the emergence of individual differences in IL-pat in early childhood⁶⁹ might set individuals on bottom-up trajectories whereby literally perceiving the world differently (i.e., differential processing of patterned visual information available

in the environment) helps to move them either towards or away from religious belief.

The above arguments notwithstanding, this study was not intended to rule out the possibility of any top-down influence of religion on implicit learning. Such influence is possible, and could quite conceivably operate concurrently with bottom-up influences of implicit learning. We attempt to nuance our interpretation by conveying this possibility in the Discussion on pages 22-23:

While the bottom-up directionality of effect we have outlined is consistent with prior work on the neurocognitive bases of religious belief^{1,2,9-12,16}, it is not possible – and is not the intent of the present study – to conclusively rule out any degree of top-down influence on the relationship between IL-pat and religious belief. Indeed, a complete rendering of the cognitive and environmental influences on a phenotype as complex as religious belief very likely includes a multiplex of interconnected “loop” architectures between bottom-up and top-down processes. Examples of top-down influences on automatic perceptual information processing are abundant^{19,23,98}. One pertinent example is research showing that induced “meaning threats” devised to challenge participants’ existential sense of meaning were associated with better implicit learning in an artificial grammar task⁹⁹. Religious belief is associated with a desire for meaning^{4,11,46}, and it is conceivable that more religious participants might tonically feel greater threat to meaning, which may have an influence on at least some forms of IL-pat. It should be noted that the improvement in grammar learning in this prior work was linked to an increased explicit motivation to find grammatical letter strings in the grammar learning task, whereas extant evidence indicates such explicit motivation does not improve learning on the SRTT^{61,69,79}. More directly to the point, even if it is the case that some amount of top-down influence has bearing on implicit learning, this would not necessarily be inconsistent with bottom-up influences of implicit learning on religious belief. While the present study focused on individual differences in a strongly bottom-up IL-pat paradigm to explore the thus far untested relationship of bottom-up IL-pat to belief, the intent of the present research is not to suggest that all bottom-up influences on belief, and perhaps not even the IL-pat studied here, operate entirely independently of top-down processes.

Point 11: As mentioned above, there is a theoretical question to be asked about the relationship between implicit learning as measured by your task and implicit cognitive style as measured by, for example, the Cognitive Reflection Test. If they are the same thing, then this paper does not really report a novel result. So you are motivated to argue that they are different. But this is a difficult argument to make because you cannot do so on empirical groups unless you run another study that looks at the correlation between implicit learning and cognitive style. As far as I know, there has not been a study like this yet. You cite the Woolhouse and Bayne study, but that one relies on a self-report measure of cognitive style: perhaps that is good enough, but I am disinclined to think so. There is

nothing really wrong with self-report measures, but cognitive style is a hybrid of attitude and ability, and self-report measures of ability are questionable indeed. In any case, the argument needs to be made that implicit learning is especially interesting to look at independent of cognitive style.

Response: We thank the reviewer for this suggestion. The new data collections that included the CRT have meaningfully improved our manuscript. These data provide an empirical basis for distinguishing IL-pat from the kind of analytic vs. intuitive thinking measured by the CRT, and the effects of IL-pat on the belief and UO variables remained after including CRT performance as a covariate. Greater detail is provided above in the response to Point 5.

Point 12: Finally, it is not clear what the theoretical motivation behind the mediation model is. Is there a debate about the relationship between "order" beliefs (others might say "teleological beliefs"; see for example work by Deb Kelemenn, Konika Banerjee, Beth Heywood) and religious beliefs that you are trying to resolve? One can, of course, test whatever model we fancy, but the reader should be led to believe that there is some kind of theoretical payoff. Otherwise, it looks like you have just put together a plausible model to pad the paper. That's a look to avoid.

Response: We have taken substantial steps to more clearly indicate the theoretical motivation for our mediation modelling based on *a priori* hypothesized pathways of effect, while also clearly acknowledging the fundamental limitations of mediation analysis. The many revisions to address this point, and much of the relevant retained text, is described and excerpted above in the responses to points 8 and 10. As indicated in the response to point 8, while we think that, with appropriate interpretive caution, the mediation modeling is worthwhile in the present context of modeling an *a priori* directional pathway to assess its *a posteriori* viability, if the reviewers feel strongly that it should be removed, we are open to that possibility.

Reviewer #2

Remarks to the Author:

Point 1: Thanks for the opportunity to review this interesting paper. The authors report that implicit learning of patterns in a computerized task predicts stronger belief in an intervening god and increased belief from childhood to adulthood. Impressively, the authors replicate this finding in two religiously diverse cultural contexts, the US and Afghanistan. They also adduce several methodological considerations in support of a causal story: implicit awareness of patterns in the environment predisposes people to believe in an intervening god (rather than the reverse). I found their arguments in this respect quite persuasive (though I was less convinced by their mediation analysis in the US sample).

Overall, though I like the study and think the result is interesting and potentially important, I have two main concerns which bear on the editorial decision. I also have some more minor recommendations for improving the paper.

Response: We greatly appreciate this reviewer's detailed feedback and critical assessment of methodological and statistical approaches, as well as the positive overall evaluation of the manuscript. We have made substantial changes in the revised version and collected new experimental data to inform interpretations relevant to the reviewer's questions. Data and code for all study variables and analyses are now fully available on the OSF. More generally, the changes that we have made following the recommendations of this reviewer have made for a substantially improved paper.

Point 2: The first issue is methodological. To my mind, in 2019, a psychology paper in a top outlet like Nature Communications should be pre-registered, and the data should be available to reviewers at the point of submission (rather than being "available from the corresponding author upon request"). Neither is the case here. The Reporting Summary and Supplementary Information indicate that a large number of participants were excluded (about 42% of the tested sample, across both cohorts), for all sorts of potentially arbitrary reasons (e.g., taking less than 25 minutes to complete the survey). In addition, the study protocol included administration of a wide range of measures that are not reported here (including measures of teleological bias, magical ideation and locus of control). These considerations raise the spectre of undue analytic flexibility and selective reporting. To cite just one example: To measure belief change, the authors had participants report their strength of belief in God from age 6 to their age at the time of the study, in 3-year intervals. But the score they computed only makes use of the first and last of these ratings. Why bother collecting the other intervals if the authors knew in advance that this was how they would compute this belief-change index?

Response: The revised submission now contains a complete csv file for all variables relevant to the present study in both samples, including additional variables that were not included in the main analyses (e.g., the teleological bias, magical ideation and locus of control variables). These files are also available on the OSF (<https://osf.io/g5ywe/>).

Further, the csv file contains columns that indicate which tasks were excluded for each participant (and why). Readers interested in understanding the statistical ramifications of various exclusion decisions may include or exclude participants using these columns. To aid in this process, we have also shared the STATA code that was used to perform the analyses described in the paper. Within this file, all major inferential statistical tests are performed on the final sample used in the manuscript, as well as on larger samples with less stringent exclusionary criteria. We undertook several precautions during and after data collection – particularly in Afghanistan – to identify potentially invalid data. The QC section of Supplementary Information now includes all main-text results using the less stringent exclusionary criteria to provide robustness checks for the decisions to exclude participants, particularly in the Afghan sample (see Supplementary Tables 10, 11, which are also included below).

As is evident in these materials, the various exclusion choices do not meaningfully alter the results. Although maximizing the size of our samples would be desirable, all else being equal, we believe that the exclusions (especially in Afghanistan) are likely to improve the quality of our data, such that participants who did not exhibit appropriate levels of attention and/or comprehension are removed. The QC section of Supplementary Information now reads as follows (pages 2 - 4):

Substantial data loss/exclusion for the Afghan sample was anticipated because issues related to cross-cultural relevance and comprehension are common for data collection in Afghanistan¹. Local experimenters were hired to conduct data collection sessions with local participants in Kabul, consistent with best practices based on Z.W.'s extensive experience collecting data in Afghanistan, e.g., as director of the Asia Foundation's Survey of the Afghan People¹. Also consistent with best practices, we implemented additional post-collection quality control protocols to ensure that all data were properly collected and recorded by the local experimenters. **This involved screening of the data to identify any irregularities associated with individual experimenters, reviewing experimenter notes for problems with study participants, and interviews with experimenters after the study to determine whether participants properly understood tasks, and whether the experimenters themselves had properly understood and implemented all study procedures.**

We received data for 354 participants in Afghanistan. The majority of data collection occurred on site at the Afghan Cultural House in Kabul. Upon reviewing the data, we identified irregularities in data collection for two local experimenters (survey and task responses collected by these experimenters showed very low variability within and across participants, and response times were consistently different from the rest of the sample). Because these two experimenters were not able to satisfactorily account for their data, we excluded all 157 participants from whom these two experimenters collected data (all during the final months of the collection period). Two additional participants were excluded because notes at the time of the study indicated that they were responding dishonestly and/or that they were uncooperative with study procedures (e.g., one participant expressed unwillingness to cooperate because they perceived that they were “working with Westerners”).

Among the 195 remaining participants, 8 were excluded from analyses for failing to finish the survey (i.e., they did not reach the end of the survey) and 9 were removed for improper completion times on the survey measures (defined as < 25 minutes or > 4 hours). An additional 14 participants were excluded because of inappropriate responding during the SRTT (defined as average RTs on the SRTT of >2000ms or <100 ms). In accordance with principles of informed consent and the Georgetown University institutional review board, participants were permitted to skip questions as desired. Sixteen participants did not complete key measures related to the hypotheses addressed by the current study (i.e., SRTT, IB measures, covariate personality/belief variables), and were thus excluded from analyses. After these exclusions, the final Afghan sample was 148 participants.

In the U.S. sample, the initial cohort of 240 was reduced to 199 for the following reasons. Five participants were removed because they did not comply with protocol for the testing session. Five participants did not complete the SRTT and two participants did not complete all of the belief measures. An additional 29 participants experienced a malfunction in the stimulus presentation during the SRTT. Specifically, the first random block consisted of only 22 targets (rather than 50). Theoretically, the reduced number of trials and task duration might have diminished participants' opportunity to increase their speed of responding over the course of the block. These 29 participants were conservatively removed from the primary analyses reported in the main text.

The online sample (i.e., European sample, Supplementary Table 2) was recruited through *Prolific*. Recent evidence indicates that *Prolific* offers higher quality data (e.g., based on participant attention and honest responding) and a more diverse participant pool compared to alternative online research platforms, such as Amazon Mechanical Turk^{2,3}. The survey contained two attention check questions embedded within the study measures (e.g., "Blue is virtually always the same color as orange?"). Twelve participants did not correctly answer both of these attention check questions and were excluded from analysis. One additional participant was excluded for being under 18 years of age at the time of the study. The constituency of this sample was primarily (83%) European (Supplementary Table 2).

In sum, the vast majority of all exclusions were related to either experimenter unreliability (in the Afghan sample) or a malfunction in stimulus presentation (in the U.S. sample). The impact of all of these decisions on the results reported in the main text are displayed in Supplementary Tables 10, 11. Results were largely unchanged across different inclusion/exclusion criteria, indicating that these inclusion/exclusion determinations did not substantially impact the observed findings.

Supplementary Table 10. Impact of QC data inclusion/exclusion on IB, Belief Change, and Universal Order in U.S. sample

	Interventionist Belief					Belief Change					Universal Order				
	β	B	SE	t	P	β	B	SE	t	P	β	B	SE	t	P
Manuscript Sample (N=199)															
IL-Pat	0.17	1.24	0.47	2.65	0.009	0.15	3.53	1.67	2.11	0.036	0.18	5.13	1.90	2.69	0.008
Schizotypy	0.11	0.53	0.30	1.75	0.082	-0.07	-1.10	1.08	-1.02	0.311	0.06	1.06	1.23	0.86	0.393
Parents' Belief	0.38	0.15	0.03	5.96	0.000	-0.04	-0.05	0.09	-0.53	0.595	0.28	0.42	0.10	4.10	0.000
No QC (N=233)															
IL-Pat	0.18	1.31	0.43	3.02	0.003	0.15	3.53	1.53	2.31	0.022	0.19	5.50	1.74	3.16	0.002
Schizotypy	0.14	0.64	0.28	2.32	0.021	-0.05	-0.78	0.97	-0.80	0.425	0.12	2.15	1.11	1.94	0.054
Parents' Belief	0.39	0.15	0.02	6.60	0.000	-0.04	-0.05	0.08	-0.68	0.499	0.28	0.41	0.09	4.51	0.000
No SRTT QC (N=228)															
IL-Pat	0.17	1.25	0.43	2.89	0.004	0.15	3.52	1.55	2.27	0.024	0.19	5.25	1.76	2.99	0.003
Schizotypy	0.13	0.62	0.28	2.22	0.027	-0.06	-0.89	0.99	-0.90	0.371	0.12	2.23	1.13	1.97	0.050
Parents' Belief	0.39	0.15	0.02	6.63	0.000	-0.05	-0.06	0.08	-0.73	0.468	0.28	0.41	0.09	4.51	0.000

Supplementary Table 11. Impact of QC data inclusion/exclusions on IB and Belief Change in Afghan sample

	Interventionist Belief					Belief Change				
	β	B	SE	t	P	β	B	SE	t	P
Manuscript Sample (N=148)										
IL-Pat	0.26	1.60	0.48	3.33	0.001	0.17	3.53	1.67	2.11	0.036
Schizotypy	0.35	1.40	0.30	4.72	0.000	0.26	3.37	1.04	3.24	0.001
Parents' Belief	0.09	0.11	0.09	1.22	0.225	0.10	0.38	0.31	1.25	0.212
No QC (N=176)										
IL-Pat	0.21	1.38	0.46	3.00	0.003	0.16	3.23	1.52	2.12	0.036
Schizotypy	0.33	1.29	0.28	4.69	0.000	0.20	2.50	0.92	2.73	0.007
Parents' Belief	0.05	0.06	0.08	0.69	0.489	0.10	0.35	0.28	1.27	0.207
No SRTT QC (N=162)										
IL-Pat	0.24	1.51	0.47	3.21	0.002	0.16	3.24	1.62	2.00	0.047
Schizotypy	0.35	1.37	0.28	4.86	0.000	0.23	2.91	0.97	3.00	0.003
Parents' Belief	0.10	0.11	0.09	1.29	0.200	0.10	0.39	0.30	1.29	0.200
No Questionnaire QC (N=162)										
IL-Pat	0.23	1.45	0.47	3.10	0.002	0.18	3.50	1.57	2.23	0.027
Schizotypy	0.33	1.31	0.29	4.52	0.000	0.23	2.88	0.97	2.95	0.004
Parents' Belief	0.04	0.05	0.08	0.60	0.547	0.10	0.35	0.28	1.24	0.217

Regarding the additional co-variate regressors mentioned in Supplementary Information (magical ideation, teleological thinking, locus of control, unusual experiences, coincidence frequency), exploratory analyses indicated that the choice of which covariate to include in the regression models had no effect on the associations between IL-pat and religious belief. The effect of IL-pat on IB with schizotypy and

parents' belief as covariates (U.S.: $\beta = 0.17$, $P = 0.009$; Afghanistan: $\beta = 0.26$, $P = 0.001$), as reported in the manuscript, was not meaningfully different from the effect of IL-pat within alternative models using any individual covariate in place of schizotypy (U.S.: β s = $0.15 - 0.20$, P s = $.002 - .019$; Afghanistan: β s = $0.21 - 0.25$, P s = $0.003 - 0.007$) or with any additional covariate used in conjunction with schizotypy and parents' belief (U.S.: β s = $0.15 - 0.20$, P s = $.002 - .018$; Afghanistan: β s = $0.23 - 0.25$, P s = $0.002 - 0.003$). Similarly, IL-pat was as predictive in the primary Belief Change model (U.S. $\beta = 0.15$, $P = 0.036$; Afghanistan: $\beta = 0.17$, $P = 0.036$) as in all alternative models in which schizotypy was replaced with a different covariate (U.S.: β s = $0.15 - 0.16$, P s = $0.03 - 0.04$; Afghanistan: β s = $0.16 - 0.17$, P s = $0.04 - 0.056$). When each covariate was used in conjunction with schizotypy and parents' belief, IL-pat remained a significant predictor of Belief Change in all models (U.S.: β s = $0.15 - 0.16$, P s = $0.03 - 0.04$; Afghanistan: β s = 0.17 , P s = $0.04 - 0.046$), with one exception (magical ideation in the U.S.: $\beta = 0.13$, $P = 0.059$, which was highly collinear with schizotypy: $r = 0.64$).

These covariate measures, which were relevant to separate cross-cultural hypotheses concerning personality and cognitive characteristics in U.S. and Middle Eastern cultural contexts, also provided some redundancy in case some measures were not viable. That is, because we could not predict which measures would or would not be sufficiently understood in the Afghan sample after Dari translation, but we anticipated that some would not be successful, it was useful to include multiple measures that reflected similar constructs. For example, the Locus of Control (Levenson, 1973) and teleological bias survey (Kelemen et al., 2013) were not successful in the Afghan sample; e.g., survey items referenced situations that had no cultural relevance for the Afghan sample. We attempted to entirely re-write the teleological bias survey, using scenarios that were more likely to be meaningful for Afghans, but this effort did not yield a scale that was properly understood. By contrast, the schizotypy survey developed by Claridge and Broks (1984) required minimal re-writing (we excluded only one item), and was well understood by the Afghan sample (Cronbach's alpha = 0.83). This measure, which was particularly relevant to the hypotheses of the present study based on a body of literature describing associations with religious belief (Barnes & Gibson, 2013) and biases to perceive order (Brugger et al., 1993; Reed et al., 2008), was thus included in the main analyses both for theoretical reasons and because it was well understood in Afghanistan. We also sought to avoid including multiple estimates of related constructs in the same single regression models (to avoid theoretical redundancy and statistical collinearity).

Regarding the Belief Change variable, we were primarily interested in the amount of change (as described in the manuscript). We did not have a hypothesis concerning the slope of change, and we are still of the opinion that slope would be a less clearly interpretable outcome from this measure – for example, it is not clear what differences in components of slope would mean given equivalent amounts of change, or how differences in slope might be considered in relation to other variables of interest (e.g., IL-pat). The primary purpose for asking participants to report ages at relatively small intermediate intervals (3-years) from childhood to adulthood was to encourage more precise and thorough efforts at recollection of change throughout development to achieve more accurate reporting of amount of change from childhood to adulthood than can be achieved by a single-item measure (e.g., “Has your belief changed since childhood?”). A slope outcome measure would also rely more heavily on the accuracy of finer-grained

distinctions in memory (e.g., difference between age 6 and age 9) – we wanted to encourage as much thought as possible about these distinctions, but judged that the resulting recollection of total change would be more reliable and the individual fine-grained distinctions. There were also concerns about whether participants would anchor their responses around the ends of the scale (e.g., reporting levels of belief at intermediate ages to indicate regular progression between the youngest and current age). We thus employed the “difference score” approach reported in the manuscript to measure amount of change.

Nonetheless, to address the reviewer’s question, we have added a section to Supplementary Information (pages 8-10) describing exploratory analyses of the slope variable (the slope variable is also included in the csv file). These analyses indicated that the choice of which Belief Change variable to use does not meaningfully alter the results reported in the manuscript:

Belief Change was surveyed using a lab-developed change of belief measure, which consisted of 9-point Likert scales on which participants reported their own strength of belief in God starting at age 6 and then at three-year intervals up to age 24. Instructions stated that participants should not respond to items beyond their present age. Asking participants to report belief at three-year intervals was intended to encourage more precise and thorough consideration of responses than can be achieved by a single item measure (e.g., “Has your belief changed since childhood?”). This method to assess change in religious belief is consistent with prior work that has demonstrated the efficacy of assessing religiosity retrospectively¹⁸⁻²¹.

To quantify change in belief, the primary outcome of interest was a *difference score* in which strength of belief in God at age 6 (the youngest age reported) was subtracted from the strength of belief at the oldest reported age. Exploratory analysis was also conducted for a *slope of change score* in which a line of best fit (using the polyfit function in MATLAB) was calculated for each participant, using each age for which they reported a strength of belief as the X variables and the corresponding belief as the Y variables. Thus, these two metrics distinguish between *how much* belief changed since age 6 (i.e., difference score) and *the rate at which* this change occurred (i.e., slope of change score). We favored the difference score approach because 1) amount of change provided a more clearly interpretable outcome measure than slope of change – for example, it was not clear what differences in components of slope would mean given equivalent amounts of change, or how differences in slope might be considered in relation to other variables of interest (e.g., IL-pat); 2) We were wary of issues related to fine-grained accuracy of retrospective data (e.g., accurately distinguishing between belief at age 6 vs. age 9); and 3) We were also mindful of potential “anchoring”²² at the ends of the scale based on strong recollections of belief in early childhood and current age (e.g., reporting levels of belief at intermediate ages to indicate regular progression between the youngest and current age).

Exploratory analyses indicated that difference and slope scores were strongly associated with one another in both samples ($r > 0.95$, $P < 0.0001$), thus the two Belief Change outcomes were likely to reflect subtle differences in estimation rather than conceptually distinct psychological phenomena. Further, a series of post-hoc tests (chi-square) following simultaneous linear regression models in which both Belief Change scores were regressed on IL-pat, schizotypal thinking, and parents' strength of belief in both the U.S. Afghan samples (using the seemingly unrelated estimate command in STATA) yielded no significant differences in the effect sizes of IL-pat on Belief Change (U.S.: $\chi^2 = 0.26$, $P = 0.61$; Afghanistan: $\chi^2 = 0.06$, $P = 0.80$), and IL-pat was associated with Belief Change in all four models (all P s $\leq .05$). Thus, the strength of the coefficients for IL-pat as a predictor of Belief Change did not differ meaningfully for the difference score vs. the slope score. Taken together, these results suggest that, while the effect size of IL-pat on change in belief from childhood to adulthood is relatively modest, this effect is robust to differences in the Belief Change score used.

Point 3: The second issue is conceptual. The take-home message of the paper is that individuals who are good at implicitly detecting patterns in the environment may intuit that the world is orderly, and thus are more likely to believe in an interventionist god. The obvious gap in this reasoning is that an interventionist god is not the only – and arguably not the best – explanation for apparent order in the universe. The authors say (lines 78-79) that “Believers are more inclined to perceive events in the world as adhering to a purpose or design rather than as a series of random, unpredictable occurrences”, but ever since Darwin it has been easy to disavow cosmic “purpose or design” without having to argue that the world is “a series of random, unpredictable occurrences”. Indeed, no-one with a scientific worldview will accept that the world is random and unpredictable, and it seems likely to me that people who are good at implicitly detecting patterns in the environment may also be more likely to adopt a scientific worldview. The authors' data are unable to speak to this possibility. By analogy, it could be that those with greater implicit appreciation of rhythmic structure are more likely to enjoy death metal than those with a poorer rhythmic sensibility – but of course they would probably also be more likely to enjoy Mozart or Stravinsky. A headline about death metal would be quite misleading. In the same way, the finding that religious believers are better at apprehending environmental patterns than non-believers is potentially misleading, as scientists may also be better at apprehending environmental patterns than non-scientists.

Response: We thank the reviewer for this nuanced and very interesting suggestion. This point, which had not occurred to us previously, was a primary basis for two new data collection efforts that we have now completed. First, all U.S. participants who completed the full in-person battery were re-contacted and asked to complete the Belief in Science Scale (Farias et al., 2013). The Belief in Science scale consists of 10 questions intended to measure ideas about science, such as its value, reliability, and ability to provide an understanding of the world and human culture. In total, 65 participants responded to our request. As excerpted below, while IL-pat remained associated with UO, IB, and Belief

Change, and UO remained associated with IB and Belief Change, belief in science was unrelated to UO and weakly *negatively* correlated with IL-pat. We also conducted an online data collection from a new group of largely European participants (to expand on and replicate our U.S. and Afghan samples). The online sample also completed the Belief in Science scale, as well as the UO measure, and all 3 measures from which IB was derived. In this sample, we replicated the finding that UO was related to IB, and again found that UO was not associated with belief in science. The new data collection and analyses related to belief in science are now described in a new section of the Results on pages 18-19:

Lastly, we investigated whether associations similar to those observed for belief in an interventionist god (i.e., associations with intuitions of universal order and implicit pattern learning) might also be observed for belief in science, a construct that may plausibly relate to explanations of order in the universe. All participants in the U.S. re-contact and European samples completed the Belief in Science Scale⁹⁴. This scale consists of 10 questions intended to measure attitudes toward science, such as its value, reliability, and ability to provide an understanding of the world and human culture.

Consistent with prior research that has examined relationships between religious belief and scientific belief⁹⁴, zero-order correlations indicated that scientific belief was inversely associated with IB (U.S. re-contact sample: $r = -0.22$, $P = 0.08$; European sample: $r = -0.36$, $P = 0.0003$) and Belief Change (U.S. re-contact sample only: $r = -0.28$, $P = 0.02$). Despite this negative direct correlation, it remained possible that stronger intuitions of universal order and/or stronger IL-pat might be associated with stronger belief in science if, for example, a subset of participants who held strong intuitions of order in the universe were biased toward scientific interpretations of such order. However, results indicated this was not the case; belief in science was unrelated to UO (U.S. re-contact sample: $r = -0.19$, $P = 0.13$; European sample: $r = -0.11$, $P = 0.27$; Supplementary Table 15) and negatively correlated with IL-pat (U.S. re-contact sample: $r = -0.28$, $P = 0.02$; Supplementary Table 14).

Further interpretation of these findings is now included in the Discussion on page 25, including acknowledging the possibility of order-related beliefs – scientific or otherwise – that might be linked to UO and/or IL-pat but not accounted for in the present study. For example, because at least one item on the Belief in Science scale directly contrasts belief in science with religious belief, it is possible that the present study underestimates the extent to which both religion AND science might operate as ordering beliefs in some individuals.

While UO and IL-pat were consistently associated with IB across the various samples we studied, UO and IL-pat were not found to be associated with scores on the Belief in Science Scale⁹⁴. Thus, it is likely that intuitions of ambient order more closely relate to IB than to belief in science, at least within the (certainly non-exhaustive) scope of what is measured by the

presently employed survey of UO and Belief in Science Scale. More generally, the present findings do not rule out the possibility that nonreligious order-related beliefs, scientific or otherwise, could be related to UO or IL-pat. For example, because at least one item on the Belief in Science scale directly contrasts belief in science with religious belief, it is possible that the present study underestimated the extent to which religion and science might operate concurrently as ordering beliefs in some individuals. However, even if the relationships of IL-pat and UO to IB were found to operate in parallel to other relationships involving other ordering beliefs, this would not necessarily diminish the relationships to IB.

Point 4: In terms of revising the paper, one main point is that the nature of the perceptual task (the SRTT) was not clear (to me at least) from the authors' description. What is it that participants actually see on a particular trial? On a first reading, I assumed that (say) an x in a blue circle would appear in one of four quadrants; but on closer inspection the x refers to the response key used to indicate one of the four locations. So, what do the targets actually look like? It would be worth making this central component of the method as clear as possible.

Response. We thank the reviewer for this feedback, as it is imperative that our primary experimental task be presented in an understandable way. Participants viewed dots (not an "x" for example) at 4 locations onscreen. These dots correspond to 4 keys (left dot = "z", middle left = "x", middle right = ".", right dot = "/"), and participants were instructed that, when each dot appeared, they should press the corresponding key. To clarify the description of the task, we have amended text in 3 areas of the paper...

Results section (page 9):

Participants responded to the positions of filled circles (targets) that rapidly appeared at four positions onscreen by pressing a corresponding key

Figure 1 caption (page 10):

Fig. 1. SRTT. Participants indicated target locations corresponding to four keys (A). **For example, when the target appeared in the left most location, participants responded by pressing the “z” key.** Pattern blocks (B) consisted of 10-target repeating (5x) sequences, containing three first-order structures (orange bar) and two second-order structures (green circles). Random blocks (C) consisted of 50 non-repeating targets.

Method section (page 29):

Participants completed a modified Serial Reaction Time Task (SRTT), a widely-used measure considered to reflect ecologically valid IL-pat²¹. Participants were instructed to quickly and accurately indicate the position of the target circle as it appeared at each of four positions arranged horizontally onscreen (i.e., **left, center-left, center-right, right**; Fig. 1). **Each target position was designated a corresponding key on the keyboard, and participants were instructed on which key was associated with each target positions prior to beginning the task (moving from left to right, the mappings were “z”, “x”, “.”, “/”).** The version of the SRTT employed in the present study consisted of six blocks (3 pattern blocks; 3 random blocks). Each pattern block was composed of a distinct 10-target repeating sequence (repeated 5 times), consisting of three first-order structures (Fig. 1b, orange bars) and two second-order triplets (Fig. 1b, green circles). Random blocks did not include regular repetitions.

Point 5: Some of the literature review is a bit out of date. In particular, there is uncritical acceptance of certain concepts and findings from the literature that have been queried or that have failed to replicate. For example, the authors describe (lines 61-63) how an “automatic bias to detect agency in the environment is thought to yield over-attribution of intentional agency at the intuitive level... which supports explicit beliefs about supernatural, intelligent agents”, yet there is very little empirical evidence that this is the case. They cite van Elk’s (2013) study here, but van Elk found that religious beliefs were not related to agency attribution. Later work by van Elk’s group suggests, if anything, that humans under-attribute intentional agency (Maij et al., 2019). Another example: on lines 71-72 they say “Gervais and Norenzayan demonstrated that promoting analytic

thinking (rather than intuitive thinking) reduced strength of religious belief”. Multiple studies have failed to replicate this finding, and Will and Ara have themselves backed away from it (see Gervais, 2017; Gervais & Norenzayan, 2018).

Response: With thanks for these notes, we have now revised to provide a more updated account of the literature. New text now indicates that empirical support for the hypersensitive agency detection device does not afford a clear and consistent interpretation (pages 4-5):

With respect to religious belief, extant theoretical accounts^{1,2,9,10,13–16,32} posit that evolved neurocognitive processes contribute to “default intuitions” about the nature of environmental stimuli which direct individuals toward explicit beliefs. **For instance, neurobiology evolved for cooperative interactions among humans is implicated as a basis for intuitions concerning morality and fairness^{2,10,15}. These intuitions, in turn, are theorized to bias individuals toward religious beliefs in divine justice and watchful gods, which align with moral intuitions, and may insulate against challenges to intuitions about justice when commensurate retribution or compensation for a given act is not materially available³³. Other theoretical work has suggested a connection of evolutionarily-favored pathogen-avoidance mechanisms (e.g., evolved avoidance of individuals who are sick to minimize transmission of pathogens) to intuitions concerning cleanliness and religious beliefs concerning purity (e.g., belief in the healing effects of touching holy individuals or items, belief in unseen agents of spiritual corruption)^{16,34}. Intuitions of anthropomorphism, which appear to be biologically rooted in systems evolved to support face processing and social-cognition³⁵, may contribute to beliefs in watchful invisible agents^{36,37}. Relatedly, the automatic bias to detect agency in the environment is thought to yield over-attribution of intentional agency at the intuitive level (i.e., intuitions that non-agentic things have agency), which supports explicit beliefs about supernatural, intelligent agents (e.g., deities)^{1,13,14}. Empirical support for this theory, however, has been mixed; recent work has suggested that individuals may *under* attribute agency³⁸, and that believers in the paranormal – but not those who endorse a set of more traditional religious beliefs – display illusory agency detection^{39,40}.**

We have also added text to indicate the failed replication of the analytic prime employed by Gervais and Norenzayan (page 5):

Other literature has suggested a link between top-down, effortful analytic thinking – a propensity to critically examine and override prepotent “automatic”⁴¹ responding – and religious disbelief^{41,42}. Gervais and Norenzayan⁴³ reported that priming analytic thinking reduced strength of religious belief. However, replication attempts have not supported the efficacy of the so-called “analytic prime” employed by Gervais and Norenzayan^{44,45}.

Point 6: Lines 216-8: The authors note that explicit accuracy was not related to IB or Belief Change, but what about explicit belief that there was a pattern? Are people who are more likely to explicitly identify patterns (whether or not they're accurate about that) more likely to be believers? There's an emerging literature on how illusions of contingencies predict supernatural/paranormal belief (e.g., see Blanco, Barberia & Matute, 2015; Griffiths, Shehabi, Murphy & Le Pelley, 2018).

Response: We thank the reviewer for this suggestion. We now report analyses indicating that neither explicit pattern detection nor more frequent endorsements of patterns (regardless of whether or not patterns were actually present) was associated with IB or Belief Change in the U.S. or Afghan samples (excerpted below).

Extant research on illusions of contingencies have utilized tasks and theoretical models that are quite distinct from those in our study, so it is not clear that any argument can be made for or against the findings of prior studies on the basis of the present data. The distinctions primarily concern the kind of contingency being measured. In particular, extant studies have been interested in contingency defined as the causal influence of the participant's actions on observed outcomes (i.e., the agency of the participant in a causal context). This kind of contingency differs from the implicit pattern learning we studied because 1) our paradigm was not related to the participant's agency (participants learned pattern sequences that proceed without any apparent opportunity for the participant to influence them), and no instruction is given regarding agency, and 2) the implicit learning sequences in the present study were not devised to indicate any form of causality, nor were participants given any instruction to consider causality as in prior studies. Additionally, many aspects of the experimental designs, instructions, and stimulus modalities in these studies are quite different from those employed in the present study. For example, the "contingency" task employed by Blanco et al. (2015) involved a series of judgements about whether or not to act by administering medicine to patients, and "illusions of causality" were assessed based on the extent to which a participant incorrectly believed that their administration of a non-contingent medicine was an effective treatment. Thus, illusory contingency was based on the participants' behavior and decision-making (and their perception of their own agency), rather than how they perceive/respond to implicitly-learned stimuli. Additionally, the authors related performance on their contingency task to superstitious and paranormal beliefs, which is a distinct outcome from the IB measures used in the present study. Griffiths and colleagues (2018) also operationalizing "illusions of causality" based on participants' own actions. Participants were tasked with rating how much their actions influenced the appearance of a light on screen. Again, the researchers were measuring participants' belief in the effect of their own actions (causal agency), and again this study related illusions of causality to superstitious belief, rather than outcomes similar to IB.

Results (pages 13-14):

In order to further confirm that implicit pattern learning – rather than explicit awareness – was driving the association with religious belief, we performed zero-order correlations for explicit accuracy on the SRTT with IB and Belief Change.

Explicit accuracy was not related to IB (U.S.: $r = 0.07$, $P = 0.30$; Afghanistan: $r = 0.13$, $P = 0.14$) or Belief Change (U.S.: $r = 0.06$, $P = 0.42$; Afghanistan: $r = 0.06$, $P = 0.51$). We also tested the association of IB and Belief Change with illusory explicit pattern detection (i.e., reporting patterns for random blocks) and more frequent endorsements of patterns regardless of block type. IB was not correlated with illusory detection (U.S.: $r = 0.08$, $P = 0.23$; Afghanistan: $r = 0.15$; $P = 0.09$) or overall tendency to reported patterns (U.S.: $r = 0.04$, $P = 0.56$; Afghanistan: $r = 0.06$, $P = 0.51$). Similarly, we did not observe an association between Belief Change and illusory detection (U.S.: $r = 0.11$, $P = 0.11$; Afghanistan: $r = 0.05$, $P = 0.55$) or an overall bias towards reporting patterns (U.S.: $r = 0.09$, $P = 0.21$; Afghanistan: $r = 0.02$, $P = 0.86$). Finally, when these explicit awareness variables were included as additional covariate regressors (along with parents' belief and schizotypal thinking; Supplementary Tables 12, 13), IL-pat remained a significant predictor of both IB (all P s ≤ 0.009) and Belief Change (all P s ≤ 0.05).

Discussion (pages 25-27):

We did not observe an association between explicit judgements about sequence blocks of the SRTT and religious belief. We are not aware of any research that has described an association between accurate, conscious detection of patterns and religious belief, though some prior research has reported an increased tendency among believers in the paranormal to report some forms of illusory contingency^{106,107}. Erroneous reporting of patterns on the random blocks, as well as overall tendency to report patterns across both block types (random and pattern), was not correlated with IB or Belief Change in the present study. When these variables were included as additional covariate regressors in the above-reported models, the effects of IL-pat on both IB and Belief Change were not altered. Because IL-pat was also uncorrelated with explicit accuracy in reporting patterns, the results consistently supported the interpretation that implicit (rather than explicit) learning was driving the observed effects.

It is not evident, however, that the present results bear substantively on prior findings of explicit contingency reporting. Prior research investigating reporting of illusory contingency among believers in the supernatural was based on tasks and theoretical models that are quite distinct from those in the present study. In particular, prior studies investigated contingency in the form of causal influence of a participant's actions on observed outcomes (i.e., the agency of the participant in a causal context). This kind of contingency differs from the implicit pattern learning we studied because 1) our paradigm was not related to the participant's agency (participants learned pattern sequences that proceeded without any apparent opportunity for the participant to influence them, and no instruction was given regarding agency), and 2) the implicit learning sequences in the present study were not devised to indicate any form of causality, nor were participants given any instruction to consider causality as

in prior studies. Additionally, many aspects of the experimental designs, instructions, and stimulus modalities in these studies are quite different from those employed in the present study. For example, the “contingency” task employed by Blanco et al. (2015) involved a series of judgements about whether or not to act by administering medicine to patients, and “illusions of causality” were assessed based on the extent to which a participant incorrectly believed their administration of a non-contingent medicine was an effective treatment. Thus, illusory contingency was based on the participants’ behavior and decision-making (and their perception of their own agency), rather than how they perceive/respond to implicitly-learned stimuli. Additionally, the authors related performance on their contingency task to superstitious and paranormal beliefs, which is a distinct outcome from the IB measures used in the present study.

Point 7: Lines 434-6: The authors state that “because Q1 and Q2 were presented before Q3 and Q4, any implied reference to a deity in Q3 or Q4 would not have influenced participants’ interpretation of Q1 or Q2.” But that presumably depends on how the items were presented – was it possible for participants to look ahead to subsequent items?

Response: Thank you for this note. The questionnaire contained all 4 items on a single screen, with questions presented in order Q1 – Q4 from top to bottom. It is possible that some participants could have skipped ahead to the third or fourth question before answering the first two. However, there does not seem to be a particular reason to think they did so. Further, participants in the new data collection we have now conducted in the European sample were only presented with questions 1 and 2, and the association between UO and IB in this sample was similar to what we observed in the initial sample.

Point 8: Lines 442-5: “Afghan participants showed lower variance... and stronger correlations between UO items... suggesting that they did not detect the subtle differences between items.” But what was the correlation between UO and IB for the Afghan participants? Isn’t that the critical feature here?

Response. The reviewer is correct that, theoretically, the critical feature would be how UO relates to IB (and the SRTT). However, the issue was that the scale was not properly understandable for the Afghan participants. Thus, the results using this variable – including its association with IB – cannot be meaningfully interpreted. This is described in the Methods on page 33:

We anticipated that UO was unlikely to be meaningful for the Afghan sample because of a lack of culture- and language-specific interpretability of the UO prompts – much of the difficulty arose in the attempt to make the prompts “secular” (i.e., **to use wording/framing that referred to an order in the universe without referring to a deity or any religious belief**). These attempts ultimately led to prompts that were not interpretable in the intended ways, as

determined by Z.W. in consultation with Dari-speaking experimenters. Therefore, scores on this measure were not analyzed in this sample.

The item correlations noted by the reviewer were conducted as a confirmatory analysis for the UO variable since this was the one variable used in the U.S. but not Afghanistan. However, this analysis was not the basis for exclusion (again, exclusion was determined by the fact that the measure was not understood properly). Based on the reviewer's comment, we have now judged that reporting these confirmatory item correlations might be more confusing than helpful. We have thus removed them from the current version of the revised Methods, but would be happy to include them again if the reviewer prefers.

Point 9: There are a number of typos in the manuscript. Here are some: "Judiasm" (line 81), "evalute" (line 129), "across religious." (line 129), "substancially" (line 131), "individual difference(s)" (line 150); "gods the U.S." (line 256) "religious individual(s)" (line 278); "Definitely a Not a Pattern" (line 379); "extend(ed) exposure" (line 385); "lead" should be "led" (line 440); "measureable" (line 456) etc. etc.

Response: We thank the reviewer. We have performed a careful proofreading of the manuscript and corrected typos that were overlooked in the original manuscript.

Reviewer #3

Remarks to the Author:

The authors investigate if individual differences in implicit pattern learning predicts both belief in an intervening god, as well as changes in theological beliefs over time. The authors use a Serial Reaction Time Task (SRTT) to represent implicit pattern learning, a composite self-report measure of belief in an ordering and intervening god, and a retrospective measure of belief change since childhood. They (a) show support for both hypothesis, in samples from the U.S. and Afghanistan; as well as, (b) beliefs in a (non-theologically) ordered universe mediates the observed relationship within the US sample. This is an interesting and well-executed contribution to our understanding of the cognitive underpinnings of religious belief. As the authors mention, there is much work on the role of intuitive information processing as a driving force behind the acquisition, maintenance, and rejection of religious belief. Two aspects of this work jump out as clear contributions:

First, much of the previous work has used simple measures of "intuition" – e.g. the Cognitive Reflection Test (CRT) or variations of heuristics and biases batteries. The approach outlined in this paper is operating at a much lower level and is less subject to deflationary counter explanations.

Second, as were much work is purported to be about "intuition", the vast majority report negative relationships between belief and "reflection", e.g. higher scores on the CRT (with one exception: Shenhav, Rand, & Greene, 2012). This work, however, does a nice job of selecting a task that explicitly focuses on intuitive information processing as the mechanism driving the relationship (though see point 1 below).

Overall, I like this paper and can see suggesting its acceptance upon the completion of several changes and clarifications.

Response: We are grateful for the reviewer's attention and valuable suggestions, and we appreciate the encouraging assessment of the manuscript. Multiple revisions and additions, each of which has improved the manuscript, were guided by the reviewer's thoughts.

Point 1: This paper frames the argument as- evidence shows religious beliefs are underpinned by intuitive factors and thus, variation in intuition-based information processing can be thought of as possibly driving variation in religious belief. However, some work in this area argues it's reflection that's driving this relationship (Pennycook, et al, 2012). This work notes how a non-trivial amount of the Cognitive Science of Religion has cited violations of intuitions to be an important component of many, if not most, religious content such as narrative (Boyer, 2007). Reflection has been described as the process of overriding intuitive responses, intuitive responses that all individuals experience, but only a subset over-ride. Thus, it's hard to know if variation in individual intuition, vs variation in individual reflection, is driving this relationship with religious belief.

Noting my point above, much of the previous literature reports variation in "reflection" and not "intuition" as driving the relationship with belief. In the paper, the authors note how attempts at identifying patterns on the SRTT can result in lower scores. One could imagine a case where there is little variation in "intuition" based performance; however, notable variation in attempts to detect a pattern, driven by individual difference in attempts to critically evaluate the information, driven by individual differences in "reflection". This may be a subtle point, but important for understanding the underlying mechanism. What would the authors say about this hypothetical case? Is there some way of understanding performance on the SRTT that would address this question?

Response: We thank the reviewer for this thoughtful question. We agree that this is an interesting possibility, and indeed that the reviewer proposes a theoretically plausible mechanism that could be at work related to "reflection" as measured in the CRT. As we understand it, the most central question for evaluating this possibility is whether there is a relationship between reflection as measured by the CRT (i.e., the engagement of effortful analytic/reflective thinking to override default prepotent responding, at least in the context of calculation-based problem solving) and implicit learning (which the reviewer refers to as "intuition-based information processing"). A second question is whether the particular attributes of the SRTT paradigm employed in this study are resistant to top-down influence (whether as help or hindrance).

To address the first question as directly as possible, we conducted additional data collection. All U.S. participants who had completed the full study battery were re-contacted and asked to complete the CRT. The data and results are described in the below excerpts from the revised manuscript. To summarize, these new data were helpful in assessing for the first time the relationship of CRT to implicit pattern learning, and specifically in determining that the kind of analytic vs. "intuitive" thinking measured by

the CRT was not related to IL-pat, and did not account for the main results we observed. As noted in the excerpted text, IL-pat remained a significant predictor of IB, Belief Change, and UO even when controlling for CRT performance.

The CRT is now described in the Methods on page 34:

Cognitive Reflection Test. The CRT⁹⁰ consists of 3 calculation-based questions devised to elicit prepotent – but incorrect – responses, referred to as “intuitive.” Overriding these prepotent responses in favor of more effortful calculation, which is more likely to yield the correct solutions, is considered analytic thinking (also referred to as reflection). The number of correct and intuitive responses are summed for each participant. The CRT was completed by the U.S. re-contact and online European samples. ***Belief in Science Scale.*** The Belief in Science Scale⁹⁴, which consists of 10 questions intended to measure attitudes toward science, such as its value, reliability, and ability to provide an understanding of the world and human culture, was also completed by the U.S. re-contact and European samples.

CRT data and analyses in the re-contacted U.S. sample are now described in the Results on pages 16-18:

The above results supported hypothesized links of intuition to IL-pat and belief – these hypotheses specifically concerned intuitions of order (i.e., UO). Subsequently, we explored a secondary question concerning the Cognitive Reflection Test (CRT⁹⁰), a measure of calculation-based problem solving devised to distinguish responding based on a more effortful and analytic approach from responding based on a less effortful and analytic (and more error-prone) approach; the latter approach has been referred to as “intuitive” thinking. Investigation of the CRT was motivated by prior research that has suggested a connection between non-belief and analytic problem solving on the CRT^{42,44,45,91–93}. It was thus of interest to determine whether IL-pat was related to CRT performance, and whether the associations of IL-pat to belief and to intuitions of order (as measured by UO) were independent of the kind of analytic vs. “intuitive” problem solving measured by the CRT. All U.S. participants who completed the full study battery were re-contacted, and respondents (“U.S. re-contact” sample; N = 65) completed the CRT. The CRT consists of three arithmetic problems. For each problem, there is a response that is considered to be prepotent (i.e., the first answer that comes to mind), which is referred to as the “intuitive” response, whereas calculating the correct answer is posited to require greater analytic, top-down effort. Correct (0-3) and intuitive (0-3) responses are summed for each participant. Incorrect responses that are not the intuitive response are ignored in the calculation of correct and intuitive scores. Following Frederick (2005), the number of correct responses was interpreted as an estimate of analytic problem-solving, or the extent to which individuals are able to override prepotent (incorrect) responses.

Consistent with the associations observed in the full U.S. sample, IL-pat was significantly correlated with IB ($r = .46, P = 0.0001$), Belief Change ($r = .39, P = 0.002$), and UO ($r = .39, P = 0.001$; Supplementary Table 14). Also consistent with results obtained from the full sample, there was no association between IL-pat and accuracy of explicit pattern judgements ($r = -0.18, P = 0.16$), indicating that learning in this sub-sample was implicit. Results for the CRT indicated that analytic (i.e., correct) and intuitive (incorrect) responses were strongly negatively correlated with each other (both samples $r < -0.85, P < 0.0001$), as in prior research⁹⁰. In the analyses of greatest interest, we found that IL-pat was not associated with analytic responding ($r = -0.07, P = 0.56$) or intuitive responding ($r = -0.03, P = 0.84$; Supplementary Table 14). Linear regression models in which CRT performance was added as an additional covariate regressor (in addition to schizotypal thinking and parents' belief) indicated that IL-pat remained a significant predictor of IB ($b = 3.22, \beta = 0.43, SE = 0.81, P < 0.001$), Belief Change ($b = 9.88, \beta = 0.40, SE = 2.96, P = 0.001$), and UO ($b = 11.64, \beta = .38, SE = 3.61, P = 0.002$; Supplementary Table 16). Together these results indicate that the relationship of IL-pat to IB, Belief Change, and UO was meaningfully distinct from CRT-based problem solving approach. That is, it is unlikely that individual differences in the tendency to effortfully apply analytic thinking to override prepotent responding (at least in the context of CRT calculation problems) are responsible for variation in IL-pat or the association between IL-pat and belief.

Supplementary Table 16. Linear regression results in U.S. re-contact sample with analytic and intuitive thinking during the CRT as additional covariate regressor

	Analytical Thinking					Intuitive Thinking				
	β	B	SE	t	P	β	B	SE	t	P
Interventionist Belief										
IL-Pat	0.43	3.22	0.81	3.98	<.001	0.43	3.26	0.80	4.06	<.001
Schizotypy	0.01	0.06	0.54	0.12	0.906	0.02	0.09	0.53	0.16	0.872
Parents' Belief	0.32	0.13	0.04	2.96	0.004	0.31	0.13	0.04	2.83	0.006
CRT Performance	-0.03	-0.02	0.09	-0.24	0.813	0.10	0.09	0.10	0.90	0.373
Belief Change										
IL-Pat	0.40	9.88	2.96	3.34	0.001	0.39	9.67	2.95	3.28	0.002
Schizotypy	0.02	0.36	1.96	0.18	0.855	0.02	0.28	1.96	0.14	0.886
Parents' Belief	-0.09	-0.12	0.16	-0.74	0.464	-0.08	-0.11	0.16	-0.67	0.503
CRT Performance	0.08	0.22	0.34	0.63	0.531	-0.10	-0.31	0.37	-0.84	0.407
Universal Order										
IL-Pat	0.38	11.64	3.61	3.23	0.002	0.37	11.44	3.61	3.16	0.002
Schizotypy	0.12	2.53	2.34	1.06	0.292	0.12	2.51	2.40	1.05	0.299
Parents' Belief	0.16	0.27	0.20	1.37	0.177	0.15	0.25	0.20	1.27	0.209
CRT Performance	0.09	0.32	0.42	0.78	0.439	-0.02	-0.08	0.45	-0.17	0.868

Interpretation of these results has also been added in the Discussion on pages 24-25:

The present research also explored the extent to which IL-pat is related to individual differences in analytic problem solving (measured by the CRT). An influence of analytic thinking on IL-pat might be posited whereby analytic thinking is associated with increased top-down searching for patterns, which could hinder IL-pat on the SRTT in *nonreligious* participants, given previous associations of analytic problem-solving on the CRT with lower levels of religious belief^{41,42}. However, the lack of association between CRT and IL-pat suggests no such influence in the present study, perhaps owing in part to the attributes of the SRTT paradigm devised to minimize top-down influence (described above). Our findings further indicated that the associations of IL-pat to IB, Belief Change, and UO were independent of CRT performance. To our knowledge, this is the first research to empirically test whether performance on the CRT is associated with implicit pattern learning. Prior research that has suggested an association of implicit learning to thinking style has assessed intuitive thinking via self-report⁵⁵⁻⁵⁷. For example, Woolhouse and Bayne (2000) measured intuition using the self-report sensing-intuition scale of the Myers-Briggs Type Indicator. These previous approaches are more similar to the self-report measure of universal order intuitions in the present study than to the CRT. In contrast, assessment of thinking style in the context of arithmetic calculation on the CRT is most likely to be related to differences in so-called “system 1” (automatic) and “system 2” (deliberate) problem-solving¹⁷, which appears conceptually distinct from the presence of broad intuitions of order that we and others^{20-23,54} have suggested may arise from IL-pat, and that the present work investigated as a potential mediator of the effect of IL-pat on belief. Arithmetic calculation is also likely to reflect common math/calculation-specific influences on elements of thinking style related to motivation, avoidance and cognitive function¹⁰³⁻¹⁰⁵.

Regarding the second question, the manuscript highlights task characteristics of the employed version of the SRTT intended to decrease the likelihood that differences in top-down searching (due to analytic thinking or otherwise) influenced task performance. These task characteristics, and their relevance for interpreting the data with respect to potential top-down influence, are addressed at several points in the revised manuscript.

In the Introduction on page 8:

Multiple experimental paradigms have been developed to investigate IL-pat. Perhaps the most widely used measure is the Serial Reaction Time Task (SRTT)^{77,78}, which prior work suggests is reflective of environmental IL-pat²¹, and which has recently been shown to load heavily on a broad implicit learning ability factor⁶⁵. Notably, extant evidence indicates that IL-pat during the SRTT is not improved – and may actually be impaired – by top-down influences, including instructions to look for patterns^{61,69,79,80} (see Methods). In addition, modifications of the SRTT have been devised to limit opportunities for explicit awareness of patterns that could be a precondition for top-down influence^{69,81}. The SRTT thus

presents a good experimental measure to investigate putative bottom-up link of IL-pat to belief: stronger IL-pat might contribute to stronger belief, but a belief-related top-down bias to seek patterns is unlikely to strengthen IL-pat, thus, any positive association observed between IL-pat and belief is most likely to be bottom-up.

In the Methods on page 30:

In addition, the opportunity to explicitly recall/rehearse target sequences in an effort to explicitly identify patterns for patterns was mitigated by removing the response-stimulus-interval (RSI) between key-press and the appearance of the next target⁸¹. During no-RSI versions of the SRTT, the next target appears immediately after the correct response is made; participants cannot use time between target appearances (because the next target appears immediately) to consciously attempt to identify a pattern structure.

And in the Discussion on pages 20-21:

The results of the current study are consistent with a bottom-up pathway of effect by which stronger implicit learning of patterns/order leads to belief in an intervening/ordering god. While top-down pathways are also conceivable whereby stronger belief in an intervening deity leads individuals to more effectively learn implicit patterns, and/or whereby religious individuals may be more likely to search for patterns because of an explicit belief in the ordering influence of a deity, a number of empirically-based indicators make such explanations less likely in this case. First, prior research has demonstrated that explicit searching does not lead to faster responding during the SRTT, and may actually hinder performance^{61,69,79}. For example, Fletcher and colleagues⁷⁹ showed that explicitly directing participants to search for a pattern impeded implicit learning on the SRTT (i.e., slower RT during the task), putatively because this explicit search disrupts elements of task performance that facilitate IL-pat. Thus, even if interventionist belief promotes explicit searching for patterns, it is unlikely that this would have led to improved performance on the SRTT in the present study.

A bottom-up interpretation of the present data is further bolstered by modifications to the SRTT paradigm that were incorporated into the present study. Namely, the sequence advanced (i.e., the next target appeared) immediately after the participant provided a correct response. This so-called no-response-time-interval (no-RSI) version of the SRTT has been demonstrated by Destrebecqz and Cleeremans and others^{69,81} to yield sequence learning without explicit knowledge acquisition. Removing time between target presentations is intended to ensure that learners are consistently engaged in the task and do not have the opportunity to consciously attempt to recall/rehearse the target sequence in an effort to identify patterns while waiting for the next target to appear. Consistent with prior work, we did not observe a significant correlation between sequence learning (i.e., rate of change on pattern sequences relative to random ones) and explicit awareness,

indicating that learning was implicit. **The lack of association between CRT performance and implicit learning on the SRTT may further suggest that differences in IL-pat were unlikely to be driven by a bias towards effortful top-down thinking.**

Nonetheless, the directionality of the hypothesized effect pathway is subsequently qualified within the Discussion to acknowledge that the proposed pathway is unlikely to provide a complete accounting of the influences on religious belief formation, and that top-down influences on bottom-up processing are still entirely possible (pages 22-23):

While the bottom-up directionality of effect we have outlined is consistent with prior work on the neurocognitive bases of religious belief^{1,2,9-12,16}, it is not possible – and is not the intent of the present study – to conclusively rule out any degree of top-down influence on the relationship between IL-pat and religious belief. Indeed, a complete rendering of the cognitive and environmental influences on a phenotype as complex as religious belief very likely includes a multiplex of interconnected “loop” architectures between bottom-up and top-down processes. Examples of top-down influences on automatic perceptual information processing are abundant^{19,23,98}. One pertinent example is research showing that induced “meaning threats” devised to challenge participants’ existential sense of meaning were associated with better implicit learning in an artificial grammar task⁹⁹. Religious belief is associated with a desire for meaning^{4,11,46}, and it is conceivable that more religious participants might tonically feel greater threat to meaning, which may have an influence on at least some forms of IL-pat. It should be noted that the improvement in grammar learning in this prior work was linked to an increased explicit motivation to find grammatical letter strings in the grammar learning task, whereas extant evidence indicates such explicit motivation does not improve learning on the SRTT^{61,69,79}. More directly to the point, even if it is the case that some amount of top-down influence has bearing on implicit learning, this would not necessarily be inconsistent with bottom-up influences of implicit learning on religious belief. While the present study focused on individual differences in a strongly bottom-up IL-pat paradigm to explore the thus far untested relationship of bottom-up IL-pat to belief, the intent of the present research is not to suggest that all bottom-up influences on belief, and perhaps not even the IL-pat studied here, operate entirely independently of top-down processes.

Point 2: Much of the previous literature has looked at overall belief in god as a main outcome of interest. This project’s focus, looking specifically at an ordering/ intervening god, is reasonable given the way the authors frame it. That said, it seems they do in fact have a measure of current level of belief (how the “change” variable is created). It would be helpful to see how IL-pat scores correlate with this outcome, given the previous literature. If for whatever reason, this relationship is less robust, some explanation would be helpful for why. If this was the case, one possibility could be that a subset of the

participants believes strongly in a god, but a god that is less causally relevant or present in their lives (e.g., some kind of deists). Though this would be surprising.

Response: We greatly appreciate this question. To summarize the lengthy response that follows, exploratory analyses indicated that the reviewer’s insight was quite correct – a subset of participants who reported believing in a god did not hold strong interventionist belief. The reviewer’s comment prompted us to investigate a variable we refer to in the revised Supplementary Information as Existence Belief. This variable is closest to the kind of single-item measurement that the reviewer refers to (because it was not obtained within the context of a survey on belief change). We had not given close consideration to this variable previously because belief (and especially our target construct of Interventionist Belief) could be measured much more meaningfully with the multiple measures combined for the IB outcome than by a single item measure. This item (initially conceived as part of a “Belief – Doubt” variable as described below) was primarily included in the test battery in relation to other research questions not directly concerning Interventionist Belief. Additionally, we anticipated that the Belief – Doubt measure was unlikely to be viable in Dari translation among Afghan Muslim participants, and this was the case. No acceptable translation for Muslim Dari speakers was found to convey the intended sense of belief (i.e., specifically relative to doubt) with respect to the existence of God, and experimenters reported that participants found questions concerning uncertainty that God exists to be insulting and/or “dangerous.” By contrast, questions concerning the level of interventionist belief were more amenable to Afghan data collection because they did not involve an implication of non-existence. Thus, even if it had been theoretically desirable, it would not have been possible to include this measure in the models involving equivalent variables in the two main samples.

Nonetheless, supplementary analyses have now been conducted to investigate the relationship of Existence Belief to other key variables in the U.S. sample. These analyses are excerpted below from Supplementary Information. In investigating this variable for the revised manuscript, we found that it was omitted from the list of additional variables in the original Supplementary Information. This prompted a close re-checking to determine whether this was the only omitted variable. One other variable, which was also a single-item measure superseded by a richer assessment, was also found. Specifically, the new Supplementary Information now lists a single-item measure that asked whether participants changed in belief in God since childhood. This measure, which was obviated by the richer Belief Change questionnaire discussed throughout the manuscript, was included largely because we were unsure whether the more complex Belief Change measure would be viable, especially in Dari among Afghan participants.

Exploring the Existence Belief variable has led to some enlightening results, very much in line with the reviewers’ insight that some individuals believe in a god without believing in the interventionist/ordering influence of a god. These analyses have helped to enrich our understanding of the particular relationship of implicit learning to interventionist belief in our sample. In particular, these analyses have served to strengthen and focus the evidence that implicit learning was specifically related to interventionist belief by distinguishing it from non-interventionist belief. The Existence Belief measure and relevant analyses are now described in Supplementary Information on pages 10-13:

Participants responded to two questions intended to measure the difference between belief vs. doubt in the existence of a God. Responses were made on a 9-point Likert scale:

- Question 1: “Rate the strength of your current belief in God” (1= do not believe God is real at all, 9 = absolutely certain God is real)
- Question 2: “Rate the strength of your doubt, if any, that God is real” (1= no doubt at all, 9 = maximum doubt).

We initially planned to score this “Belief-Doubt” measure by subtracting Item 2 from Item 1, to obtain a potentially more nuanced assessment of belief in the existence of God than what is obtainable from a single item. However, responses to these two items mirrored each other to such a strong extent ($r = -.87$, $P < 0.0001$) that this measure was essentially reducible to a single item. Further, because subtracting doubt from belief could yield identical scores for individuals with different strength of belief that God exists (e.g., 7 for Question 1 and 5 for Question 2 vs. 3 for Question 1 and 1 for Question 2), we elected to focus only on Question 1 to estimate participants’ “Existence Belief” for the analyses described below.

We anticipated a strong, positive correlation between Existence Belief and IB, which was a much richer variable and much more directly related to our primary hypothesis concerning implicit pattern learning and belief in the ordering influence of God, and we anticipated that any association between IL-pat and Existence Belief would be a largely redundant reflection of the association between IL-pat and IB (see below). Additionally, the Existence Belief measure was determined to be non-viable in the Afghan sample. No acceptable translation for Muslim Dari speakers was found to convey the intended sense of belief (i.e., specifically relative to doubt) with respect to the existence of God, and experimenters reported that participants found questions concerning uncertainty that God exists to be insulting and/or “dangerous.” By contrast, questions concerning the level of interventionist belief were possible because they did not involve an implication of non-existence. Thus, even if it had been theoretically desirable, it would not have been possible to include Existence Belief in the models involving equivalent variables in the two main samples. Nonetheless, supplementary analyses were conducted to investigate the relationship of Existence Belief to other key variables in the U.S. sample only.

Results

Existence belief and IB. As expected, Existence Belief was positively correlated with each of the three individual measures of interventionist belief (all $r_s > 0.47$, $P_s < 0.0001$) and with the Interventionist Belief component score (IB; $r = 0.66$, $P < 0.0001$). In order to quantify differences between Existence Belief and IB, we rescaled both variables such that scores for both variables would range from -1 to 1. Because belief that God exists is presumably a prerequisite for belief in God as an intervening influence, average IB would not be expected to exceed Existence Belief (i.e., those reporting belief in an intervening God would generally be expected to believe that God exists, but the reverse might not be true). Paired t-tests using the rescaled belief measures supported this prediction;

participants held significantly lower IB ($M = -0.19$, $SE = 0.04$) than Existence Belief ($M = 0.06$, $SE = 0.05$; $t(198) = -7.07$, $P < 0.0001$). That is, participants reported higher belief that God exists relative to their belief in God as an intervening influence. Further, we identified a marked difference in the distribution of responses for Existence Belief and IB (Supplementary Fig. 3; one-sample Kolmogorov–Smirnov: $P < 0.001$). In particular, we observed a large clustering of participants who reported little to no belief in an intervening God (i.e., a positive skew for IB). However, the majority of participants *did* endorse an Existence Belief, and this variable was characterized by a fairly bi-modal distribution. These differences suggest that a subset of participants believed in a god, but one that is relatively non-interventionist.

Existence Belief and IL-pat. Exploratory analyses to examine whether Existence Belief was associated with IL-pat revealed no correlation ($r = 0.05$, $P = 0.50$). Thus, while IB and Existence Belief are strongly correlated, only the former was associated with IL-pat. However, because Existence Belief was closely related to IB in some participants (but not in others), we next asked whether IL-pat was more associated with Existence Belief in individuals for whom IB more closely approximated Existence Belief and less associated with Existence Belief in those for whom Existence Belief was more divergent from IB (putatively explaining the discrepant relationships of IL-pat to IB vs. Existence Belief). In order to do this, we first calculated a “Belief Difference” score by subtracting rescaled Existence Belief from rescaled IB (a score of 0 indicated equivalent Existence Belief and IB; positive values indicated stronger IB relative to Existence Belief). To explore whether IL-pat differentially predicted Existence Belief based on how closely Existence Belief related to IB, we interacted the absolute value of Belief Difference with IL-pat to predict Existence Belief. We used the absolute value of Belief Difference because our question concerned similarity (i.e., how close to 0 is the difference between the two belief variables) rather than relative magnitude. This model revealed a significant Belief Difference X IL-pat interaction, such that the association between IL-pat and Existence Belief was stronger for those participants whose IB more closely approximated Existence Belief ($\beta = -0.20$, $P = 0.002$). In other words, the strength of the relationship between IL-pat and Existence Belief was related to the similarity between IB and Existence Belief. Taken together, these post-hoc analyses demonstrate that, consistent with our hypothesis concerning IL-pat as a contributor to IB, IL-pat was more predictive of Existence Belief in participants for whom the level of existence belief was closely tied to the level of IB, and less predictive in those whose belief in God appeared to be less tied to belief in the intervening influence of God (i.e., those who report higher levels of Existence Belief relative to IB). One plausible interpretation is that IL is more strongly related to Existence Belief in a subset of individuals for whom belief that God exists is largely focused and/or dependent on belief in the interventionist influence of God, and less related to Existence Belief in others for whom belief in the existence of God depends more on other factors.

Supplementary Figure 3. Distribution of Existence Belief and IB. IB was characterized by a positive skew whereas Existence Belief was more bimodal (distribution significantly different at $P < 0.001$). On average, participants reported higher Existence Belief than IB ($P < 0.0001$).

Additional analyses to further characterize the associations between IL-pat and Existence Belief also identified an instructive relationship between Existence Belief and Belief Change. Continuing from the above text, this is described in Supplementary Information on pages 13-15:

The above analyses indicated that the extent to which IL-pat was predictive of Existence Belief varied by how closely Existence Belief was related to IB (i.e., Belief Difference). A second regression model in which the absolute value of Belief Difference was interacted with IL-pat and regressed on Belief Change revealed results similar to those described above. Individuals with more similar Existence Belief and IB showed a stronger association between IL-pat and Belief Change (US: $\beta = -0.16$, $P = 0.024$), indicating that the effect of IL-pat on Belief Change may also be largely dependent on the extent to which belief in God reflects belief in the interventionist influence of God.

In contrast to Existence Belief, however, IL-pat *was* predictive of Belief Change in the full sample. Additional analyses were performed to elucidate the reason for this distinction. First, although Existence Belief and Belief Change were positively correlated ($r = 0.54$, $P < 0.0001$), a paired t-test indicated a significant difference between rescaled Existence Belief (scores from -1 to 1; $M = 0.06$, $SE = 0.05$) and rescaled Belief Change ($M = -0.05$, $SE = 0.03$; $t(196) = -2.47$, $P = 0.014$).

Additionally, the association between Belief Change and Existence Belief was not uniform across the full range of Existence Belief (Supplementary Fig. 4), reflecting the fact that the Belief Change measure was sensitive to the trajectory of belief (i.e., increase, decrease, amount of change) whereas Existence Belief was not. Participants who decreased in strength of belief since childhood were overrepresented at the low end (i.e., 1-3) of the Existence Belief scale (because it was not possible to increase in belief to reach the lowest levels of Existence Belief). Similarly, participants who increased in strength of belief since

childhood were overrepresented at the high end (i.e., 7-9) of the Existence Belief scale (because it was not possible to decrease in belief to reach the highest levels of Existence Belief). However, the middle Existence Belief group (i.e., 4-6) contained a wide range of change in belief values (both positive and negative change).

To explore this further, we divided the sample into 3 groups based on Existence Belief (Low = 1-3, Moderate = 4-6, High = 7-9). A one-way ANOVA revealed significant differences in Belief Change (z-scored) between these groups ($F(196) = 46.59, P < 0.0001$), with only the High belief group showing above-average Belief Change. We also observed differences in the relationship between Existence Belief and Belief Change. Most notably, in the Low belief group ($N = 69$), Existence Belief and Belief Change were anticorrelated ($r = -0.26, P = 0.03$), and in the Moderate belief group ($N = 47$) there was no association between Existence Belief and Belief Change ($r = -0.08, P = 0.60$). Indeed, in the low belief group, participants with the highest Existence Belief (i.e., 3; average Belief Change = $-0.96, SE = .14$) actually *decreased* in belief significantly *more* since childhood relative to those with the lowest Existence Belief (i.e., 1; average Belief Change = $-0.39, SE = 0.21; t(46) = 2.25, P = 0.03$), and in the Moderate belief group, participants with the lowest Existence Belief (i.e., 4; average Belief Change = $0.25, SE = 0.37$) increased in belief nominally more than participants with the highest Existence Belief (6; average Belief Change = $-0.09, SE = 0.17$).

The data thus indicate against a superficial prediction that Belief Change should straightforwardly track with Existence Belief (i.e., that those who increase more in belief should have higher Existence belief whereas those who decrease more should have lower Existence Belief). In fact, it is evident that this was frequently not the case (e.g., individuals who decreased in belief frequently had higher Existence Belief than other individuals who increased in belief). Inferences based on these statistical tests and nominal comparisons should be treated with some caution due to small and/or imbalanced sample sizes, but the data suggest that Belief Change is distinct from – and at times inversely related to – Existence Belief. The overall pattern of results (non-linear relationships, negative values for Belief Change versus positive values for Existence Belief) indicate that sensitivity to the trajectory of belief, which was an attribute of the Belief Change variable but not the Existence Belief variable, is an empirically meaningful basis for differentiating these variables in relation to each other and in relation to IL-pat.

Supplementary Figure 4. Belief Change varies in its association with Existence Belief. For those in the low belief group (green box), Belief Change was negatively associated with Existence Belief. There is no association at moderate belief (pink box) and a positive relationship for those with high belief (grey box).

Point 3: Though I understand mediation analysis is a common statistical practice in psychology, there are issues with using it to support causal claims (Green, Ha, & Bullock, 2010). So, I would just suggest a more nuanced tone when interpreting the mediation result. This is still inherently a correlational finding. That said, there could be more here to provide support for this conclusion. I'd be curious what this model looks like when you flip the MV and DV – so what is the overall relationship between IL-pat and OU with and without including interventionist beliefs as a mediator. If the original model is much stronger, that would help some here.

Response: We appreciate this question and agree that it is important to be quite clear about the limitations on any causal interpretation of mediation analyses. Following from the reviewer's comment, we specifically note the points raised by Green, Ha, and Bullock (2010) with respect to the difficulty of establishing causation and the sort of comprehensive research program required to justify strong causal claims, which is obviously beyond the scope of a single study. In general, we have revised the manuscript throughout to minimize the use of causal language, clarify the ultimately correlational nature of mediation analysis, and explicate the fact that statistical mediation does not establish causation.

This is first addressed in the initial instance of mediation within the results on page 15:

UO was also significantly correlated with both IB (U.S.: $r = 0.48, P < 0.001$) and Belief Change (U.S. sample: $r = 0.36, P < 0.001$; Fig. 3., b path). Consistent with our hypotheses, UO significantly mediated the effects of IL-pat on both IB (indirect effect $P = 0.005$) and Belief Change (indirect effect $P = 0.006$; bootstrapped bias-corrected 95% CI; Fig. 3) in the U.S. sample. **These analyses**

do not establish causation because mediation analysis is fundamentally correlational.

We then expand on this to emphasize that causal claims cannot be established via mediation analysis in the Discussion on page 24:

It is important to emphasize that the mediation analyses we conducted to test models of the hypothesized pathway should not be taken as establishing causation. Mediation is fundamentally a correlation-based technique, and is frequently over-interpreted to make causal claims¹⁰⁰⁻¹⁰². Establishing causation requires a comprehensive research program that should ideally include longitudinal and intervention-based paradigms, a careful accounting of the influence of a host of measured and unmeasured mediating variables, and replication across varied experimental paradigms¹⁰⁰. While this is beyond the scope of a single study, the pathway of effect modeled in the present study provides a framework for more comprehensive investigations of causation in the observed relationships, and the present data on belief change from childhood to adulthood suggest that longitudinal study might be particularly informative.

More specifically related to the point the reviewer raises, we have added and revised fairly substantially in order to more clearly motivate and specify the hypothesized directionality of the pathway of effect modeled in our mediation analyses. Relevant new text is now included in the Results section on page 14:

Having identified relationships between IL-pat and explicit belief, we sought to further test the hypothesis that intuitions about order in the universe might mediate these relationships. Given prior work indicating that IL-pat contributes to intuitions of order^{17,20-23} and that intuitions contribute to religious belief^{1,2,10-14,16,32}, we hypothesized a pathway of effect in which IL-pat contributes to intuitions of order that incline individuals toward religious belief narratives. **The hypothesized directionality of this pathway was also based on theoretical considerations, e.g., that intuitions of order can plausibly develop from implicit learning of order even without explicit learning of religious narratives, and that the content of what is learned via IL-pat (i.e., order itself) is conceptually closer to intuitions directly concerning order than to religious beliefs about interventionist deities (see Discussion).**

And a more expanded treatment is now presented in the Discussion on page 23:

Regarding the specific directional pathway represented by our mediation models, this directionality is based on a hypothesized pathway of effect within which there are *a priori* reasons to position intuitions of universal order prior to interventionist belief (i.e., as the mediator and dependent variable respectively). Specifically, the directionality of the pathway is based on the *a priori* consideration that implicit learning of patterns/order in

environmental stimuli is less likely to directly yield specific beliefs about deities, and more likely to yield a general sense/intuition that there is ambient order. Broad intuitions about the presence of order do not depend on externally learned narratives about the identities and powers of deities, and thus appear more likely to arise intrinsically (i.e., within the individual, directly from that individual's bottom-up implicit learning of patterns/order). Relatedly, intuitions of order appear more conceptually proximate to the content being learned via implicit learning of order than to the content of beliefs about deistic intervention (i.e., both the intuitions and the implicitly learned content directly concern order itself, rather than explanations of order). Thus, a progression from IL-pat to UO appears more likely than a direct leap from implicit learning to religious narratives about interventionist deities. As noted above, the hypothesized pathway of effect is also based on prominent extant accounts, developed to interpret decades of research on implicit learning, which indicate that IL-pat gives rise to intuitions of order^{17,20-25}. Thus, in view of theoretical considerations and extant literature, we hypothesized a pathway of effect whereby implicit learning influences UO and UO influences interventionist belief (i.e., implicit learning yields broad intuitions of order that predispose individuals toward belief narratives that fit these intuitions).

This new text is in addition to text that was largely retained from the original manuscript that addresses motivations for the hypothesized directional pathway.

In the Introduction on pages 3-5:

Several theories of human cognition delineate trajectories of influence by which bottom-up processes direct individuals toward the formation of certain explicit beliefs^{17,20-25}. According to these accounts, intuitions (i.e., reportable experiences of knowledge that was not consciously learned) develop as products of bottom-up gathering of environmental signals/information via unconscious information processing²⁰⁻²⁵. Because individuals are not aware of such bottom-up influences, intuitions drawn from unconscious processing may instead be consciously interpreted via explicit belief narratives that provide a rationalized context for beliefs and behaviors^{2,23,24}. Indeed, intuitions frequently bias more explicit top-down views and judgements^{2,13,17,26}, and certain explicit beliefs may be more compelling and difficult to override when they stem from intuitive impressions^{2,13,17,27}. These effects have been found to operate across diverse modalities of sensory information processing²⁸⁻³⁰. For example, in the context of interpersonal evaluations, humans rely on rapid, nonconscious face processing to form intuitions of trustworthiness²⁹⁻³¹, which has substantial influence on subsequent decision-making³¹.

With respect to religious belief, extant theoretical accounts^{1,2,9,10,13-16,32} posit that evolved neurocognitive processes contribute to “default intuitions” about the nature of environmental stimuli which direct individuals toward explicit beliefs. **For instance, neurobiology evolved for cooperative interactions**

among humans is implicated as a basis for intuitions concerning morality and fairness^{2,10,15}. These intuitions, in turn, are theorized to bias individuals toward religious beliefs in divine justice and watchful gods, which align with moral intuitions, and may insulate against challenges to intuitions about justice when commensurate retribution or compensation for a given act is not materially available³³. Other theoretical work has suggested a connection of evolutionarily-favored pathogen-avoidance mechanisms (e.g., evolved avoidance of individuals who are sick to minimize transmission of pathogens) to intuitions concerning cleanliness and religious beliefs concerning purity (e.g., belief in the healing effects of touching holy individuals or items, belief in unseen agents of spiritual corruption)^{16,34}. Intuitions of anthropomorphism, which appear to be biologically rooted in systems evolved to support face processing and social-cognition³⁵, may contribute to beliefs in watchful invisible agents^{36,37}. Relatedly, the automatic bias to detect agency in the environment is thought to yield over-attribution of intentional agency at the intuitive level (i.e., intuitions that non-agentic things have agency), which supports explicit beliefs about supernatural, intelligent agents (e.g., deities)^{1,13,14}. Empirical support for this theory, however, has been mixed; recent work has suggested that individuals may *under* attribute agency³⁸, and that believers in the paranormal – but not those who endorse a set of more traditional religious beliefs – display illusory agency detection^{39,40}.

And later in the Introduction on pages 6-7:

In view of the associations between intuition and religious beliefs, it is notable that multiple lines of work in cognitive science have specifically implicated order-related perceptual information processing as a basis of intuitions^{21-23,54}. In particular, this work points to bottom-up learning of predictive order (patterns) in the environment without conscious awareness (i.e., implicit pattern learning; IL-pat)^{20-23,25,54} as an underlying influence. Reber²¹ identifies IL-pat as a bottom-up perceptual basis of intuitions of order, including “tacit-knowledge” of complex visuospatial patterns after extended exposure. This work is situated within a broader framework developed by Epstein and others in which, “The implication of automatic implicit learning from experience is that the information acquired... is the primary source of intuitive ‘knowing without knowing how one knows’”²³, and in which such intuition serves as a bridge between implicit and explicit levels of cognition^{23,24,54}. Convergently, research into individual differences in IL-pat⁵⁵⁻⁵⁸ has demonstrated that IL-pat is positively associated with **self-reported measures of intuition across several paradigms.**

The work identifying bottom-up information processing and intuitions rooted in evolved neurocognitive systems as influences on belief suggests the hypothesis that processing information related to order in the environment via IL-pat could influence the formation of belief in an intervening/ordering god. Specifically, accounts implicating IL-pat as a basis of intuitions^{21-23,54}, and intuition as a basis of beliefs^{2,4,9,15,17,24}, indicate a potential pathway of effect whereby implicit learning of actual patterns/order in the environment could yield

intuitions of ambient order that, in turn, influence belief in an ordering god. In other words, intuitions of order that result from IL-pat might disproportionately direct stronger implicit learners toward explicit beliefs in an intervening/ordering god.

Implicit pattern learning occurs frequently in real-world settings and operates on a broad range of stimuli, including many forms of visual modality input⁵⁹; merely being in an environment with recurring elements is sufficient to implicitly learn associations^{60,61}. **Individual differences in implicit learning emerge early in development^{60,62,63}, with performance approximating adult levels⁶⁴. While extended longitudinal measurement is challenging, extant evidence suggests that differences in IL-pat are relatively stable across time⁶⁵, including during early childhood⁶⁶, and may be genetically determined^{67,68}.** Thus, any impacts of IL-pat on intuitions of order are likely to begin in young childhood via exposure to learnable patterns in the environment. If IL-pat affects belief, it could thus manifest in individual differences in change in strength of religious belief from childhood to adulthood.

Relevant text also addresses the directional interpretation of the observed association between IL-pat and more explicit constructs based on task attributes of the SRTT.

In the Introduction on page 8:

Multiple experimental paradigms have been developed to investigate IL-pat. Perhaps the most widely used measure is the Serial Reaction Time Task (SRTT)^{77,78}, which prior work suggests is reflective of environmental IL-pat²¹, and which has recently been shown to load heavily on a broad implicit learning ability factor⁶⁵. Notably, extant evidence indicates that IL-pat during the SRTT is not improved – and may actually be impaired – by top-down influences, including instructions to look for patterns^{61,69,79,80} (see Methods). In addition, modifications of the SRTT have been devised to limit opportunities for explicit awareness of patterns that could be a precondition for top-down influence^{69,81}. The SRTT thus presents a good experimental measure to investigate putative bottom-up link of IL-pat to belief: stronger IL-pat might contribute to stronger belief, but a belief-related top-down bias to seek patterns is unlikely to strengthen IL-pat, thus, any positive association observed between IL-pat and belief is most likely to be bottom-up.

In the Methods on page 30:

In addition, the opportunity to explicitly recall/rehearse target sequences in an effort to explicitly identify patterns for patterns was mitigated by removing the response-stimulus-interval (RSI) between key-press and the appearance of the next target⁸¹. During no-RSI versions of the SRTT, the next target appears immediately after the correct response is made; participants cannot use time between target appearances (because the next target appears immediately) to consciously attempt to identify a pattern structure.

And in the Discussion on pages 20-21:

The results of the current study are consistent with a bottom-up pathway of effect by which stronger implicit learning of patterns/order leads to belief in an intervening/ordering god. While top-down pathways are also conceivable whereby stronger belief in an intervening deity leads individuals to more effectively learn implicit patterns, and/or whereby religious individuals may be more likely to search for patterns because of an explicit belief in the ordering influence of a deity, a number of empirically-based indicators make such explanations less likely in this case. First, prior research has demonstrated that explicit searching does not lead to faster responding during the SRTT, and may actually hinder performance^{61,69,79}. For example, Fletcher and colleagues⁷⁹ showed that explicitly directing participants to search for a pattern impeded implicit learning on the SRTT (i.e., slower RT during the task), putatively because this explicit search disrupts elements of task performance that facilitate IL-pat. Thus, even if interventionist belief promotes explicit searching for patterns, it is unlikely that this would have led to improved performance on the SRTT in the present study.

A bottom-up interpretation of the present data is further bolstered by modifications to the SRTT paradigm that were incorporated into the present study. Namely, the sequence advanced (i.e., the next target appeared) immediately after the participant provided a correct response. This so-called no-response-time-interval (no-RSI) version of the SRTT has been demonstrated by Destrebecqz and Cleeremans and others^{69,81} to yield sequence learning without explicit knowledge acquisition. Removing time between target presentations is intended to ensure that learners are consistently engaged in the task and do not have the opportunity to consciously attempt to recall/rehearse the target sequence in an effort to identify patterns while waiting for the next target to appear. Consistent with prior work, we did not observe a significant correlation between sequence learning (i.e., rate of change on pattern sequences relative to random ones) and explicit awareness, indicating that learning was implicit.

As noted in the response to Point 1 above, the directionality of the hypothesized effect pathway is subsequently qualified within the Discussion to acknowledge that it is extremely unlikely to provide a complete accounting of the influences on religious belief formation, and that top-down influences on bottom-up processing are still entirely possible (pages 22-23):

While the bottom-up directionality of effect we have outlined is consistent with prior work on the neurocognitive bases of religious belief^{1,2,9-12,16}, it is not possible – and is not the intent of the present study – to conclusively rule out any degree of top-down influence on the relationship between IL-pat and religious belief. Indeed, a complete rendering of the cognitive and environmental influences on a phenotype as complex as religious belief very likely includes a multiplex of interconnected “loop” architectures between bottom-up and top-

down processes. Examples of top-down influences on automatic perceptual information processing are abundant^{19,23,98}. One pertinent example is research showing that induced “meaning threats” devised to challenge participants’ existential sense of meaning were associated with better implicit learning in an artificial grammar task⁹⁹. Religious belief is associated with a desire for meaning^{4,11,46}, and it is conceivable that more religious participants might tonically feel greater threat to meaning, which may have an influence on at least some forms of IL-pat. It should be noted that the improvement in grammar learning in this prior work was linked to an increased explicit motivation to find grammatical letter strings in the grammar learning task, whereas extant evidence indicates such explicit motivation does not improve learning on the SRTT^{61,69,79}. More directly to the point, even if it is the case that some amount of top-down influence has bearing on implicit learning, this would not necessarily be inconsistent with bottom-up influences of implicit learning on religious belief. While the present study focused on individual differences in a strongly bottom-up IL-pat paradigm to explore the thus far untested relationship of bottom-up IL-pat to belief, the intent of the present research is not to suggest that all bottom-up influences on belief, and perhaps not even the IL-pat studied here, operate entirely independently of top-down processes.

To re-state the above, the directionality modelled in the mediation analysis is not drawn from the result of the model; rather, the model we constructed reflects the theorized directionality of a hypothesized pathway of effect. Empirically, the most crucial results are that IL-pat was correlated with UO, that UO was correlated with IB, and that IL-pat was correlated with IB. Had these correlations not been observed, our theoretical model would not be supported by these data. In addition, if analysis of the model we constructed to reflect the hypothesized directional pathway did not yield a significant indirect effect of IL-pat on IB via UO, that would have indicated against the theorized directionality of the pathway. Thus, while the analyses we conducted are not a basis for making a directional hypothesis, they suggest that the *a priori* directional hypothesis we formulated on theoretical grounds remains plausible *a posteriori* in view of the data.

This is noted in the Results on page 15:

Nonetheless, results suggest that the *a priori* directional hypothesis remains plausible *a posteriori* in view of the data. By contrast, if modeling the pathway had not yielded significant indirect effects, this would have empirically indicated against the hypothesized model.

Based on the reviewers’ suggestion, exploratory analyses were conducted to test a model in which IB was positioned as the mediator and UO as the dependent variable. Additional exploratory models tested the finer-grained relationships between IL-pat, UO, and each of the three belief variables from which IB was derived. We also ran a fully reversed model in which the IV and DV were exchanged (i.e., IB/Belief Change → UO → IL-pat). These exploratory analyses are now noted in the Results on pages 15-16:

Though the hypothesized pathway of effect is based on theory and prior literature, rather than empirical comparisons between competing models, further exploratory analyses (Supplementary Information, Supplementary Figs. 6, 7) in which IB and individual belief measures were modeled as mediators, and in which IB was modeled as the independent variable and IL-pat as the dependent variable, provided modest but consistent additional support for the hypothesized model.

And described in Supplementary Information on pages 17-18:

Results reported in the main text are consistent with an *a priori* model in which intuitions of universal order mediate the effect of IL-pat on IB and Belief Change (in the U.S. sample). While the mediation model was developed on theoretical grounds rather than empirical comparisons of alternative models, we performed additional exploratory analyses to in which alternative models were compared. While absolute model fit statistics are not applicable to these just-specified models, it is possible to derive the proportion of the total effect that is mediated in each model.

First, we tested fully reversed mediation models in which the belief measures were positioned as the independent variables and IL-pat was positioned as the dependent variable (i.e., Belief \rightarrow UO \rightarrow IL-pat; Supplementary Fig. 6.) The model in which IB was the independent variable yielded a nonsignificant indirect effect ($\beta = 0.07$, $P = 0.08$; bootstrapped bias-corrected 95% CI, as compared to $\beta = 0.07$, $P = 0.005$ in the original model) and a smaller proportion of the total effect that was mediated (0.35 compared to 0.48 in the original model). The model in which Belief Change was the independent variable yielded an indirect effect that remained marginally significant but also accounted for a smaller proportion of the total effect ($P = 0.047$; proportion of total effect that was mediated = 0.38 compared to 0.43 in the original model). A model in which IB was positioned as the mediator (rather than the dependent variable) and UO was positioned as the dependent variable (instead of the mediator; Supplementary Fig. 7) yielded an indirect effect ($\beta = .087$, $P = .022$, as compared to $\beta = .090$, $P = 0.005$ in the original model), and the proportion of the total effect that was mediated was again smaller than in the original model (0.44 compared to 0.48 in the original model).

Further exploratory analyses tested whether UO mediated the effects of IL-pat on each of the three individual belief measures from which the IB component score was derived (BDIS, Self Overlap, and World Overlap). These analyses revealed significant indirect effects in each case (all P s ≤ 0.006 , bootstrapped bias-corrected 95% CI). By contrast, models in which the mediator and DV were reversed for each of these belief measures (such that each belief measure was positioned as the mediator, with UO as the dependent variable), revealed nonsignificant indirect effects for the BDIS and World Overlap variables, and smaller proportions of the total effect mediated for all three models (BDIS = 0.33; World Overlap = 0.30; Self

Overlap = 0.36) compared to what was observed with UO as the mediator (BDIS = 0.73; World Overlap = 0.43; Self Overlap = 0.37). Similarly, the fully reversed mediation models with each individual IB measure positioned as the independent variable and IL-pat as the dependent variable yielded a nonsignificant indirect effect for the Self Overlap model, and smaller proportions of the total effect mediated for all three models (BDIS = 0.65 p = .017; World Overlap = 0.36 p = .032; Self Overlap = 0.28) than in the above-described models with IL-pat as the independent variable and each of the three IB measures as dependent variables. Thus, taken as a group, these exploratory analyses suggest modest but consistent empirical support for the hypothesized models relative to alternatives models.

Supplementary Figure 6. Fully reversed mediation model. Alternative models position religious belief variables of a) IB, and b) Belief Change as the IV and implicit pattern learning as the DV. Mediated effects are nonsignificant (a) and weaker (b) than those indicated in Fig. 3 in the main text.

Supplementary Figure 7. Alternative mediation models. Alternative models positioned religious belief variables of a) IB, and b) Belief Change as the MV and

UO as DV. Evidence of mediation is nominally weaker than in the theoretically-based model (Fig. 3) described in the main text.

Thus, while the hypothesized directionality was based on an *a priori* theory, it is perhaps instructive that all alternative models indicated a nominally weaker mediated effect than the originally hypothesized model. Nonetheless, it is primarily because the mediation modeling employed in the present study was intended to assess the viability of the specific hypothesized directional pathway in view of the data, rather than to compare between multiple hypotheses, that we have retained a description of the model and results in the text. To underscore the point, even if switching the DV and MV had yielded a *stronger* indirect effect, the directionality of our originally hypothesized model would still be rooted in theory and viable *a posteriori*. If the reviewers feel strongly on this point, we are open to the possibility of removing the mediation modeling elements of the manuscript, and focusing only on the concurrence of the relevant bivariate correlations.

Point 4: One would expect an effect of parental beliefs on the development of children's beliefs in an organized and ordered universe. Therefore, it seems a bit notable that this isn't included in the UO model – given the argument for including it in the IB model. Why is this? What does this relationship look like when you control for parental belief? It may be helpful to have a regression table in the supplement that orders these models and allows for side-by-side comparisons.

Response: We thank the reviewer for this suggestion. We had not previously considered the possibility of parental influence on UO, but we agree with the reviewer that parental influences on such intuitions are entirely possible and worthy of consideration. We have now replaced the previous UO model (which included only schizotypal thinking as a covariate) with one that includes parental belief as an additional regressor. Inclusion of parental belief did not meaningfully change the results from the original model. The below text, which includes a reference to the relevant supplementary regression table, is excerpted from Results (page 15):

Based on indications that schizotypal thinking may be associated with a tendency to perceive order^{86,87}, and to account for potential influences of parental religious belief^{84,85}, we included schizotypal thinking **and parents' religious belief** as covariates in a linear regression model to predict UO. IL-pat remained a significant predictor of UO in this model ($b = 5.12$, $\beta = 0.18$, $SE = 1.90$, $P = 0.008$, **Supplementary Table 10**).

Point 5: You cite Gervais & Norenzayan, 2012 as one of the main papers linking intuition with belief. However, it probably would be good/ transparent to note there are some failed replications for the most prominent manipulation in that paper (Sanchez, et al., 2017; Camerer, et al, 2018).

Response: With thanks, we have now revised to indicate the failed replication of the analytic prime employed by Gervais and Norenzayan (Page 5):

Other literature has suggested a link between top-down, effortful analytic thinking – a propensity to critically examine and override prepotent “automatic”⁴¹ responding – and religious disbelief^{41,42}. Gervais and Norenzayan⁴³ reported that priming analytic thinking reduced strength of religious belief. However, replication attempts have not supported the efficacy of the so-called “analytic prime” employed by Gervais and Norenzayan^{44,45}.

Point 6: You note the zero-order correlations for each belief variable and IL-pat in the supplement, but it would be great to see this in the main text. The $p=.06$ is not concerning given the main result. I was curious/ suspicious when I got to that section that I didn’t see any zero-order correlations; especially with you presenting the other zero orders after the main model, on line 216. It makes sense to control for parental beliefs to try and isolate the cognitive effect, so squashing the reader’s concerns at the onset, and being upfront with the trending relationship (which is just an arbitrary cutoff) seems better to me (personally).

Response: We thank the reviewer for this suggestion. We laughed at ourselves a bit when we read this because, when we were writing up the manuscript, we were on the fence about whether to include the zero-order correlations in the main text or Supplementary Information. A colleague suggested the zero-order correlations might not add enough beyond the IB variable to merit main text status, and that tipped the scales. All that is to say that we agree there is value in presenting the zero order correlations, particularly because it emphasizes how consistent the relationship with implicit pattern learning is across three different measures of belief in each of two samples. We have now moved the relevant text originally included in Supplementary Information into the main text (Results; pages 11-12):

We further performed zero-order correlations (Supplementary Tables 5, 6) with scores on the individual intervening belief measures to confirm that the relationship between IL-pat and IB was not a result of associations with just one or two of the belief measures from which IB was derived. Implicit pattern learning was positively associated with BDIS at a trend level in the U.S. ($r = 0.13, P = 0.06$) and significantly in Afghanistan ($r = 0.19, P = 0.02$). In both samples, implicit pattern learning was also significantly associated with Self Overlap (U.S.: $r = 0.19, P = 0.007$; Afghanistan: $r = 0.19, P = 0.02$) and World Overlap (U.S.: $r = 0.16, P = 0.03$; Afghanistan: $r = 0.26, P = 0.001$). These findings indicated a consistent underlying association between implicit pattern learning and diverse individual measures of belief in an interventionist god across these two culturally disparate samples.

Point 7: One curiosity I have is if there is any evidence that IL-pat can be heritable. My thoughts here are that the main mechanism being cited is biologically based and thus could easily be heritable. If this assumption is true, parents' IL-pat would not be fully independent from their children's – unless I'm missing something. I think it would be great to have one or two sentences somewhere in the discussion. I know that this is very minor given the zero-order I just noted in #2; however, something that helps an overly cautious reader realize that this is not a concern and help guide their understanding here would be welcomed.

Response: This is an interesting consideration, and there is some work to suggest that implicit learning is at least partially genetically determined (thus, it may be heritable). We now explicitly note the possibility of heritable individual differences in IL-pat in the Introduction (page 7):

Individual differences in implicit learning emerge early in development^{60,62,63}, with performance approximating adult levels⁶⁴. While extended longitudinal measurement is challenging, extant evidence suggests that differences in IL-pat are relatively stable across time⁶⁵, including during early childhood⁶⁶, and may be genetically determined^{67,68}. Thus, any impacts of IL-pat on intuitions of order are likely to begin in young childhood via exposure to learnable patterns in the environment. If IL-pat affects belief, it could thus manifest in individual differences in change in strength of religious belief from childhood to adulthood.

This point is now also included in the Discussion (pages 21-22):

The observed association between IL-pat and change in belief from childhood to adulthood can be taken as further indication against the likelihood of religious belief exerting a top-down influence on IL-pat. Change in belief across the lifespan, and particularly from childhood to adulthood, is not uncommon^{41,88}. Thus, to the extent that individual differences in implicit learning are present at early ages^{60,62,63}, and may be stable across time⁶⁴⁻⁶⁶, it appears more likely that differences in IL-pat could drive changes in belief across development than that changes in belief, or adult level of belief in an interventionist god as measured in the present study, are primary drivers of differences in IL-pat. Evidence of genetic influences on IL-pat^{67,68}, further suggests sources of individual differences in IL-pat that may be present prior to, and largely independent of, top-down influences.

If IL-pat is partially heritable, this would appear to be generally consistent with a bottom-up, rather than top-down, interpretation of the link between IL-pat and belief (inherited/genetic sources of individual differences are unlikely to be influenced by top-down effects of religious belief). If the IL-pat of parents is somewhat similar to that of their children, and if IL-pat does indeed influence belief, then it is conceivable that participants in the present study who exhibited stronger IL-pat might have also had more religious parents (and having more religious parents could certainly influence one's level

of belief). The parental belief covariate in the present study should largely account for any such influences. If the reviewer's comment is referring to the potential for this kind of indirect influence of parental IL-pat, and if the reviewer feels this should be more clearly addressed in the manuscript, a note can be added to the Discussion to raise this possibility and mention that the parental belief covariate reduces such potential confounding effects. Alternatively, if there is an element of this point that we are not fully understanding, we would appreciate help from the reviewer to further clarify the issue so that we can consider how best to address it.

****REVIEWERS' COMMENTS:**

Reviewer #1 (Remarks to the Author):

Thank you for your diligence in your revisions, and your willingness to provide more open access to your data and code. Your effort to engage with and respond to my comments is unusually conscientious, for which you deserve commendation. I look forward to seeing this work in print, and for any future work on the same theme.

Reviewer #2 (Remarks to the Author):

The authors have done a very thorough job of responding to the concerns I raised earlier, and I'm happy to recommend publication of the new version of their manuscript.

Best wishes,
Ryan McKay

Reviewer #3 (Remarks to the Author):

Overall I am very impressed with the changes made to this manuscript. I feel the authors have made substantial efforts to move it in the direction of an interesting, high quality, and theoretically important contribution. I have only two or so additions I would like to see. Provided these fairly simple changes can be addressed (my comments in bold below), I would be happy to support its publication.

Point 1: Response a: We thank the reviewer for this thoughtful question. We agree that this is an interesting possibility, and indeed that the reviewer proposes a theoretically plausible mechanism that could be at work related to “reflection” as measured in the CRT. As we understand it, the most central question for evaluating this possibility is whether there is a relationship between reflection as measured by the CRT (i.e., the engagement of effortful analytic/reflective thinking to override default prepotent responding, at least in the context of calculation-based problem solving) and implicit learning (which the reviewer refers to as “intuition-based information processing”). A second question is whether the particular attributes of the SRTT paradigm employed in this study are resistant to top-down influence (whether as help or hindrance).

To address the first question as directly as possible, we conducted additional data collection. All U.S. participants who had completed the full study battery were re-contacted and asked to complete the CRT. The data and results are described in the below excerpts from the revised manuscript. To summarize, these new data were helpful in assessing for the first time the relationship of CRT to implicit pattern learning, and specifically in determining that the kind of analytic vs. “intuitive” thinking measured by the CRT was not related to IL-pat, and did not account for the main results we observed. As noted in the excerpted text, IL-pat remained a significant predictor of IB, Belief Change, and UO even

when controlling for CRT performance.

Point 1: Response a: This is excellent. I appreciate that you took the time, and effort, to collect this additional data. To my mind, this moves these results up in value. Give much of the work on this topic uses the CRT, we are now able to more directly contrast this relationship with the IL-pat measure. I found these results fascinating and initially quite surprising. I would have assumed there would be some relationship between CRT scores (especially intuition scores) and performance on the IL-pat. Though the authors have noted this is the first paper to contrast the two, I'd advocate for finding a way to highlight this more in the intro or discussion – but that's just based off of my own interests.

Further, I'm surprised there was such a small relationship between the three key belief variables and either CRT score, given past work on related constructs. Two things that would be helpful, and address a potential concern of a reader familiar with this work: (i) note the little to non-relationship between CRT and the three key belief variables in the actual manuscript; and (ii) perhaps add in the new "Existence Belief" variable (see below for comments) to the table in the Supplemental Appendix table 14. – to give a broader understanding of how this more "standard" variable compares. Ideally, we will see Existence Belief correlates with the three key belief variables, as we would expect; and the CRT correlates with Existence Belief, as we would expect. Then we have good evidence that (i) where the variables can be matched to existing literature, these data match what we would expect; and (ii) the faint relationship between CRT and IB and the non-relationship between IL-pat and Existence Belief means there is something really interesting and new about this version of belief and this novel measure (novel in the current context).

Point 1: Response b: Regarding the second question, the manuscript highlights task characteristics of the employed version of the SRTT intended to decrease the likelihood that differences in top-down searching (due to analytic thinking or otherwise) influenced task performance. These task characteristics, and their relevance for interpreting the data with respect to potential top-down influence, are addressed at several points in the revised manuscript.

Nonetheless, the directionality of the hypothesized effect pathway is subsequently qualified within the Discussion to acknowledge that the proposed pathway is unlikely to provide a complete accounting of the influences on religious belief formation, and that top-down influences on bottom-up processing are still entirely possible (pages 22-23):

Point 1: Response b: This seems reasonable to me, given the non-correlations with CRT and the response to Reviewer 2 regarding perceived patterns.

Point 2: Response: We greatly appreciate this question. To summarize the lengthy response that follows, exploratory analyses indicated that the reviewer's insight was quite correct – a subset of participants who reported believing in a god did not hold strong interventionist belief. The reviewer's comment prompted us to investigate a variable we refer to in the revised Supplementary Information as Existence Belief. This variable is closest to the kind of single-item measurement that the reviewer refers to (because it was not obtained within the context of a survey on belief change). We had not given close

consideration to this variable previously because belief (and especially our target construct of Interventionist Belief) could be measured much more meaningfully with the multiple measures combined for the IB outcome than by a single item measure. This item (initially conceived as part of a “Belief – Doubt” variable as described below) was primarily included in the test battery in relation to other research questions not directly concerning Interventionist Belief. Additionally, we anticipated that the Belief – Doubt measure was unlikely to be viable in Dari translation among Afghan Muslim participants, and this was the case. No acceptable translation for Muslim Dari speakers was found to convey the intended sense of belief (i.e., specifically relative to doubt) with respect to the existence of God, and experimenters reported that participants found questions concerning uncertainty that God exists to be insulting and/or “dangerous.” By contrast, questions concerning the level of interventionist belief were more amenable to Afghan data collection because they did not involve an implication of non-existence. Thus, even if it had been theoretically desirable, it would not have been possible to include this measure in the models involving equivalent variables in the two main samples. Nonetheless, supplementary analyses have now been conducted to investigate the relationship of Existence Belief to other key variables in the U.S. sample. These analyses are excerpted below from Supplementary Information. In investigating this variable for the revised manuscript, we found that it was omitted from the list of additional variables in the original Supplementary Information. This prompted a close re-checking to determine whether this was the only omitted variable. One other variable, which was also a single-item measure superseded by a richer assessment, was also found. Specifically, the new Supplementary Information now lists a single-item measure that asked whether participants changed in belief in God since childhood. This measure, which was obviated by the richer Belief Change questionnaire discussed throughout the manuscript, was included largely because we were unsure whether the more complex Belief Change measure would be viable, especially in Dari among Afghan participants.

Exploring the Existence Belief variable has led to some enlightening results, very much in line with the reviewers’ insight that some individuals believe in a god without believing in the interventionist/ordering influence of a god. These analyses have helped to enrich our understanding of the particular relationship of implicit learning to interventionist belief in our sample. In particular, these analyses have served to strengthen and focus the evidence that implicit learning was specifically related to interventionist belief by distinguishing it from non-interventionist belief. The Existence Belief measure and relevant analyses are now described in Supplementary Information on pages 10-13:

Point 2: Response: I appreciate the authors going back and identifying the variables that were not initially noted in the main manuscript, as well as generating this “Existence Belief” variable of high interest. I also can appreciate the pragmatic logic of not asking sensitive questions in highly religious contexts.

That said, I noticed there was no mention of this fairly important result in the main manuscript. I understand the focus of this paper is on a particular kind of belief (IB); however, this work is building on a larger literature, focused predominantly on investigating the relationship between intuitive cognition and a broader variant of belief – Existence Belief. Thus, I think it is important to note that, with regard to belief in the existence of a god, a superordinate belief category that interventionist beliefs fall within, the IL-pat does not seem to have predictive power.

I appreciate the extensive investigation carried out in the SI on the relationship between EB and IB. I don't think anything to this level needs to be added to the main manuscript - only that readers are clear which relationships work, and which do not. I think as long as this is clearly noted in the manuscript, there is still much to take from this data and the paper should still be published.

Note: that the Supplementary Figure 4. references colored boxes, however they are all black in both the versions I was able to see.

Point 3: Response: We appreciate this question and agree that it is important to be quite clear about the limitations on any causal interpretation of mediation analyses. Following from the reviewer's comment, we specifically note the points raised by Green, Ha, and Bullock (2010) with respect to the difficulty of establishing causation and the sort of comprehensive research program required to justify strong causal claims, which is obviously beyond the scope of a single study. In general, we have revised the manuscript throughout to minimize the use of causal language, clarify the ultimately correlational nature of mediation analysis, and explicate the fact that statistical mediation does not establish causation.

This is first addressed in the initial instance of mediation within the results on page 15:

We then expand on this to emphasize that causal claims cannot be established via mediation analysis in the Discussion on page 24:

More specifically related to the point the reviewer raises, we have added and revised fairly substantially in order to more clearly motivate and specify the hypothesized directionality of the pathway of effect modeled in our mediation analyses. Relevant new text is now included in the Results section on page 14: And a more expanded treatment is now presented in the Discussion on page 23:

This new text is in addition to text that was largely retained from the original manuscript that addresses motivations for the hypothesized directional pathway.

In the Introduction on pages 3-5:

And later in the Introduction on pages 6-7:

In the Introduction on page 8:

In the Methods on page 30:

And in the Discussion on pages 20-21:

As noted in the response to Point 1 above, the directionality of the hypothesized effect pathway is subsequently qualified within the Discussion to acknowledge that it is extremely unlikely to provide a complete accounting of the influences on religious belief formation, and that top-down influences on bottom-up processing are still entirely possible (pages 22-23):

To re-state the above, the directionality modelled in the mediation analysis is not drawn from the result of the model; rather, the model we constructed reflects the theorized directionality of a hypothesized pathway of effect. Empirically, the most crucial results are that IL-pat was correlated with UO, that UO was correlated with IB, and that IL-pat was correlated with IB. Had these correlations not been observed, our theoretical model would not be supported by these data. In addition, if analysis of the model we constructed to reflect the hypothesized directional pathway did not yield a significant indirect effect of IL-pat on IB via UO, that would have indicated against the theorized directionality of the pathway. Thus, while the analyses we conducted are not a basis for making a directional hypothesis,

they suggest that the a priori directional hypothesis we formulated on theoretical grounds remains plausible a posteriori in view of the data.

Based on the reviewers' suggestion, exploratory analyses were conducted to test a model in which IB was positioned as the mediator and UO as the dependent variable. Additional exploratory models tested the finer-grained relationships between IL-pat, UO, and each of the three belief variables from which IB was derived. We also ran a fully reversed model in which the IV and DV were exchanged (i.e., IB/Belief Change \diamond UO \diamond IL-pat). These exploratory analyses are now noted in the Results on pages 15-16: And described in Supplementary Information on pages 17-18:

... A model in which IB was positioned as the mediator (rather than the dependent variable) and UO was positioned as the dependent variable (instead of the mediator; Supplementary Fig. 7) yielded an indirect effect ($\beta = .087$, $P = .022$, as compared to $\beta = .090$, $P = 0.005$ in the original model), and the proportion of the total effect that was mediated was again smaller than in the original model (0.44 compared to 0.48 in the original model)... 

Thus, while the hypothesized directionality was based on an a priori theory, it is perhaps instructive that all alternative models indicated a nominally weaker mediated effect than the originally hypothesized model. Nonetheless, it is primarily because the mediation modeling employed in the present study was intended to assess the viability of the specific hypothesized directional pathway in view of the data, rather than to compare between multiple hypotheses, that we have retained a description of the model and results in the text. To underscore the point, even if switching the DV and MV had yielded a stronger indirect effect, the directionality of our originally hypothesized model would still be rooted in theory and viable a posteriori. If the reviewers feel strongly on this point, we are open to the possibility of removing the mediation modeling elements of the manuscript, and focusing only on the concurrence of the relevant bivariate correlations.

Point 3: Response: The italicized section above is what I wanted to see. This (and much of the additional analysis conducted here) do a good job of supporting the claim that the hypothesized model presented in the manuscript is the strongest ordering. As the authors touch on above in one of the additions, this is what you would want to see to back the path their narrative is claiming.

I appreciate that the authors' effort in adding much additional content. I also appreciate that they have made clear the theoretical motivation for their key path, I think it is still of value to show the other orderings and see if there is empirical evidence for only this theorized path, or if there are other potential options. I'm glad to see that (at least in this data set, and although only slightly) their model shows the strongest result.

Point 4: Response: We thank the reviewer for this suggestion. We had not previously considered the possibility of parental influence on UO, but we agree with the reviewer that parental influences on such intuitions are entirely possible and worthy of consideration. We have now replaced the previous UO model (which included only schizotypal thinking as a covariate) with one that includes parental belief as an additional regressor. Inclusion of parental belief did not meaningfully change the results from the original model. The below text, which includes a reference to the relevant supplementary regression table, is excerpted from Results (page 15):

Point 4: Response: Interesting. Like I noted below, and given the author's response to the heritability comment – I think it's quite interesting that there is little effect of the addition on parental beliefs.

Point 5: Response: With thanks, we have now revised to indicate the failed replication of the analytic prime employed by Gervais and Norenzayan (Page 5):

Point 5: Response: Good

Point 6: Response: We thank the reviewer for this suggestion. We laughed at ourselves a bit when we read this because, when we were writing up the manuscript, we were on the fence about whether to include the zero-order correlations in the main text or Supplementary Information. A colleague suggested the zero-order correlations might not add enough beyond the IB variable to merit main text status, and that tipped the scales. All that is to say that we agree there is value in presenting the zero order correlations, particularly because it emphasizes how consistent the relationship with implicit pattern learning is across three different measures of belief in each of two samples. We have now moved the relevant text originally included in Supplementary Information into the main text (Results; pages 11-12):

Point 6: Response: Thank you. I think more results like this should be presented in main manuscripts. It's clear the relationship is as you would expect, it's just that the P value fell on the other side of an arbitrary line.

Point 7: Response: This is an interesting consideration, and there is some work to suggest that implicit learning is at least partially genetically determined (thus, it may be heritable). We now explicitly note the possibility of heritable individual differences in IL-pat in the Introduction (page 7):

This point is now also included in the Discussion (pages 21-22):

If IL-pat is partially heritable, this would appear to be generally consistent with a bottom-up, rather than top-down, interpretation of the link between IL-pat and belief (inherited/genetic sources of individual differences are unlikely to be influenced by top-down effects of religious belief). If the IL-pat of parents is somewhat similar to that of their children, and if IL-pat does indeed influence belief, then it is conceivable that participants in the present study who exhibited stronger IL-pat might have also had more religious parents (and having more religious parents could certainly influence one's level of belief). The parental belief covariate in the present study should largely account for any such influences. If the reviewer's comment is referring to the potential for this kind of indirect influence of parental IL-pat, and if the reviewer feels this should be more clearly addressed in the manuscript, a note can be added to the Discussion to raise this possibility and mention that the parental belief covariate reduces such potential confounding effects. Alternatively, if there is an element of this point that we are not fully understanding, we would appreciate help from the reviewer to further clarify the issue so that we can consider how best to address it.

Point 7: Response: My comment here was asking about influence of parental IL-pat. I agree including

the parental belief variable, though not perfect, addresses much of my question. I would just advocate adding a note in the discussion to address the possibility of such an interesting relationship for future work to investigate.

Reviewer #1 (Remarks to the Author):

Thank you for your diligence in your revisions, and your willingness to provide more open access to your data and code. Your effort to engage with and respond to my comments is unusually conscientious, for which you deserve commendation. I look forward to seeing this work in print, and for any future work on the same theme.

We greatly appreciate this reviewer's revisions and feedback on our manuscript. Their thoughts have certainly improved the quality of the work.

Reviewer #2 (Remarks to the Author):

The authors have done a very thorough job of responding to the concerns I raised earlier, and I'm happy to recommend publication of the new version of their manuscript.

Best wishes,

Ryan McKay

We are very appreciative of this Reviewer's detailed comments during the first revision stage. Particularly, his suggestion to incorporate Belief in Science has allowed us to form more specific conclusions about our findings.

Reviewer #3 (Remarks to the Author):

Overall I am very impressed with the changes made to this manuscript. I feel the authors have made substantial efforts to move it in the direction of an interesting, high quality, and theoretically important contribution. I have only two or so additions I would like to see. Provided these fairly simple changes can be addressed (my comments in bold below), I would be happy to support its publication.

Thanks to this reviewer for this positive feedback. Our responses to the requested additional changes are indicated below.

Point 1: Response a: We thank the reviewer for this thoughtful question. We agree that this is an interesting possibility, and indeed that the reviewer proposes a theoretically plausible mechanism that could be at work related to "reflection" as measured in the CRT. As we understand it, the most central question for evaluating this possibility is whether there is a relationship between reflection as measured by the CRT (i.e., the engagement of effortful analytic/reflective thinking to override default prepotent responding, at least in the context of calculation-based problem solving) and implicit learning (which the reviewer refers to as "intuition-based information processing"). A second question is whether the particular attributes of the SRTT paradigm employed in this study are resistant to top-down influence (whether as help or hindrance).

To address the first question as directly as possible, we conducted additional data collection. All U.S. participants who had completed the full study battery were re-contacted and asked to complete the CRT. The data and results are described in the below excerpts from the revised manuscript. To summarize, these new data were helpful in assessing for the first time the relationship of CRT to implicit pattern learning, and specifically in determining that the kind of analytic vs. "intuitive" thinking measured by the CRT was not related to IL-pat, and did not account for the main results we observed. As noted in the excerpted text, IL-pat remained a significant predictor of IB, Belief Change, and UO even when controlling for CRT performance.

Point 1: Response a: This is excellent. I appreciate that you took the time, and effort, to collect this additional data. To my mind, this moves these results up in value. Give much of the work on this topic uses the CRT, we are now able to more directly contrast this relationship with the IL-pat measure. I found these results fascinating and initially quite surprising. I would have assumed there would be some relationship between CRT scores (especially intuition scores) and performance on the IL-pat. Though the authors have noted this is the first paper to contrast the two, I'd advocate for finding a way to highlight this more in the intro or discussion – but that's just based off of my own interests. Further, I'm surprised there was such a small relationship between the three key belief variables and either CRT score, given past work on related constructs. Two things that would be helpful, and address a potential concern of a reader familiar with this work: (i) note the little to non-relationship between CRT and the three key belief variables in the actual manuscript; and (ii) perhaps add in the new "Existence Belief" variable (see below for comments) to the table in the Supplemental Appendix table 14. – to give a broader understanding of how this more "standard" variable compares. Ideally, we will see Existence Belief correlates with the three key belief variables, as we would expect; and the CRT correlates with Existence Belief, as we would expect. Then we have good evidence that (i) where the variables can be matched to existing literature, these data match what we would expect; and (ii) the faint relationship between CRT and IB and the non-relationship between IL-pat and Existence Belief means there is something really interesting and new about this version of belief and this novel measure (novel in the current context).

We agree with this reviewer that the relationship between the CRT and IL-Pat and religious belief variables was an interesting addition to the manuscript. However, we feel that further expanding upon these associations is somewhat outside of the main scope of the paper, and may distract from the focus/narrative to some extent, especially for a broad readership not deeply immersed in the literature concerning CRT and/or SRTT. As currently written, we do describe associations between the CRT and IB and Belief Change in the results section (lines 340-346) and discussion (lines 473-494), and believe that this is sufficient for most readers familiar with prior work relating analytic/intuitive thinking to religious belief. Interested readers can also find additional information in Supplementary Information, which includes correlations between CRT and religious belief variables as well as results of linear regression models with the CRT as an added covariate.

Similarly, we agree that the exploration of the relationship between IL-Pat and Existence Belief that was performed as part of the revision process was instructive regarding the relationship between implicit pattern learning and belief (i.e., the critical aspect of intervening belief). Nonetheless, we feel that moving this content to the main text might distract readers (especially the broad readership of *Nature Communications*, many of whom are not deeply familiar with the belief-related cognition literature) from the primary focus of the paper: the relationship between implicit learning of patterns, intuitions of order, and belief in a plausible cause of such order (an intervening God). However, if the editor (or reviewer) has a strong preference on the above points, we are open to further revision.

Point 1: Response b: Regarding the second question, the manuscript highlights task characteristics of the employed version of the SRTT intended to decrease the likelihood that differences in top-down searching (due to analytic thinking or otherwise) influenced

task performance. These task characteristics, and their relevance for interpreting the data with respect to potential top-down influence, are addressed at several points in the revised manuscript.

Nonetheless, the directionality of the hypothesized effect pathway is subsequently qualified within the Discussion to acknowledge that the proposed pathway is unlikely to provide a complete accounting of the influences on religious belief formation, and that top-down influences on bottom-up processing are still entirely possible (pages 22-23):

Point 1: Response b: This seems reasonable to me, given the non-correlations with CRT and the response to Reviewer 2 regarding perceived patterns.

We appreciate this reviewer's suggestions to incorporate the CRT.

Point 2: Response: We greatly appreciate this question. To summarize the lengthy response that follows, exploratory analyses indicated that the reviewer's insight was quite correct – a subset of participants who reported believing in a god did not hold strong interventionist belief. The reviewer's comment prompted us to investigate a variable we refer to in the revised Supplementary Information as Existence Belief. This variable is closest to the kind of single-item measurement that the reviewer refers to (because it was not obtained within the context of a survey on belief change). We had not given close consideration to this variable previously because belief (and especially our target construct of Interventionist Belief) could be measured much more meaningfully with the multiple measures combined for the IB outcome than by a single item measure. This item (initially conceived as part of a "Belief – Doubt" variable as described below) was primarily included in the test battery in relation to other research questions not directly concerning Interventionist Belief.

Additionally, we anticipated that the Belief – Doubt measure was unlikely to be viable in Dari translation among Afghan Muslim participants, and this was the case. No acceptable translation for Muslim Dari speakers was found to convey the intended sense of belief (i.e., specifically relative to doubt) with respect to the existence of God, and experimenters reported that participants found questions concerning uncertainty that God exists to be insulting and/or "dangerous." By contrast, questions concerning the level of interventionist belief were more amenable to Afghan data collection because they did not involve an implication of non-existence. Thus, even if it had been theoretically desirable, it would not have been possible to include this measure in the models involving equivalent variables in the two main samples.

Nonetheless, supplementary analyses have now been conducted to investigate the relationship of Existence Belief to other key variables in the U.S. sample. These analyses are excerpted below from Supplementary Information. In investigating this variable for the revised manuscript, we found that it was omitted from the list of additional variables in the original Supplementary Information. This prompted a close re-checking to determine whether this was the only omitted variable. One other variable, which was also a single-item measure superseded by a richer assessment, was also found. Specifically, the new Supplementary Information now lists a single-item measure that asked whether participants changed in belief in God since childhood. This measure, which was obviated by the richer Belief Change questionnaire discussed throughout the manuscript, was included largely because we were unsure whether the more complex Belief Change measure would be viable, especially in Dari among Afghan participants.

Exploring the Existence Belief variable has led to some enlightening results, very much in line with the reviewers' insight that some individuals believe in a god without believing in the interventionist/ordering influence of a god. These analyses have helped to enrich our understanding of the particular relationship of implicit learning to interventionist belief in our sample. In particular, these analyses have served to strengthen and focus the evidence that implicit learning was specifically related to interventionist belief by distinguishing it from non-interventionist belief. The Existence Belief measure and relevant analyses are now described in Supplementary Information on pages 10-13:

Point 2: Response: I appreciate the authors going back and identifying the variables that were not initially noted in the main manuscript, as well as generating this "Existence Belief" variable of high interest. I also can appreciate the pragmatic logic of not asking sensitive questions in highly religious contexts.

That said, I noticed there was no mention of this fairly important result in the main manuscript. I understand the focus of this paper is on a particular kind of belief (IB); however, this work is building on a larger literature, focused predominantly on investigating the relationship between intuitive cognition and a broader variant of belief – Existence Belief. Thus, I think it is important to note that, with regard to belief in the existence of a god, a superordinate belief category that interventionist beliefs fall within, the IL-pat does not seem to have predictive power.

I appreciate the extensive investigation carried out in the SI on the relationship between EB and IB. I don't think anything to this level needs to be added to the main manuscript - only that readers are clear which relationships work, and which do not. I think as long as this is clearly noted in the manuscript, there is still much to take from this data and the paper should still be published.

Note: that the Supplementary Figure 4. references colored boxes, however they are all black in both the versions I was able to see.

Similar to our response to Point 1, we also feel that the Existence Belief results, while certainly informative (as this Reviewer points out), are also not of central importance to the manuscript. We wish to maintain a clear focus on the associations between implicit pattern learning and a particular form of belief (i.e., IB), and worry that introducing another type of religious belief may unnecessarily complicate the manuscript. Specialized readerships (or other interested readers) can find this additional information in Supplementary Information. Once again, however, we are open to additional revisions if the editor or reviewer feels strongly about this point.

Regarding the colored boxes in Supplementary Figure 4, we are not completely sure what may have caused this issue (the three boxes originally appeared in 3 different colors), but in order to avoid this problem for other readers, the text now refers to the boxes by their position in the figure (i.e., Left, Middle, Right).

Point 3: Response: We appreciate this question and agree that it is important to be quite clear about the limitations on any causal interpretation of mediation analyses. Following from the reviewer's comment, we specifically note the points raised by Green, Ha, and Bullock (2010) with respect to the difficulty of establishing causation and the sort of

comprehensive research program required to justify strong causal claims, which is obviously beyond the scope of a single study. In general, we have revised the manuscript throughout to minimize the use of causal language, clarify the ultimately correlational nature of mediation analysis, and explicate the fact that statistical mediation does not establish causation.

This is first addressed in the initial instance of mediation within the results on page 15: We then expand on this to emphasize that causal claims cannot be established via mediation analysis in the Discussion on page 24:

More specifically related to the point the reviewer raises, we have added and revised fairly substantially in order to more clearly motivate and specify the hypothesized directionality of the pathway of effect modeled in our mediation analyses. Relevant new text is now included in the Results section on page 14:

And a more expanded treatment is now presented in the Discussion on page 23:

This new text is in addition to text that was largely retained from the original manuscript that addresses motivations for the hypothesized directional pathway.

To re-state the above, the directionality modelled in the mediation analysis is not drawn from the result of the model; rather, the model we constructed reflects the theorized directionality of a hypothesized pathway of effect. Empirically, the most crucial results are that IL-pat was correlated with UO, that UO was correlated with IB, and that IL-pat was correlated with IB. Had these correlations not been observed, our theoretical model would not be supported by these data. In addition, if analysis of the model we constructed to reflect the hypothesized directional pathway did not yield a significant indirect effect of IL-pat on IB via UO, that would have indicated against the theorized directionality of the pathway. Thus, while the analyses we conducted are not a basis for making a directional hypothesis, they suggest that the a priori directional hypothesis we formulated on theoretical grounds remains plausible a posteriori in view of the data.

Based on the reviewers' suggestion, exploratory analyses were conducted to test a model in which IB was positioned as the mediator and UO as the dependent variable. Additional exploratory models tested the finer-grained relationships between IL-pat, UO, and each of the three belief variables from which IB was derived. We also ran a fully reversed model in which the IV and DV were exchanged (i.e., IB/Belief Change \diamond UO \diamond IL-pat). These exploratory analyses are now noted in the Results on pages 15-16:

And described in Supplementary Information on pages 17-18:

.... A model in which IB was positioned as the mediator (rather than the dependent variable) and UO was positioned as the dependent variable (instead of the mediator; Supplementary Fig. 7) yielded an indirect effect ($\beta = .087$, $P = .022$, as compared to $\beta = .090$, $P = 0.005$ in the original model), and the proportion of the total effect that was mediated was again smaller than in the original model (0.44 compared to 0.48 in the original model)...

Thus, while the hypothesized directionality was based on an a priori theory, it is perhaps instructive that all alternative models indicated a nominally weaker mediated effect than the originally hypothesized model. Nonetheless, it is primarily because the mediation modeling employed in the present study was intended to assess the viability of the specific hypothesized directional pathway in view of the data, rather than to compare between

multiple hypotheses, that we have retained a description of the model and results in the text. To underscore the point, even if switching the DV and MV had yielded a stronger indirect effect, the directionality of our originally hypothesized model would still be rooted in theory and viable a posteriori. If the reviewers feel strongly on this point, we are open to the possibility of removing the mediation modeling elements of the manuscript, and focusing only on the concurrence of the relevant bivariate correlations.

Point 3: Response: The italicized section above is what I wanted to see. This (and much of the additional analysis conducted here) do a good job of supporting the claim that the hypothesized model presented in the manuscript is the strongest ordering. As the authors touch on above in one of the additions, this is what you would want to see to back the path their narrative is claiming.

I appreciate that the authors' effort in adding much additional content. I also appreciate that they have made clear the theoretical motivation for their key path, I think it is still of value to show the other orderings and see if there is empirical evidence for only this theorized path, or if there are other potential options. I'm glad to see that (at least in this data set, and although only slightly) their model shows the strongest result.

We agree with this reviewer, and believe that presenting this information in the Supplementary Information for interested readers is a worthwhile addition to the manuscript.

Point 4: Response: We thank the reviewer for this suggestion. We had not previously considered the possibility of parental influence on UO, but we agree with the reviewer that parental influences on such intuitions are entirely possible and worthy of consideration. We have now replaced the previous UO model (which included only schizotypal thinking as a covariate) with one that includes parental belief as an additional regressor. Inclusion of parental belief did not meaningfully change the results from the original model. The below text, which includes a reference to the relevant supplementary regression table, is excerpted from Results (page 15):

Point 4: Response: Interesting. Like I noted below, and given the author's response to the heritability comment – I think it's quite interesting that there is little effect of the addition on parental beliefs.

Indeed, we agree that this is an interesting new finding that came about as a result of the revision process.

Point 5: Response: With thanks, we have now revised to indicate the failed replication of the analytic prime employed by Gervais and Norenzayan (Page 5):

Point 5: Response: Good

This was noted by all the reviewers, so the clarification was certainly necessary.

Point 6: Response: We thank the reviewer for this suggestion. We laughed at ourselves a bit when we read this because, when we were writing up the manuscript, we were on the fence about whether to include the zero-order correlations in the main text or Supplementary Information. A colleague suggested the zero-order correlations might not add enough beyond the IB variable to merit main text status, and that tipped the scales. All that is to say that we agree there is value in presenting the zero order correlations, particularly

because it emphasizes how consistent the relationship with implicit pattern learning is across three different measures of belief in each of two samples. We have now moved the relevant text originally included in Supplementary Information into the main text (Results; pages 11-12):

Point 6: Response: Thank you. I think more results like this should be presented in main manuscripts. It's clear the relationship is as you would expect, it's just that the P value fell on the other side of an arbitrary line.

Thank you again for the requested addition.

Point 7: Response: This is an interesting consideration, and there is some work to suggest that implicit learning is at least partially genetically determined (thus, it may be heritable). We now explicitly note the possibility of heritable individual differences in IL-pat in the Introduction (page 7):

This point is now also included in the Discussion (pages 21-22):

If IL-pat is partially heritable, this would appear to be generally consistent with a bottom-up, rather than top-down, interpretation of the link between IL-pat and belief (inherited/genetic sources of individual differences are unlikely to be influenced by top-down effects of religious belief). If the IL-pat of parents is somewhat similar to that of their children, and if IL-pat does indeed influence belief, then it is conceivable that participants in the present study who exhibited stronger IL-pat might have also had more religious parents (and having more religious parents could certainly influence one's level of belief). The parental belief covariate in the present study should largely account for any such influences. If the reviewer's comment is referring to the potential for this kind of indirect influence of parental IL-pat, and if the reviewer feels this should be more clearly addressed in the manuscript, a note can be added to the Discussion to raise this possibility and mention that the parental belief covariate reduces such potential confounding effects. Alternatively, if there is an element of this point that we are not fully understanding, we would appreciate help from the reviewer to further clarify the issue so that we can consider how best to address it.

Point 7: Response: My comment here was asking about influence of parental IL-pat. I agree including the parental belief variable, though not perfect, addresses much of my question. I would just advocate adding a note in the discussion to address the possibility of such an interesting relationship for future work to investigate.

With apologies, we are not entirely clear what this reviewer is requesting. We endeavored to address the potential heritability of IL-pat with the new text that was added to the Introduction and Discussion. As we understand this point, if IL-pat has a genetic component, then individual differences in implicit pattern learning may be present prior to any potential top-down influences. Therefore, the effects of IL-pat on intuitions of order and belief in an intervening God are likely to begin in young childhood. We do also note that extended longitudinal measurement (e.g., obtaining a measure of IL-pat in parents, then measuring IL-pat in children over time) is challenging. We agree with the reviewer that conducting such research is certainly an interesting direction for future work, although further expanding discussion of this point beyond what is presently included in the manuscript could be somewhat of a digression from the primary hypotheses and findings. Once again, we are open to further discussion on this point if the editor or reviewer feels strongly.